# Label-Imbalanced and Group-Sensitive Classification under Overparameterization

**Ganesh Ramachandra Kini**
University of California, Santa Barbara
kini@ucsb.edu

**Orestis Paraskevas**
University of California, Santa Barbara
orestis@ucsb.edu

**Samet Oymak**
University of California, Riverside
oymak@ece.ucr.edu

**Christos Thrampoulidis**
University of British Columbia
cthrampo@ece.ubc.ca

## Abstract

The goal in label-imbalanced and group-sensitive classification is to optimize relevant metrics such as balanced error and equal opportunity. Classical methods, such as weighted cross-entropy, fail when training deep nets to the terminal phase of training (TPT), that is training beyond zero training error. This observation has motivated recent flurry of activity in developing heuristic alternatives following the intuitive mechanism of promoting larger margin for minorities. In contrast to previous heuristics, we follow a principled analysis explaining how different loss adjustments affect margins. First, we prove that for all linear classifiers trained in TPT, it is necessary to introduce *multiplicative*, rather than *additive*, logit adjustments so that the interclass margins change appropriately. To show this, we discover a connection of the multiplicative CE modification to the cost-sensitive support-vector machines. Perhaps counterintuitively, we also find that, at the start of training, the same multiplicative weights can actually harm the minority classes. Thus, while additive adjustments are ineffective in the TPT, we show that they can speed up convergence by countering the initial negative effect of the multiplicative weights. Motivated by these findings, we formulate the *vector-scaling (VS) loss*, that captures existing techniques as special cases. Moreover, we introduce a natural extension of the VS-loss to group-sensitive classification, thus treating the two common types of imbalances (label/group) in a unifying way. Importantly, our experiments on state-of-the-art datasets are fully consistent with our theoretical insights and confirm the superior performance of our algorithms. Finally, for imbalanced Gaussian-mixtures data, we perform a generalization analysis, revealing tradeoffs between balanced / standard error and equal opportunity.

## 1 Introduction

### 1.1 Motivation and contributions

Equitable learning in the presence of data imbalances is a classical machine learning (ML) problem, but one with increasing importance as ML decisions are adapted in increasingly more complex applications directly involving people [3]. Two common types of imbalances are those appearing in *label-imbalanced* and *group-sensitive* classification. In the first type, examples from a target class are heavily outnumbered by examples from the rest of the classes. The standard metric of average misclassification error is insensitive to such imbalances and among several classical alternatives the *balanced error* is a widely used metric. In the second type, the broad goal is to ensure fairness with respect to a protected underrepresented group

35th Conference on Neural Information Processing Systems (NeurIPS 2021).

(e.g. gender, race). While acknowledging that there is no universal fairness metric [27, 13], several suggestions have been made in the literature including *Equal Opportunity* favoring same true positive rates across groups [15].

Methods for imbalanced data are broadly categorized into data- and algorithm- level ones. In the latter category, belong *cost-sensitive methods* and, specifically, those that modify the training loss to account for varying class/group penalties. Corresponding state-of-the-art (SOTA) research is motivated by observations that classical methods, such as weighted cross-entropy (wCE) fail when training overparameterized deep nets without regularization and with train-loss minimization continuing well beyond zero train-error, in the so-called *terminal phase of training (TPT)* ([43] and references therein). Intuitively, failure of wCE when trained in TPT is attributed to the failure to appropriately adjust the relative margins between different classes/groups in a way that favors minorities. To overcome this challenge, recent works have proposed a so-called logit-adjusted (LA) loss that modifies the cross-entropy (CE) loss by including extra *additive* hyper-parameters acting on the logits [24, 8, 32]. Even more recently, [54] suggested yet another modification that introduces *multiplicative* hyper-parameters on the logits leading to a class-dependent temperature (CDT) loss. Empirically, both adjustments show performance improvements over wCE. However, it remains unclear: *Do both additive and multiplicative hyper-parameters lead to margin-adjustments favoring minority classes? If so, what are the individual mechanisms that lead to this behavior? How effective are different adjustments at each stage of training?*

This paper answers the above questions. Specifically, we argue that multiplicative hyper-parameters are most effective for margin adjustments in TPT, while additive parameters can be useful in the initial phase of training. Importantly, this intuition justifies our algorithmic contribution: we introduce the *vector-scaling (VS) loss* that combines both types of adjustments and attains improved performance on SOTA imbalanced datasets. Finally, using the same set of tools, we extend the VS-loss to instances of group-sensitive classification. We make multiple contributions as summarized below; see also Figure 1.

• **Explaining the distinct roles of additive/multiplicative adjustments.** We show that when optimizing in TPT *multiplicative* logit adjustments are critical. Specifically, we prove for linear models that multiplicative adjustments find classifiers that are solutions to cost-sensitive support-vector-machines (CS-SVM), which by design create larger margins for minority classes. While effective in TPT, we also find that, at the start of training, the same adjustments can actually harm minorities. Instead, *additive* adjustments can speed up convergence by countering the initial negative effect of the multiplicative ones. The analytical findings are consistent with our experiments.

• **An improved algorithm: VS-loss.** Motivated by the unique roles of the two different types of adjustments, we propose the vector-scaling (VS) loss that combines the best of both worlds and outperforms existing techniques on benchmark datasets.

• **Introducing logit-adjustments for group-imbalanced data.** We introduce a version of VS-loss tailored to group-imbalanced datasets, thus treating, for the first time, loss-adjustments for label and group imbalances in a unifying way. For the latter, we propose a new algorithm combining our VS-loss with the previously proposed DRO-method to achieve state-of-the-art performance in terms of both Equal Opportunity and worst-subgroup error.

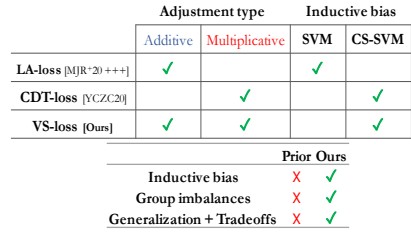

| | Adjustment type | | Inductive bias | |
|---|---|---|---|---|
| | Additive | Multiplicative | SVM | CS-SVM |
| **LA-loss** [MJR+20+++] | ✓ | | ✓ | |
| **CDT-loss** [YCZC20] | | ✓ | | ✓ |
| **VS-loss** [Ours] | ✓ | ✓ | | ✓ |

| | Prior | Ours |
|---|---|---|
| Inductive bias | X | ✓ |
| Group imbalances | X | ✓ |
| Generalization + Tradeoffs | X | ✓ |

**Figure 1:** Summary of contributions.

• **Generalization analysis / fairness trade-offs.** We present a sharp generalization analysis of the VS-loss on binary overparameterized Gaussian mixtures. Our formulae are explicit in terms of data geometry, priors, parameterization ratio and hyperparameters; thus, leading to tradeoffs between standard error and fairness measures. We find that VS-loss can improve both balanced and standard error over CE. Interestingly, the optimal hyperparameters that minimize balanced error also optimize Equal Opportunity.

## 1.2 Connections to related literature

**CE adjustments.** The use of wCE for imbalanced data is rather old [53], but it becomes ineffective under overparameterization, e.g. [6]. This deficiency has led to the idea of *additive* label-based parameters $\iota_y$ on the logits [24, 8, 50, 32, 52]. Specifically, [32] proved that setting $\iota_y = \log(\pi_y)$ ($\pi_y$ denotes the prior of class $y$) leads to a *Fisher consistent* loss, termed LA-loss, which outperformed other heuristics (e.g., focal loss [28]) on SOTA datasets. However, Fisher consistency is only relevant in the large sample size limit. Instead, we focus on overparameterized models. In a recent work, [54] proposed the CDT-loss, which instead uses *multiplicative* label-based parameters $\Delta_y$ on the logits. The authors arrive at the CDT-loss as a heuristic means of compensating for the empirically observed phenomenon of that the last-layer minority features deviate between training and test instances [25]. Instead, we arrive at the CDT-loss via a different viewpoint: we show that the multiplicative weights are necessary to move decision boundaries towards majorities when training overparameterized linear models in TPT. Moreover, we argue that while additive weights are not so effective in the TPT, they can help in the initial phase of training. Our analysis sheds light on the individual roles of the two different modifications proposed in the literature and naturally motivates the VS-loss in (2). Compared to the above works we also demonstrate the successful use of VS-loss in group-imbalanced setting and show its competitive performance over alternatives in [45, 18, 40]. Beyond CE adjustments there is active research on alternative methods to improve fairness metrics, e.g. [23, 56, 29, 41]. These are orthogonal to CE adjustments and can potentially be used in conjunction.

**Relation to vector-scaling calibration.** Our naming of the VS-loss is inspired by the vector scaling (VS) calibration [14], a *post-hoc procedure* that modifies the logits $\mathbf{v}$ *after training* via $\mathbf{v} \to \mathbf{\Delta} \odot \mathbf{v} + \boldsymbol{\iota}$, where $\odot$ is the Hadamard product. [55] shows that VS can improve calibration for imbalanced classes, but, in contrast to VS calibration, the multiplicative/additive scalings in our VS-loss are part of the loss and directly affect training.

**Blessings/curses of overparameterization.** Overparameterization acts as a catalyst for deep neural networks [38]. In terms of optimization, [47, 42, 20, 2] show that gradient-based algorithms are *implicitly biased* towards favorable min-norm solutions. Such solutions, are then analyzed in terms of generalization showing that they can in fact lead to benign overfitting e.g. [4, 16]. While implicit bias is key to benign overfitting it may come with certain downsides. As a matter of fact, we show here that certain hyper-parameters (e.g. additive ones) can be ineffective in the interpolating regime in promoting fairness. Our argument essentially builds on characterizing the implicit bias of wCE/LA/CDT-losses. Related to this, [46] demonstrated the ineffectiveness of $\omega_y$ in learning with groups.

## 2 Problem setup

**Data.** Let training set $\{(\mathbf{x}_i, g_i, y_i)\}_{i=1}^n$ consisting of $n$ i.i.d. samples from a distribution $\mathcal{D}$ over $\mathcal{X} \times \mathcal{G} \times \mathcal{Y}$; $\mathcal{X} \subseteq \mathbb{R}^d$ is the input space, $\mathcal{Y} = [C] := \{1, \ldots, C\}$ the set of $C$ labels, and, $\mathcal{G} = [K]$ refers to group membership among $K \geq 1$ groups. Group-assignments are known for training data, but unknown at test time. For concreteness, we focus here on the binary setting, i.e. $C = 2$ and $\mathcal{Y} = \{-1, +1\}$; we present multiclass extensions in the Experiments and in the Supplementary Material (SM). We assume throughout that $y = +1$ is minority class.

**Fairness metrics.** Given a training set we learn $f_\mathbf{w} : \mathcal{X} \mapsto \mathcal{Y}$ parameterized by $\mathbf{w} \in \mathbb{R}^p$. For instance, linear models take the form $f_\mathbf{w} = \langle \mathbf{w}, h(\mathbf{x}) \rangle$ for some feature representation $h : \mathcal{X} \mapsto \mathbb{R}^p$. Given a new sample $\mathbf{x}$, we decide class membership $\hat{y} = \text{sign}(f_\mathbf{w}(\mathbf{x}))$. The (standard) *risk* or *misclassification error* is $\mathcal{R} := \mathbb{P}\{\hat{y} \neq y\}$. Let $s = (y, g)$ define a subgroup for given values of $y$ and $g$. We also define the *class-conditional risks* $\mathcal{R}_\pm = \mathbb{P}\{\hat{y} \neq y | y = \pm 1\}$, and, the *sub-group-conditional risks* $\mathcal{R}_{\pm,j} = \mathbb{P}\{\hat{y} \neq y | y = \pm 1, g = j\}$, $j \in [K]$. The *balanced error* averages the conditional risks of the two classes: $\mathcal{R}_{\text{bal}} := (\mathcal{R}_+ + \mathcal{R}_-)/2$. Assuming $K = 2$ groups, Equal Opportunity requires $\mathcal{R}_{+,1} = \mathcal{R}_{+,2}$ [15]. More generally, we consider the (signed) *difference of equal opportunity (DEO)* $\mathcal{R}_{\text{deo}} := \mathcal{R}_{+,1} - \mathcal{R}_{+,2}$. In our experiments, we also measure the worst-case subgroup error $\max_{(y \in \pm 1, g \in [K])} \mathcal{R}_{y,g}$.

**Terminal phase of training (TPT).** Motivated by modern training practice, we assume overparameterized $f_\mathbf{w}$ so that $\mathcal{R}_{\text{train}} = \frac{1}{n} \sum_{i \in [n]} \mathbb{1}[\text{sign}(f_\mathbf{w}(\mathbf{x}_i)) \neq y_i]$ can be driven to zero. Typically, training such large models continues well-beyond zero training error as the training loss is being pushed toward zero. As in [43], we call this the *terminal phase of training*.

## 2.1 Algorithms

**Cross-entropy adjustments.** We introduce the **vector-scaling (VS) loss**, which combines *both* additive and multiplicative logit adjustments, previously suggested in the literature in isolation. The following is the **binary VS-loss** for labels $y \in \{\pm 1\}$, weight parameters $\omega_\pm > 0$, additive logit parameters $\iota_\pm \in \mathbb{R}$, and multiplicative logit parameters $\Delta_\pm > 0$:

$$\ell_{\text{VS}}(y, f_{\mathbf{w}}(\mathbf{x})) = \omega_y \cdot \log\left(1 + e^{\iota_y} \cdot e^{-\Delta_y y f_{\mathbf{w}}(\mathbf{x})}\right). \tag{1}$$

For imbalanced datasets with $C > 2$ classes, the **VS-loss** takes the following form:

$$\ell_{\text{VS}}(y, \mathbf{f}_{\mathbf{w}}(\mathbf{x})) = -\omega_y \log\left(e^{\Delta_y \mathbf{f}_y(\mathbf{x}) + \iota_y} \Big/ \sum_{c \in [C]} e^{\Delta_c \mathbf{f}_c(\mathbf{x}) + \iota_c}\right). \tag{2}$$

Here $\mathbf{f}_{\mathbf{w}} : \mathbb{R}^d \to \mathbb{R}^C$ and $\mathbf{f}_{\mathbf{w}}(\mathbf{x}) = [\mathbf{f}_1(\mathbf{x}), \ldots, \mathbf{f}_C(\mathbf{x})]$ is the vector of logits. The VS-loss (Eqns. (1),(2)) captures existing techniques as special cases by tuning accordingly the additive/multiplicative hyperparameters. Specifically, we recover: (i) **weighted CE (wCE) loss** by $\Delta_y = 1, \iota_y = 0, \omega_y = \pi_y^{-1}$; (ii) **LA-loss** by $\Delta_y = 1$; (iii) **CDT-loss** by $\iota_y = 0$.

With the goal of (additionally) ensuring fairness with respect to *sensitive groups*, we extend the VS-loss by introducing parameters $(\Delta_{y,g}, \iota_{y,g}, \omega_{y,g})$ that depend *both* on class and group membership (specified by $y$ and $g$, respectively). Our proposed **group-sensitive VS-loss** is as follows (multiclass version can be defined accordingly):

$$\ell_{\text{Group-VS}}(y, g, f_{\mathbf{w}}(\mathbf{x})) = \omega_{y,g} \cdot \log\left(1 + e^{\iota_{y,g}} \cdot e^{-\Delta_{y,g} y f_{\mathbf{w}}(\mathbf{x})}\right). \tag{3}$$

**CS-SVM.** For linear classifiers $f_{\mathbf{w}}(\mathbf{x}) = \langle \mathbf{w}, h(\mathbf{x}) \rangle$ with $h : \mathcal{X} \to \mathbb{R}^p$, CS-SVM [31] solves

$$\min_{\mathbf{w}} \ \|\mathbf{w}\|_2 \ \text{sub. to} \begin{cases} \langle \mathbf{w}, h(\mathbf{x}_i) \rangle \geq \delta & , y_i = +1 \\ \langle \mathbf{w}, h(\mathbf{x}_i) \rangle \leq -1 & , y_i = -1 \end{cases}, i \in [n], \tag{4}$$

for hyper-parameter $\delta \in \mathbb{R}_+$ representing the ratio of margins between classes. $\delta = 1$ corresponds to (standard) SVM, while tuning $\delta > 1$ (resp. $\delta < 1$) favors a larger margin $\delta/\|\hat{\mathbf{w}}_\delta\|_2$ for the minority vs $1/\|\hat{\mathbf{w}}_\delta\|_2$ for the majority classes. Thus, $\delta \to +\infty$ (resp. $\delta \to 0$) corresponds to the decision boundary starting right at the boundary of class $y = -1$ (resp. $y = +1$).

**Group-sensitive SVM.** The *group-sensitive* version of CS-SVM (GS-SVM), for $K = 2$ protected groups adjusts the constraints in (4) so that $y_i \langle \mathbf{w}, h(\mathbf{x}_i) \rangle \geq \delta$ (or $\geq 1$), if $g_i = 1$ (or $g_i = 2$.) $\delta > 1$, GS-SVM favors larger margin for the sensitive group $g = 1$. Refined versions when classes are also imbalanced modify the constraints to $y_i \langle \mathbf{w}, \mathbf{h}(\mathbf{x}_i) \rangle \geq \delta_{y_i, g_i}$. Both CS-SVM and GS-SVM are feasible iff data are linearly separable (see SM). However, we caution that the GS-SVM hyper-parameters are in general harder to interpret as "margin-ratios".

## 3 Insights on the VS-loss

Here, we shed light on the distinct roles of the VS-loss hyper-parameters $\omega_y, \iota_y$ and $\Delta_y$.

### 3.1 CDT-loss vs LA-loss: Why multiplicative weights?

We first demonstrate the unique role played by the multiplicative weights $\Delta_y$ through a motivating experiment on synthetic data in Fig. 2. We generated a binary Gaussian-mixture dataset of $n = 100$ examples in $\mathbb{R}^{300}$ with data means sampled independently from the Gaussian distribution and normalized such that $\|\boldsymbol{\mu}_{+1}\|_2 = 2\|\boldsymbol{\mu}_{-1}\|_2 = 4$. We set prior $\pi_+ = 0.1$ for the minority class +1. For varying model size values $p \in [5 : 5 : 50, 75 : 25 : 300]$ we trained linear classifier $f_{\mathbf{w}}(x) = \langle \mathbf{w}, h(\mathbf{x}) \rangle$ using only the first $p$ features, i.e. $h(\mathbf{x}) = \mathbf{x}(1 : p) \in \mathbb{R}^p$. This allows us to investigate performance versus the parameterization ratio $\gamma = p/n$. [1] We train the model $\mathbf{w}$ using the following special cases of the VS-loss (Eqn. (1)): (i) *CDT-loss* with $\Delta_+ = \delta_\star^{-1}, \Delta_- = 1$ ($\delta_\star > 0$ is set to the value shown in the inset plot; see SM for details). (ii) *LDAM-loss:* $\iota_+ = \pi^{-1/4}, \iota_- = (1 - \pi)^{-1/4}$ (special case of *LA-loss* [8]). (iii) *LA loss:* $\iota_+ = \log\left(\frac{1-\pi}{\pi}\right), \iota_- = \log\left(\frac{\pi}{1-\pi}\right)$ (Fisher-consistent values [32]). We ran gradient descent and averaged over 25 independent experiments. The *balanced error* was computed on a test set of size $10^4$ and reported values are shown in red/blue/black markers. We also plot the

---

[1] Such simple models have been used in e.g. [16, 10, 9, 11, 49] for analytic studies of double descent [5, 38] in terms of classification error. Fig. 2(a) reveals a double descent for the balanced error.

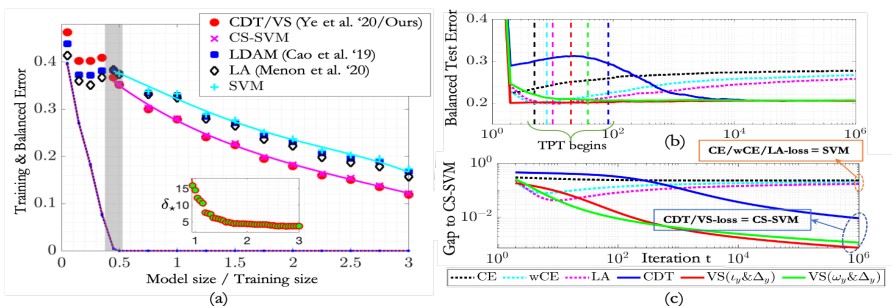

**Figure 2:** Insights on various cost-sensitive modifications of the CE-loss. **(a)** CDT has superior balanced-error performance over LA in the separable regime. Also, its performance matches that of CS-SVM, unlike LA matching SVM; Sec. 3.1 for more details. Solid lines follow theory of Sec. 4. **(b)** Although critical in TPT, multiplicative weights (aka CDT) can harm minority classes in initial phase of training by guiding the classifier in the wrong direction. Properly tuned additive weights (aka LA) can mitigate this effect and speed up convergence. This explains why VS can be superior compared to CDT (see Observation 1). Dashed lines show where TPT starts for each loss. **(c)** CDT and VS converge to CS-SVM, unlike LA and wCE. We prove this in Theorem 1.

training errors, which are zero for $\gamma \gtrsim 0.45$. The shaded region highlights the transition to the overparameterized / separable regime. In this regime, we continued training in the TPT. The plots reveal the following clear message: *The CDT-loss has better balanced-error performance compared to the LA-loss when both trained in TPT.* Moreover, they offer an intuitive explanation by uncovering a connection to max-margin classifiers: *In the TPT, (a) LA-loss performs the same as SVM, and, (b) CDT-loss performs the same as CS-SVM.*

We formalize those empirical observations in the theorem below, which holds for arbitrary linearly separable datasets (beyond Gaussian mixtures of the experiment). Specifically, for a sequence of norm-constrained minimizations of the VS-loss, we show that: As the norm constraint $R$ increases (thus, the problem approaches the original unconstrained loss), the direction of the constrained minimizer $\mathbf{w}_R$ converges to that of the CS-SVM solution $\hat{\mathbf{w}}_{\Delta_-/\Delta_+}$.

**Theorem 1** (VS-loss=CS-SVM). *Fix a binary training set $\{\mathbf{x}_i, y_i\}_{i=1}^n$ with at least one example from each of the two classes. Assume feature map $h(\cdot)$ such that the data are linearly separable, that is $\exists \mathbf{w} : y_i \mathbf{w}^T h(\mathbf{x}_i) \geq 1, \forall i \in [n]$. Consider training a linear model $f_\mathbf{w}(\mathbf{x}) = \langle \mathbf{w}, h(\mathbf{x}) \rangle$ by minimizing the VS-loss $\mathcal{L}_n(\mathbf{w}) := \sum_{i \in [n]} \ell_{\mathrm{VS}}(y_i, f_\mathbf{w}(\mathbf{x}_i))$ with $\ell_{\mathrm{VS}}$ defined in (1) for positive parameters $\Delta_\pm, \omega_\pm \geq 0$ and arbitrary $\iota_\pm$. Define the norm-constrained optimal classifier $\mathbf{w}_R = \arg\min_{\|\mathbf{w}\|_2 \leq R} \mathcal{L}_n(\mathbf{w})$. Let $\hat{\mathbf{w}}_\delta$ be the CS-SVM solution of (4) with $\delta = \Delta_-/\Delta_+$. Then, $\lim_{R \to \infty} \mathbf{w}_R/\|\mathbf{w}_R\|_2 = \hat{\mathbf{w}}_\delta/\|\hat{\mathbf{w}}_\delta\|_2$.*

On the one hand, the theorem makes clear that $\omega_\pm$ and $\iota_\pm$ become ineffective in the TPT as they all result in the same SVM solutions. On the other hand, the multiplicative parameters $\Delta_\pm$ lead to the same classifier as that of CS-SVM, thus favoring solutions that move the classifier towards the majority class provided that $\Delta_- > \Delta_+ \Leftrightarrow \delta > 1$. The proof is given in the SM together with extensions for multiclass datasets. In the SM, we also strengthen Theorem 1 by characterizing the *implicit bias* of gradient-flow on VS-loss. Finally, we show that group-sensitive VS-loss with $\Delta_{y,g} = \Delta_g$ converges to the corresponding GS-SVM.

**Remark 1.** *Thm 1 is reminiscent of Thm. 2.1 in [44] who showed for a regularized ERM with CE-loss that when the regularization parameter vanishes, the normalized solution converges to the SVM classifier. Our result connects nicely to [44] extending their theory to VS-loss / CS-SVM, as well as, to the group-case. In a similar way, our result on the implicit bias of gradient-flow on the VS-loss connects to more recent works [47, 20] that pioneered corresponding results for CE-loss. Although related, our results on the properties of the VS-loss are not obtained as special cases of these existing works. We remark that, when combined with a recent result by [19], our Theorem 1 also implies that gradient descent on the VS-loss with sufficiently small step size converges in direction to the solution of the CS-SVM. In other words, Theorem 1 characterizes the implicit bias of gradient descent on the VS-loss. As a final note, in Fig. 2(b,c) we kept constant learning rate 0.1. Significantly faster*

*convergence is observed with normalized GD schemes [36, 21]; see the SM for a detailed numerical study. We also note that Thm. 1 gives a modern interpretation to the CS-SVM via the lens of implicit bias theory.*

## 3.2 VS-loss: Best of two worlds

We have shown that multiplicative weights $\Delta_\pm$ are responsible for good balanced accuracy in the TPT. Here, we show that, at the initial phase of training, the same multiplicative weights can actually harm the minority classes. The following observation supports this claim.

**Observation 1.** *Assume $f_{\mathbf{w}}(x) = 0$ at initialization. Then, the gradients of CDT-loss with multiplicative logit factors $\Delta_y$ are identical to the gradients of wCE-loss with weights $\omega_y = \Delta_y$. Thus, we conclude the following where say $y = +1$ is minority. On the one hand, wCE, which typically sets $\omega_+ > \omega_-$ (e.g., $\omega_y = 1/\pi_y$), helps minority examples by weighing down the loss over majority. On the other hand, the CDT-loss requires the reverse direction $\Delta_+ < \Delta_-$ as per Theorem 1, thus initially it guides the classifier in the wrong direction to penalize minorities.*

To see why the above is true note that for $f_{\mathbf{w}}(x) = \langle \mathbf{w}, h(\mathbf{x}) \rangle$ the gradient of VS-loss is $\nabla_{\mathbf{w}} \ell_{\mathrm{VS}}(y, f_{\mathbf{w}}(\mathbf{x})) = -\omega_y \Delta_y \, \sigma\big( -\Delta_y y f_{\mathbf{w}}(x) + \iota_y \big) \cdot y h(\mathbf{x})$ where $\sigma(t) = (1 + \exp(-t))^{-1}$ is the sigmoid function. It is then clear that at $f_{\mathbf{w}}(\mathbf{x}) = 0$, the logit factor $\Delta_y$ plays the same role as the weight $\omega_y$. From Theorem 1, we know that pushing the margin towards majorities (which favors balancing the conditional errors) requires $\Delta_+ < \Delta_-$. Thus, gradient of minorities becomes smaller, initially pushing the optimization in the wrong direction. Now, we turn our focus at the impact of $\iota_y$'s at the start of training. Noting that $\sigma(\cdot)$ is increasing function, we see that setting $\iota_+ > \iota_-$ increases the gradient norm for minorities. This leads us to a second observation: *By properly tuning the additive logit adjustments $\iota_y$ we can counter the initial negative effect of the multiplicative adjustment, thus speeding up training.* The observations above naturally motivated us to formulate the VS-loss in Eqn. (2) bringing together the best of two worlds: the $\Delta_y$'s that play a critical role in the TPT and the $\iota_y$'s that compensate for the harmful effect of the $\Delta_y$'s in the beginning of training.

Figure 2(b,c) illustrate the discussion above. In the binary linear classification setting of Fig. 2(a), we investigate the effect of the additive adjustments on the training dynamics. Specifically, we trained using gradient descent: (i) *CE*; (ii) *wCE* with $\omega_y = 1/\pi_y$; (iii) *LA-loss* with $\iota_y = \log(1/\pi_y)$; (iv) *CDT-loss* with $\Delta_+ = \delta_\star^{-1}, \Delta_- = 1$; (v) *VS-loss* with $\Delta_+ = \delta_\star^{-1}, \Delta_- = 1$, $\iota_y = \log(1/\pi_y)$ and $\omega_y = 1$; (vi) *VS-loss* with same $\Delta$'s, $\iota_y = 0$ and $\omega_y = 1/\pi_y$. Figures 2(b) and (c) plot balanced test error $\mathcal{R}_{\mathrm{bal}}$ and angle-gap to CS-SVM solution as a function of iteration number for each algorithm. The vertical dashed lines mark the iteration after which training error stays zero and we enter the TPT. Observe in Fig. 2(c) that CDT/VS-losses, both converge to the CS-SVM solution as TPT progresses verifying Theorem 1. This also results in lowest test error in the TPT in Fig. 2(b). However, compared to CDT-loss, the VS-loss enters faster in the TPT and converges orders of magnitude faster to small values of $\mathcal{R}_{\mathrm{bal}}$. Note in Fig. 2(c) that this behavior is correlated with the speed at which the two losses converge to CS-SVM. Following the discussion above, we attribute this favorable behavior during the initial phase of training to the inclusion of the $\iota_y$'s. This is also supported by Fig. 2(c) as we see that LA-loss (but also wCE) achieves significantly better values of $\mathcal{R}_{\mathrm{bal}}$ at the first stage of training compared to CDT-loss. In Sec. 5.1 we provide deep-net experiments on an imbalanced CIFAR-10 dataset that further support these findings.

## 4  Generalization analysis and fairness tradeoffs

Our results in the previous section regarding VS-loss/CS-SVM hold for arbitrary linearly-separable training datasets. Here, under additional distributional assumptions, we establish a sharp asymptotic theory for VS-loss/CS-SVM and their group-sensitive counterparts.

**Data model.** We study binary Gaussian-mixture generative models (GMM) for the data distribution $\mathcal{D}$. For the label $y \in \{\pm 1\}$, let $\pi := \mathbb{P}\{y = +1\}$. Group membership is decided conditionally on the label such that $\forall j \in [K] : \mathbb{P}\{g = j | y = \pm 1\} = p_{\pm,j}$, with $\sum_{j \in [K]} p_{+,j} = \sum_{j \in [K]} p_{-,j} = 1$. Finally, the feature conditional given label $y$ and group $g$ is a multivariate Gaussian of mean $\boldsymbol{\mu}_{y,g} \in \mathbb{R}^d$ and covariance $\boldsymbol{\Sigma}$, i.e. $\mathbf{x} | (y, g) \sim \mathcal{N}(\boldsymbol{\mu}_{y,g}, \boldsymbol{\Sigma})$. Specifically for *label-imbalances*, we let $K = 1$ and $\mathbf{x} | y \sim \mathcal{N}(\boldsymbol{\mu}_y, \mathbf{I}_d)$ (see SM for $\boldsymbol{\Sigma} \neq \mathbf{I}_d$). For *group-imbalances*, we focus on two groups with $p_{+,1} = p_{-,1} = p < 1 - p = p_{+,2} = p_{-,2}$, $j = 1, 2$ and

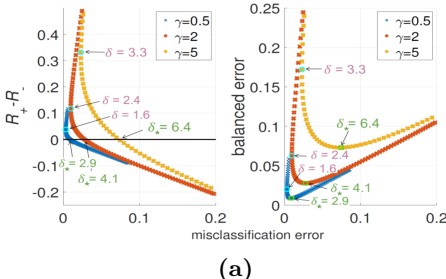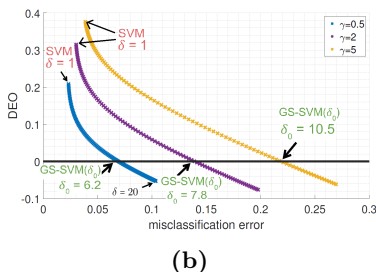

**(a)**                      **(b)**

**Figure 3:** Fairness tradeoffs between classification error and error-imbalance/balanced-error/DEO on GMM data achieved by **(a)** CS-SVM for class prior $\pi = 0.05$ and **(b)** GS-SVM for group prior $p = 0.05$, as a function of the margin-ratio hyperparameter $\delta \geq 1$ and for various values of overparameterization $\gamma$. Plots in (a) are generated using our sharp predictions in Theorem 2. Plots in (b) use corresponding result for GS-SVM given in the SM. See text for interpretations.

$\mathbf{x} \mid (y, g) \sim \mathcal{N}(y\boldsymbol{\mu}_g, \mathbf{I}_d)$. In both cases, $\mathbf{M}$ denotes the matrix of means, i.e. $\mathbf{M} = [\boldsymbol{\mu}_+ \quad \boldsymbol{\mu}_-]$ and $\mathbf{M} = [\boldsymbol{\mu}_1 \quad \boldsymbol{\mu}_2]$, respectively. Also, consider the eigen-decomposition: $\mathbf{M}^T\mathbf{M} = \mathbf{V}\mathbf{S}^2\mathbf{V}^T$, $\mathbf{S} > \mathbf{0}_{r \times r}, \mathbf{V} \in \mathbb{R}^{2 \times r}, r \in \{1, 2\}$, with $\mathbf{S}$ an $r \times r$ diagonal positive-definite matrix and $\mathbf{V}$ an orthonormal matrix obeying $\mathbf{V}^T\mathbf{V} = \mathbf{I}_r$. We study linear classifiers with $h(\mathbf{x}) = \mathbf{x}$.

**Learning regime.** We focus on the separable regime. For the models above, linear separability undergoes a sharp phase-transition as $d, n \to \infty$ at a proportional rate $\gamma = \frac{d}{n}$. That is, there exists threshold $\gamma_\star := \gamma_\star(\mathbf{V}, \mathbf{S}, \pi) \leq 1/2$ for the label-case, such that data are linearly separable with probability approaching one provided that $\gamma > \gamma_\star$ (accordingly for the group-case) [7, 34, 10, 22, 26]. See SM for formal statements and explicit definitions.

**Analysis of CS/GS-SVM.** We use $\overset{P}{\longrightarrow}$ to denote convergence in probability and $Q(\cdot)$ the standard normal tail. We let $(x)_- := \min\{x, 0\}$; $\mathbb{1}[\mathcal{E}]$ the indicator function of event $\mathcal{E}$; $\mathcal{B}_2^r$ the unit ball in $\mathbb{R}^r$; and, $\mathbf{e}_1 = [1, 0]^T, \mathbf{e}_2 = [0, 1]^T$ standard basis vectors in $\mathbb{R}^2$. We further need the following definitions. Let random variables as follows: $G \sim \mathcal{N}(0, 1)$, $Y$ symmetric Bernoulli with $\mathbb{P}\{Y = +1\} = \pi$, $E_Y = \mathbf{e}_1\mathbb{1}[Y = 1] - \mathbf{e}_2\mathbb{1}[Y = -1]$ and $\Delta_Y = \delta \cdot \mathbb{1}[Y = +1] + \mathbb{1}[Y = -1]$, for $\delta > 0$. With these define key function $\eta_\delta : \mathbb{R}_{\geq 0} \times \mathcal{B}_2^r \times \mathbb{R} \to \mathbb{R}$ as $\eta_\delta(q, \boldsymbol{\rho}, b) := \mathbb{E}\left[\left(G + E_Y^T\mathbf{V}\mathbf{S}\boldsymbol{\rho} + \frac{bY - \Delta_Y}{q}\right)_-^2\right] - (1 - \|\boldsymbol{\rho}\|_2^2)\gamma$. Finally, define $(q_\delta, \boldsymbol{\rho}_\delta, b_\delta)$ as the *unique* triplet (see SM for proof) satisfying $\eta_\delta(q_\delta, \boldsymbol{\rho}_\delta, b_\delta) = 0$ and $(\boldsymbol{\rho}_\delta, b_\delta) := \arg\min_{\|\boldsymbol{\rho}\|_2 \leq 1, b \in \mathbb{R}} \eta_\delta(q_\delta, \rho, b)$. Note that these triplets can be easily computed numerically for given values of $\gamma, \delta, \pi, p$ and means' Gramian $\mathbf{M}^T\mathbf{M} = \mathbf{V}\mathbf{S}^2\mathbf{V}^T$.

**Theorem 2** (Balanced error of CS-SVM)**.** *Let GMM data with label imbalances and learning regime as described above. Consider the CS-SVM classifier in (4) with $h(\mathbf{x}) = \mathbf{x}$, intercept $b$ (i.e. constraints $\langle \mathbf{x}, \mathbf{w} \rangle + b \geq \{\delta \text{ or } 1\}$ in (4)) and fixed margin-ratio $\delta > 0$. Define $\overline{\mathcal{R}}_+ := Q\left(\mathbf{e}_1^T\mathbf{V}\mathbf{S}\boldsymbol{\rho}_\delta + b_\delta/q_\delta\right)$ and $\overline{\mathcal{R}}_- := Q\left(-\mathbf{e}_2^T\mathbf{V}\mathbf{S}\boldsymbol{\rho}_\delta - b_\delta/q_\delta\right)$. Then, as $n, d \to \infty$ with $d/n = \gamma > \gamma_\star$, it holds that $\mathcal{R}_+ \overset{P}{\longrightarrow} \overline{\mathcal{R}}_+$ and $\mathcal{R}_- \overset{P}{\longrightarrow} \overline{\mathcal{R}}_-$. In particular, $\mathcal{R}_{bal} \overset{P}{\longrightarrow} \overline{\mathcal{R}}_{bal} := (\overline{\mathcal{R}}_+ + \overline{\mathcal{R}}_-)/2$.*

The theorem further shows $(\|\hat{\mathbf{w}}_\delta\|_2, \frac{\hat{\mathbf{w}}_\delta^T\boldsymbol{\mu}_+}{\|\hat{\mathbf{w}}_\delta\|_2}, \frac{\hat{\mathbf{w}}_\delta^T\boldsymbol{\mu}_-}{\|\hat{\mathbf{w}}_\delta\|_2}, \hat{b}_\delta) \overset{P}{\longrightarrow} (q_\delta, \mathbf{e}_1^T\mathbf{V}\mathbf{S}\boldsymbol{\rho}_\delta, \mathbf{e}_2^T\mathbf{V}\mathbf{S}\boldsymbol{\rho}_\delta, b_\delta)$. Thus, $b_\delta$ is the asymptotic the intercept, $q_\delta^{-1}$ is the asymptotic classifier's margin $1/\|\hat{\mathbf{w}}_\delta\|_2$ to the majority, and $\boldsymbol{\rho}_\delta$ determines the asymptotic alignment of the classifier with the class mean. The proof uses the convex Gaussian min-max theorem (CGMT) framework [48, 51]; see SM for background, the proof, as well as, (a) simpler expressions when the means are antipodal ($\pm\boldsymbol{\mu}$) and (b) extensions to general covariance model ($\boldsymbol{\Sigma} \neq \mathbf{I}$). The experiment (solid lines) in Figure 2(a) validates the theorem's predictions. Also, in the SM, we characterize the DEO of GS-SVM for GMM data. Although similar in nature, that characterization differs to Thm. 2 since each class is now itself a Gaussian mixture as described in the model above.

**Fairness tradeoffs.** The theory above allow us to study tradeoffs between misclassification / balanced error / DEO in Fig. 3. Fig. 3(a) focuses on label imbalances. We make the following observations. (1) The optimal value $\delta_\star$ minimizing $\mathcal{R}_{bal}$ also achieves perfect

balancing between the conditional errors of the two classes, that is $\mathcal{R}_+ = \mathcal{R}_- = Q(\frac{\ell_- + \ell_+}{2})$. We prove this interesting property in the SM by deriving an explicit formula for $\delta_\star$ that only requires computing the triplet $(q_1, \boldsymbol{\rho}_1, b_1)$ for $\delta = 1$ corresponding to the standard SVM. Such closed-form formula is rather unexpected given the seemingly involved nonlinear dependency of $\mathcal{R}_{\text{bal}}$ on $\delta$ in Thm. 2. In the SM, we also use this formula to formulate a theory-inspired heuristic for hyperparameter tuning, which shows good empirical performance on simple datasets such as imbalanced MNIST. (2) The value of $\delta$ minimizing standard error $\mathcal{R}$ (shown in magenta) is not equal to 1, hence CS-SVM also improves $\mathcal{R}$ (not only $\mathcal{R}_{\text{bal}}$). In Fig. 3(b), we investigate the effect of $\delta$ and the improvement of GS-SVM over SVM. The largest DEO and smallest misclassification error are achieved by the SVM ($\delta = 1$). But, with increasing $\delta$, misclassification error is traded-off for reduction in absolute value of DEO. Interestingly, for some $\delta_0 = \delta_0(\gamma)$ (with value increasing with $\gamma$) GS-SVM guarantees Equal Opportunity (EO) $\mathcal{R}_{\text{deo}} = 0$ (without explicitly imposing such constraints as in [39, 12]).

## 5 Experiments

We show experimental results further justifying theoretical findings. (Code available in [1]).

### 5.1 Label-imbalanced data

Our first experiment (Table 1) shows that *non-trivial combinations* of additive/multiplicative adjustments can improve balanced accuracy over *individual* ones. Our second experiment (Fig. 4) validates the theory of Sec. 3 by examining how these adjustments affect training.

**Datasets.** Table 1 evaluates LA/CDT/VS-losses on imbalanced instances of CIFAR-10/100. Following [8], we consider: (1) *STEP* imbalance, reducing the sample size of half of the classes to a fixed number. (2) Long-tailed (*LT*) imbalance, which exponentially decreases the number of training images across different classes. We set an imbalance ratio $N_{\max}/N_{\min} = 100$, where $N_{\max} = \max_y N_y, N_{\min} = \min_y N_y$ and $N_y$ are sample sizes of class $y$. For consistency with [17, 8, 32, 54] we keep a balanced test set and in addition to evaluating our models on it, we treat it as our validation set and use it to tune our hyperparameters. More sophisticated tuning strategies (perhaps using bi-level optimization) are deferred to future work. We use data-augmentation exactly as in [17, 8, 32, 54]. See SM for more implementation details.

**Model and Baselines.** We compare the following: *(1) CE-loss. (2) Re-Sampling* that includes each data point in the batch with probability $\pi_y^{-1}$. *(3) wCE* with weights $\omega_y = \pi_y^{-1}$. *(4) LDAM-loss* [8], special case of LA-loss where $\iota_y = \frac{1}{2}(N_{\min}/N_y)^{1/4}$ is *subtracted* from the logits. *(5) LDAM-DRW* [8], combining LDAM with deferred re-weighting. *(6) LA-loss* [32], with the Fisher-consistent parametric choice $\iota_y = \tau \log(\pi_y)$. *(7) CDT-loss* [54], with $\Delta_y = (N_y/N_{max})^\gamma$. *(8) VS-loss*, with combined hyperparameters $\iota_y = \tau \log(\pi_y)$ and $\Delta_y = (N_y/N_{\max})^\gamma$, pa-

**Table 1:** Top-1 accuracy results on balanced validation set (%).

| Dataset | CIFAR 10 | | CIFAR 100 | |
|---|---|---|---|---|
| Imbalance Profile | LT-100 | STEP-100 | LT-100 | STEP-100 |
| CE | $71.94 \pm 0.38$ | $62.69 \pm 0.50$ | $38.82 \pm 0.69$ | $39.49 \pm 0.16$ |
| Re-Sampling | 71.2 | 65.0 | 34.7 | 38.4 |
| wCE | 72.6 | 67.3 | 40.5 | 40.1 |
| LDAM [8]. | 73.35 | 66.58 | 39.60 | 39.58 |
| LDAM-DRW [8] | 77.03 | 76.92 | 42.04 | 45.36 |
| LA ($\tau = \tau^*$) [32] | $80.81 \pm 0.30$ | $78.23 \pm 0.52$ | $42.87 \pm 0.32$ | $45.69 \pm 0.27$ |
| CDT ($\gamma = \gamma^*$) [54] | $79.55 \pm 0.35$ | $73.26 \pm 0.29$ | $42.57 \pm 0.32$ | $44.12 \pm 0.17$ |
| VS ($\tau = \tau^*, \gamma = \gamma^*$) | $\mathbf{80.82} \pm 0.37$ | $\mathbf{79.10} \pm 0.66$ | $\mathbf{43.52} \pm 0.46$ | $\mathbf{46.53} \pm 0.17$ |

rameterized by $\tau, \gamma > 0$ respectively [2]. The works introducing (5)-(7) above, all trained for a different number of epochs, with dissimilar regularization and learning rate schedules. For consistency, we follow the training setting in [8]. Thus, for LDAM we adapt results reported by [8], but for LA and CDT, we reproduce our own in that setting. Finally, for a fair comparison we ran LA-loss for optimized $\tau = \tau^*$ (rather than $\tau = 1$ in [32]).

**VS-loss balanced accuracy.** Table 1 shows Top-1 accuracy on balanced validation set (averaged over 5 runs). We use a grid to pick the best $\tau$ / $\gamma$ / $(\tau, \gamma)$-pair for the LA /

---

[2]Here, the hyperparameter $\gamma$ is used with some abuse of notation and is important to not be confused with the parameterization ratio in the linear models in Sec. 3 and 4. We have opted to use the same notation as in [54] to ease direct comparisons of experimental findings.

CDT / VS losses on the validation set. Since VS includes LA and CDT as special cases (corresponding to $\gamma = 0$ and $\tau = 0$ respectively), we expect that it is at least as good as the latter over our hyper-parameter grid search. We find that the optimal $(\tau^*, \gamma^*)$-pairs correspond to non-trivial combinations of each individual parameter. Thus, VS-loss has better balanced accucy as shown in the table. See SM for optimal hyperparameters choices.

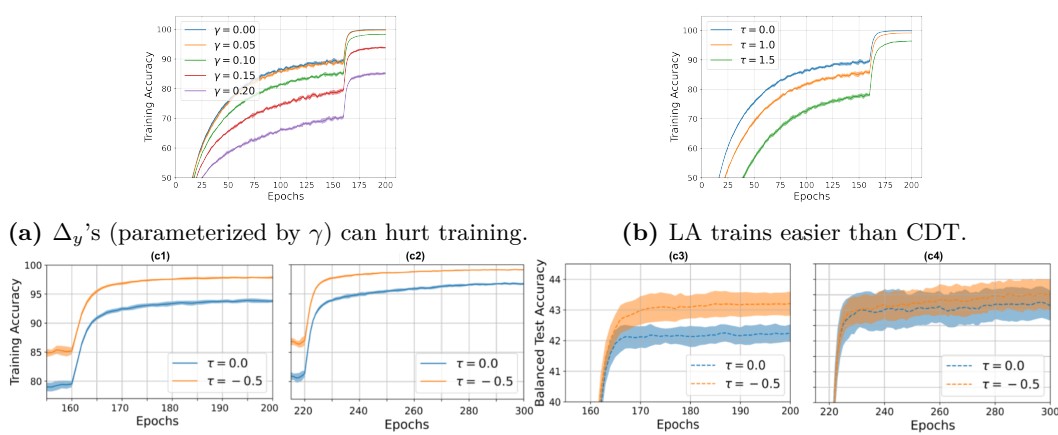

**(a)** $\Delta_y$'s (parameterized by $\gamma$) can hurt training. **(b)** LA trains easier than CDT.

**(c)** $\iota_y$'s mitigate the effect of $\Delta_y$'s (c1,c2), but $\Delta_y's$ dominate TPT performance (c3,c4).

**Figure 4:** Experiments on CIFAR10 with Long-tailed LT-100 imbalance demonstrating the effects of additive/multiplicative parameters at different phases of training. All results are averaged over 5 runs and shaded regions indicate the 95% confidence intervals. See text for details and interpretations.

**How hyperparameters affect training?** We perform three experiments. **(a)** Figure 4(a) shows that larger values of hyperparameter $\gamma$ (corresponding to more dispersed $\Delta_y$'s between classes) hurt training performance and delay entering to TPT. Complementary Figures 4(c1,c2) show that eventually, if we train longer, then, train accuracy approaches 100%. These findings are in line with Observation 1 in Sec. 3.2. **(b)** Figure 4(b) shows training accuracy of LA-loss for changing hyperparameter $\tau$ controlling additive adjustments. On the one hand, increasing values of $\tau$ delay training accuracy to reach 100%. On the other hand, when compared to the effect of $\Delta_y$'s in Fig. 4(a), we observe that the impact of additive adjustments on training is significantly milder than that of multiplicative adjustments. Thus, LA trains easier than CDT. **(c)** Figure 4(c) shows train and balanced accuracies for (i) CDT-loss in blue: $\tau = 0$, $\gamma = 0.15$, (ii) VS-loss in orange: $\tau = -0.5$, $\gamma = 0.15$. In Fig. 4(c1,c3) we trained for 200 epochs, while in Fig. 4(c2,c4) we trained for 300 epochs. For $\gamma = 0.15$, CDT-loss does *not* reach good training accuracy within 200 epochs ($\sim 93\%$ at epoch 200 in Fig. 4(c1)), but the addition of $\iota_y$'s with $\tau = -0.5$ mitigates this effect achieving improved $\sim 97\%$ accuracy at 200 epochs. This also translates to balanced test accuracy: VS-loss has better accuracy at the end of training in Fig. 4(c3). Yet, CDT-loss has not yet entered the interpolating regime in this case. So, we ask: What changes if we train longer so that both CDT and VS loss get (closer) to interpolation. In Fig. 4(c2) train accuracy of both algorithms increases when training continues to 300 epochs. Again, thanks to the $\iota_y$'s VS-loss trains faster. However, note in Figure 4(c4) that the balanced accuracies of the two methods are now very close to each other. Thus, in the interpolating regime what dominates the performance are the multiplicative adjustments which are same for both losses. This is in agreement with the finding of Theorem 1 and the synthetic experiment in Fig. 2(b,c).

## 5.2 Group-sensitive data

The message of our experiments on group-imbalanced datasets is three-fold. (1) We demonstrate the practical relevance of logit-adjusted CE modifications to settings with imbalances at the level of (sub)groups. (2) We show that such methods are competitive to alternative state-of-the-art; specifically, distributionally robust optimization (DRO) algorithms. (3) We propose combining logit-adjustments with DRO methods for even superior performance.

**Dataset.** We study a setting with spurious correlations —strong associations between label and background in image classification— which can be cast as a subgroup-sensitive classification problem [45]. We consider the Waterbirds dataset [45]. The goal is to classify images as either 'waterbirds' or 'landbirds', while their background —either 'water' or 'land'— can be spuriously correlated with the type of birds. Formally, each example has label $y \in \mathcal{Y} = \{\pm 1\} \equiv \{\text{waterbird}, \text{landbird}\}$ and belongs to a group $g \in \mathcal{G} = \{\pm 1\} \equiv \{\text{water}, \text{land}\}$. Let then $s = (y, g) \in \{\pm 1\} \times \{\pm 1\}$ be the four sub-groups with $(+1, -1)$, $(-1, +1)$ being minorities (specifically, $\hat{p}_{+1,+1} = 0.22, \hat{p}_{+1,-1} = 0.012, \hat{p}_{-1,+1} = 0.038$ and $\hat{p}_{-1,-1} = 0.73$.). Denote $N_s$ the number of training examples belonging to sub-group $s$ and $N_{\max} := \max_s N_s$. For notational consistency with Sec. 2, we note that the imbalance here is in subgroups; thus, Group-VS-loss in (3) consists of logit adjustments that depend on the subgroup $s = (y, g)$.

**Model and Baselines.** As in [45], we train a ResNet50 starting with pretrained weights on Imagenet. Let $\beta_{s=(y,g)} = (N_{(y,g)}/N_{\max})$. We propose training with the group-sensitive VS-loss in (3) with $\Delta_{y,g} = \Delta_s = \beta_s^{\gamma}$ and $\iota_s = -\beta_s^{-\gamma}$ with $\gamma = 0.3$. We compare against CE and the DRO method of [45]. We also implement a new training scheme that combines Group-VS+DRO. We show additional results for Group-LA/CDT (not previously used in group contexts). For fair comparison, we reran the baseline experiments with CE and report our reproduced numbers. Since class +1 has no special meaning here, we use Symm-DEO $= (|\mathcal{R}_{(+1,+1)} - \mathcal{R}_{(+1,-1)}| + |\mathcal{R}_{(-1,+1)} - \mathcal{R}_{(-1,-1)}|)/2$ and also report balanced and worst sub-group accuracies. We did not fine-tune $\gamma$ as the heuristic choice already shows the benefit of Group-VS-loss. We expect further improvements tuning over validation set.

**Results.** Table 2 reports test values obtained at last epoch (300 in total). Our Group-VS loss significantly improves performance (measured with all three fairness metrics) over CE, providing a cure for the poor CE performance under overparameterization reported in [46]. Group-CDT/VS have comparable performances, with or without DRO. Also, both outperform Group-LA that only uses additive adjustments. While these conclusions hold for the specific heuristic tuning of $\iota_y$'s, $\Delta_y$'s described above, they are in alignment with our Theorem 1. Interestingly, Group-VS improves by a small margin the worst

**Table 2:** Symmetric DEO, balanced and worst-case subgroup accuracies on Waterbirds dataset; averages over 10 runs, along with standard deviations.

| Loss | Symm. DEO | Bal. acc. | Worst acc. |
|---|---|---|---|
| CE | 25.3±0.66 | 84.9±0.29 | 68.1±2.2 |
| Group LA | 24.0±2.4 | 84.2±3.0 | 70.1±2.6 |
| Group CDT | 18.5±0.46 | 87.2±1.2 | 75.4±2.2 |
| Group VS | **18.1±0.65** | **88.1±0.38** | **76.7±2.3** |
| CE + DRO | 16.3±0.37 | 88.7±0.31 | 75.2±2.1 |
| Group LA + DRO | 16.3±0.82 | 88.7±0.40 | 74.3±2.5 |
| Group CDT + DRO | **11.7±0.15** | **90.3±0.2** | **79.9±1.5** |
| Group VS + DRO | 11.8±0.70 | 90.2±0.22 | 78.9±1.0 |

accuracy over CE+DRO, despite the latter being specifically designed to minimize that objective. Our proposed Group-VS + DRO outperforms the CE+DRO algorithm used in [45] when training continues in TPT. Finally, Symm. DEO appears correlated with balanced accuracy, in alignment with our discussion in Sec. 4 (see Fig. 3(a)).

## 6  Concluding remarks

We presented a theoretically-grounded study of recently introduced cost-sensitive CE modifications for imbalanced data. To optimize key fairness metrics, we formulated a new such modification subsuming previous techniques as special cases and provided theoretical justification, as well as, empirical evidence on its superior performance against existing methods. We suspect the VS-loss and our better understanding on the individual roles of different hyperparameters can benefit NLP and computer vision applications; we expect future work to undertake this opportunity with additional experiments. When it comes to group-sensitive learning, it is of interest to extend our theory to other fairness metrics of interest. Ideally, our precise asymptotic theory could help contrast different fairness definitions and assess their pros/cons. Our results are the first to theoretically justify the benefits/pitfalls of specific logit adjustments used in [24, 8, 32, 54]. The current theory is limited to settings with fixed features. While this assumption is prevailing in most related theoretical works [20, 37, 16, 4, 35], it is still far from deep-net practice where (last-layer) features are learnt jointly with the classifier. We expect recent theoretical developments on that front [43, 33, 30] to be relevant in our setting when combined with our ideas.

## Acknowledgments

This work is supported by the National Science Foundation under grant Numbers CCF-2009030, by HDR-193464, by a CRG8 award from KAUST and by an NSERC Discovery Grant. C. Thrampoulidis would also like to acknowledge his affiliation with University of California, Santa Barbara. S. Oymak is partially supported by the NSF award CNS-1932254 and by the NSF CAREER award CCF-2046816.

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
