# Supplementary Material for "Label-Imbalanced and Group-Sensitive Classification under Overparameterization"

**Ganesh Ramachandra Kini**
University of California, Santa Barbara
kini@ucsb.edu

**Orestis Paraskevas**
University of California, Santa Barbara
orestis@ucsb.edu

**Samet Oymak**
University of California, Riverside
oymak@ece.ucr.edu

**Christos Thrampoulidis**
University of British Columbia
cthrampo@ece.ubc.ca

## Abstract

The goal in label-imbalanced and group-sensitive classification is to optimize relevant metrics such as balanced error and equal opportunity. Classical methods, such as weighted cross-entropy, fail when training deep nets to the terminal phase of training (TPT), that is training beyond zero training error. This observation has motivated recent flurry of activity in developing heuristic alternatives following the intuitive mechanism of promoting larger margin for minorities. In contrast to previous heuristics, we follow a principled analysis explaining how different loss adjustments affect margins. First, we prove that for all linear classifiers trained in TPT, it is necessary to introduce *multiplicative*, rather than *additive*, logit adjustments so that the interclass margins change appropriately. To show this, we discover a connection of the multiplicative CE modification to the cost-sensitive support-vector machines. Perhaps counterintuitively, we also find that, at the start of training, the same multiplicative weights can actually harm the minority classes. Thus, while additive adjustments are ineffective in the TPT, we show that they can speed up convergence by countering the initial negative effect of the multiplicative weights. Motivated by these findings, we formulate the *vector-scaling (VS) loss*, that captures existing techniques as special cases. Moreover, we introduce a natural extension of the VS-loss to group-sensitive classification, thus treating the two common types of imbalances (label/group) in a unifying way. Importantly, our experiments on state-of-the-art datasets are fully consistent with our theoretical insights and confirm the superior performance of our algorithms. Finally, for imbalanced Gaussian-mixtures data, we perform a generalization analysis, revealing tradeoffs between balanced / standard error and equal opportunity.

## Organization of the supplementary material

The supplementary material (SM) is organized as follows.

1. In Section A we provide additional technical information on the label-imbalanced experiments of Sec. 5.1. We also show experiments of imbalanced MNIST dataset.

35th Conference on Neural Information Processing Systems (NeurIPS 2021).

2. In Section B we provide missing details and additional results on the group-imbalanced experiments of Sec. 5.2.

3. In Section C we present synthetic experiments on both label-imbalanced and group-sensitive datasets further supporting our theoretical findings in Sections 3 and 4.

4. In Section D we present and prove a more general version of Theorem 1 (specifically, see Theorem 3) on the connection of overparameterized VS-loss and to CS-SVM. We also discuss multiclass extensions (see Theorem 4) and implicit bias of gradient flow (see Theorem 5).

5. In Section E, we present theoretical results on optimal tuning of CS-SVM. First, we state and prove Lemma 2 which establishes a structural connection between the solution of CS-SVM to the solution of the standard SVM, allowing to view the former as a post-hoc adjustment to the latter. Then, we use this property together with the sharp characterizations of Theorem 2 to derive an explicit formula for the optimal margin ratio under Gaussian mixture data.

6. In Section F we prove Theorem 2 on generalization of CS-SVM. We also discuss related works on sharp high-dimensional asymptotics and provide necessary background on the convex Gaussian min-max theorem. Finally, we include formulas for the phase-transition threshold of CS-SVM.

7. Finally, in Section G we state and prove Theorem 7 characterizing the DEO of GS-SVM as mentioned in Section 4.

To ease readability and accessibility, we also opted to keep the main manuscript. The SM starts at page 17.

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

# A    Additional Experiments on Label-Imbalanced Datasets

In this section, we provide omitted information (due to space limits) on the results of Section 5.1, as well, as additional experiments.

## A.1    Deep-net experiments

Here we provide additional implementation details and a more extensive discussion on the results presented in Table 1 in Section 5.1 of the main text.

**Technical details:** Following [13], we train a ResNet-32 [25], using batch size 128 and SGD with momentum 0.9 and weight decay $2 \times 10^{-4}$. For the first 5 epochs we use a linear warm up schedule until baseline learning rate of 0.1. We train for a total of 200 epochs, while decaying our learning rate by 0.1 at epochs 160 and 180. For STEP-100 imbalance we trained for 300 rather than 200 epochs and adjusted the learning rate accordingly as we found this type of imbalance more difficult to learn. We remark that the values for LDAM (adapted from [13]) used learning rate decay 0.01 and last-layer feature/classifier normalization. We have found convergence difficult otherwise. For other losses, we do *not* use the above normalization of weights to isolate the impact of loss modifications.

**Implementation details.** A seed is used for each of the 5 runs and the weights of the network are initialized with the same values for all the losses that we train. We only show 95% confidence intervals for CE, LA, CDT and VS losses which we implemented. For the remaining algorithms (e.g., LDAM), we report averages over 5 realizations as given in [13]. For LA, CDT and VS losses, we have tuned the hyper-parameters $(\tau, \gamma)$ over the validation set as described in Section 5.1 (see Remark 2 and Table 3). More sophisticated tuning strategies over the validation set (e.g., based on bilevel optimization or Hyperband [42, 38]) and the corresponding performance assessment on test set are left to future work. Same as in [25, 13, 46, 78] before training we augment the data by padding the images to size $40 \times 40$, flipping them horizontally at random and then random cropping them to their original size. We use PyTorch [59] building on codes provided by [13, 78]. Training is performed on 2 NVIDIA RTX-3080 GPUs.

**Remark 2** (On the $(\tau, \gamma)$ parameterization of $\iota_y$'s & $\Delta_y$'s)**.** *As mentioned in Section 5.1, our deep-net experiments with VS-loss for label-imbalances, use the following parameterization for the additive and multiplicative logit factors in terms of two hyperparameters $\tau$ and $\gamma$:*

$$\iota_y = \tau \log(N_y/N_{\text{tot}}) \qquad and \qquad \Delta_y = (N_y/N_{\max})^{\gamma}, \qquad (5)$$

*where $N_y$ is the train-sample size of class $y$, $N_{\max} = \max_y N_y$ and $N_{\text{tot}} = \sum_y N_y$. This parameterizations follow [46] and [78], respectively. A convenient feature is that setting $\tau = 0$ recovers the CDT-loss, and setting $\gamma = 0$ recovers the LA-loss.*

**Results and discussion.** Table 1 shows that our VS-loss performs favorably over the other methods across all experiments. The margins of improvement depend on the dataset / imbalance-type. Also, observe that in most cases LA-loss performs better than CDT-loss. This is likely because the CDT loss enters the TPT slower for the shown amount of training. Interestingly, VS-loss, even though it resembles the CDT-loss in the fact that it also adjusts the logits multiplicatively, does *not* seem to suffer from the same problem. In Section 3.1, we presented experiments showing that: (i) If given enough time to train, CDT-loss can achieve similar or better results than LA-loss. (ii) The addition of the $\iota_y$'s in the VS-loss can mitigate the effect of $\Delta_y$ on the speed of convergence. In that sense, VS-loss fulfills the theoretical intuition in Section 3.1, as the method that combines additive and multiplicative adjustments for high accuracy and fast convergence.

**Tuning results.** To promote reproducibility of our results and to give some insight on the range of $\tau$ and $\gamma$, in Table 3 we present the values of the hyperparameters that we determined through tuning and used to generate Table 1. As we discussed in Sec. 3.1, large values of $\tau$ and $\gamma$ can hinder training. Thus, when training with the VS loss, which adjusts the logits both in an additive and in a multiplicative way, it seems beneficial to use smaller values of these parameters, than when training with the LA or the CDT losses. Additionally, note that if searching over a grid, it is possible that the best values found for the VS-loss, will be

**Table 3:** Hyperparameter tuning results for each dataset, imbalance profile and loss function.

| Dataset | CIFAR 10 | | CIFAR 100 | |
|---|---|---|---|---|
| **Imbalance Profile** | **LT-100** | **STEP-100** | **LT-100** | **STEP-100** |
| LA $(\tau = \tau^*)$ [46] | 2.25 | 2.25 | 1.375 | 0.875 |
| CDT $(\gamma = \gamma^*)$ [78] | 0.4 | 0.3 | 0.1 | 0.1 |
| VS $(\tau = \tau^*, \gamma = \gamma^*)$ | $(1.25, 0.15)$ | $(1.5, 0.2)$ | $(0.75, 0.05)$ | $(0.5, 0.05)$ |

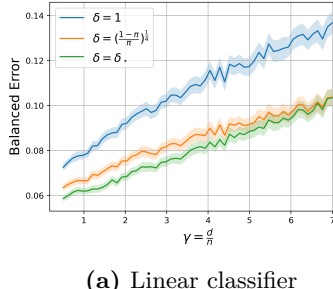

**(a)** Linear classifier

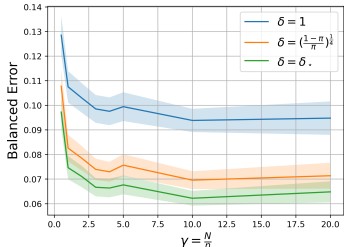

**(b)** Random-features classifier

**Figure 5:** A comparison of CS-SVM balanced error against the overparameterization ratio $\gamma$, for the standard hard margin SVM $(\delta = 1)$, for a heuristic $\delta = \left(\frac{1-\pi}{\pi}\right)^{\frac{1}{4}}$ and for our *approximation* of the optimal $\delta$ $(\delta = \tilde{\delta}_\star)$ obtained by the data-dependent heuristic in Section E.1.1. The experiment is performed on the MNIST dataset in a one-vs-rest classification task where the goal is to separate the minority class containing images of the digit 7 from the majority class containing images of all other digits. See text for details.

the same as those of the LA or CDT losses (but never worse than them). Searching over a fine enough grid though should yield parameter values for which VS-loss outperforms both of them. Finally, note that the $(\tau, \gamma)$-parameterization of the $\iota_y, \Delta_y$'s is itself restrictive and other alternatives might yield further improvements when combining both types of adjustments as observed in the other cases.

## A.2   Experiments on the MNIST dataset

Here, we present additional results on imbalanced MNIST data trained with linear and random-feature models. These results complement the synthetic experiment of Figure 2(a).

Specifically, we designed an experiment where we perform binary one-vs-rest classification on the MNIST dataset to classify digit 7 from the rest. Specifically, we split the dataset in two classes, the minority class containing images of the digit 7 and the majority class containing images of all other digits. To be consistent with our notation we assign the label $+1$ to the minority class and the label $-1$ to the majority class. Here, $d = 784$ and $\pi = 0.1$ is the prior for the minority class. All test-error evaluations were performed on a test set of 1000 samples. The results of the experiments were averaged over 200 realizations and the 90% confidence intervals for the mean are shown in Figure 5 as shaded regions.

We ran two experiments. In the first one depicted in Figure 5(a), we trained linear classifiers using the standard SVM (blue), the CS-SVM with a heuristic value $\delta = \left(\frac{1-\pi}{\pi}\right)^{\frac{1}{4}}$ (orange), and the CS-SVM with our heuristic data-dependent estimate of the optimal $\tilde{\delta}_\star$ (green). We compute such an estimate based on a recipe inspired by our exact expression in (33) for the GMM; see Section E.1.1 for details. We compute the three classifiers on training sets of varying sizes $n = d/\gamma$ for a range of values of $\gamma$ and report their balanced error. We observe that CS-SVM always outperforms SVM (aka $\delta = 1$) and the heuristic optimal tuning of CS-SVM consistently outperforms the choice $\delta = \left(\frac{1-\pi}{\pi}\right)^{\frac{1}{4}}$.

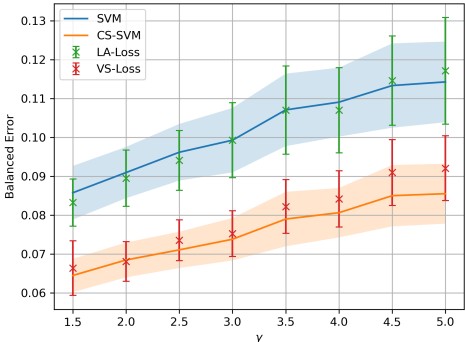

**Figure 6:** In the overparameterized regime, our VS loss converges to the CS-SVM classifier, while the LA-loss converges to the inferior —in terms of balanced-error performance— SVM. The experiment was performed on the MNIST dataset in a one-vs-rest classification task where the goal is to separate the minority class containing images of the digit 7 from the majority class containing images of all other digits. See text for details.

Next, in Figure 5(b) for the same dataset we trained a Random-features classifier. Specifically, for each one of the $n = 300$ training samples $\mathbf{x}_i \in \mathbb{R}^{d=784}$ we generate random features $\widetilde{\mathbf{x}}_i = \mathrm{ReLU}(\mathbf{A}\mathbf{x}_i)$ for a matrix $\mathbf{A} \in \mathbb{R}^{N \times d}$ which we sample once such that it has entries IID standard normal and is then standardized such that each column becomes unit norm. In this case we control $\gamma$ by varying the number $N = \gamma n$ of rows of that matrix $\mathbf{A}$. Observe here that the balanced error decreases as $\gamma$ increases (an instance of benign overfitting, e.g. [24, 7, 45] and that again the estimated optimal $\delta_\star$ results in tuning of CS-SVM that outperforms the other depicted choices.

In Figure 6 we repeat the experiment of Figure 5(a) only this time additionally to training CS-SVM for $\delta = 1$ and for $\delta = \tilde{\delta}_\star$ we also train using the LA-loss and our VS-loss. For the VS loss we use (1) with the following choice of parameters: $\omega_\pm = 1$, $\iota_\pm = 0$ and $\Delta_y = \tilde{\delta}_\star^{-1} \mathbb{1}[y = +1] + \mathbb{1}[y = -1]$ (see Section E.1.1 for $\tilde{\delta}_\star$). In a similar manner, LA-loss is defined using the same formula (1), but with parameters $\Delta_\pm = 1$, $\omega_\pm = 1$ and $\iota_+ = \pi^{-1/4}, \iota_- = (1 - \pi)^{-1/4}$ (as suggested in [13]).

The figure confirms our theoretical expectations: training with gradient descent on the LA and VS losses asymptotically (in the number of iterations) converge to the SVM and CS-SVM solutions respectively.

The training is performed over 200 epochs and for computing the gradient we iterate through the dataset in batches of size 64. The results are averaged over 200 realizations and the 90% confidence intervals are plotted as shaded regions for the CS-SVM model and as errorbars for the VS loss.

## B Further details and additional experiments on group-imbalances

### B.1 Deep-net experiments

In this section, we elaborate on our proposed method of combining our group logit-adjusted losses with the DRO method. In all experiments, we chose $\Delta_s = (N_s/N_{\max})^\gamma$, $\iota_s = -(N_s/N_{\max})^{-\gamma}$ with $\gamma = 0.3$. For example, Group-LA has $\iota_s = -(N_s/N_{\max})^{-0.3}$ and $\Delta_s = 0$.

**Group-VS+DRO algorithm.** For completeness, we elaborate on our proposed method of combining DRO with our Group VS-loss (see bottom half of Table 2). We recall from [62] that their proposed CE+DRO algorithm seeks a model that minimizes the worst subgroup empirical risk by instead minimizing the worst subgroup CE-loss: $\max_{s \in \mathcal{S}} \mathbb{E}_{(\mathbf{x},y) \sim \hat{P}_s}[\ell_{\mathrm{CE}}(y, f_{\mathbf{w}}(\mathbf{x}))]$,

where $\hat{P}_s$ is the empirical distribution on training samples from subgroup $s$. Instead, our

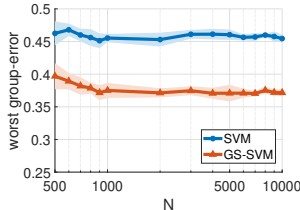
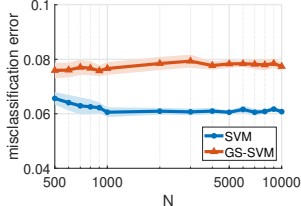

**(a)** Worst case sub-group error          **(b)** Misclassification error

**Figure 7:** The benefit of GS-SVM (corr. Group-VS loss) compared to SVM (corr. wCE) in achieving smaller *worst case sub-group error* without significant loss on the misclassification error in the Waterbirds dataset. Training a linear model with $N$-dimensional Random-feature map over pretrained ResNet-18 features as in [63].

Group-VS+DRO method attempts to solve the following distributionally robust optimization problem:

$$\min_{\mathbf{w}} \max_{s \in \mathcal{S}} \mathbb{E}_{(\mathbf{x},y) \sim \hat{P}_s}[\ell_{\text{Group–VS}}(y, s, f_{\mathbf{w}}(\mathbf{x}))],$$

with $\ell_{\text{Group–VS}}(y, s, f_{\mathbf{w}}(\mathbf{x})) = \omega_s \cdot \log\left(1 + e^{\iota_s} \cdot e^{-\Delta_s y f_{\mathbf{w}}(\mathbf{x})}\right)$ (see Equation (3)). To solve the above non-convex non-differentiable minimization, we employ the same online optimization algorithm given in [62, Algorithm 1], but changing the CE loss to the Group-VS.

### B.2 GS-SVM experiments

Section 5.2 demonstrated, for a deep-net model trained on the Waterbird dataset, the efficacy of the Group-VS loss compared to the CE and DRO algorithms used in [62]. Here, we follow [63] who, similar to us, focused in overparameterized training in the TPT. Specifically, [63] showed that wCE trained on a Random-feature model applied on top of a pretrained ResNet results in large *worst-group error* when trained in TPT. In their analysis, they observed that this is because weighted logistic loss in the separable regime behaves like SVM, which is insensitive to groups. Here, we repeat their experiment only this time we use the Group-VS loss. In line with our results thus far, Group-VS loss shows improved performance in this setting as well.

**Algorithm.** Concretely, since we are training linear models (on random feature maps), we know from Theorem 1 that Group-VS loss converges to GS-SVM. Thus, for simplicity, we directly trained the following instance of GS-SVM and compared it against SVM:

$$\min_{\mathbf{w}} \quad \|\mathbf{w}\|_2 \qquad \text{sub. to} \quad y_i(h(\mathbf{x}_i)^T \mathbf{w} + b) \geq \delta_{s_i}, \ i \in [n]. \tag{6}$$

Above, $\delta_{s_i} = \delta_{(y_i,g_i)} = \left(\frac{1}{\hat{p}_{(y,g)}}\right)^4$, $h : \mathcal{X} \to \mathbb{R}^N$ is the random-feature map (see Section A.2), and $\mathbf{x}_i, i \in [n]$ are $d$-dimensional pretrained ResNet18 features (same as those used in [63]). Here, $n = 4795$, $N$ took a range of values from 500 to 10000 and $d = 512$. For those values of $N$ the data are separable, thus SVM/GS-SVM are feasible.

**Experiment #1: GS-SVM vs SVM (or, Group-VS vs wCE).** Figure 7 shows worst-group and missclassification errors of GS-SVM and SVM as a function of the feature dimension $N$. The curves show averages over 10 realizations of the random projection matrix along with standard deviations depicted using shaded error-bars. We confirm that:

- GS-SVM consistently outperforms standard SVM in the overparameterized regime in terms of worst-group error
- This gain comes without significant losses on the misclassification error.

**Experiment #2: GS-SVM vs Sub-sampling.** As a means of improving over wCE, [63] proposed instead the use of *CE with subsampling,* for better *worst case sub-group error.* In Figure 8 we compare the performance of three algorithms: (i) SVM, (ii) GS-SVM, and (iii) SVM with subsampling (corresponding to CE with subsampling). For the latter, we chose

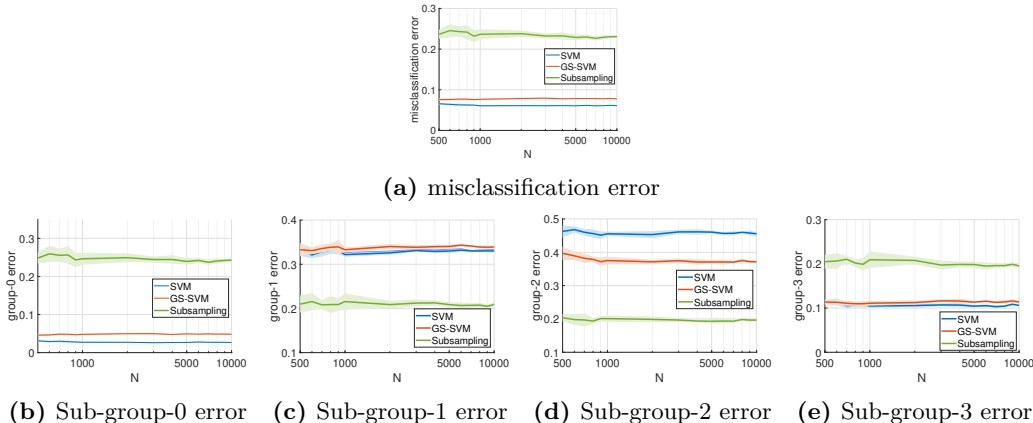

**(a)** misclassification error

**(b)** Sub-group-0 error  **(c)** Sub-group-1 error  **(d)** Sub-group-2 error  **(e)** Sub-group-3 error

**Figure 8:** Misclassification and conditional sub-group errors of SVM (blue), GS-SVM with heuristic tuning $\delta_{(y,g)} = p_{(y,g)}^{-4}$ (red), and, SVM with subsampling (green) for the Waterbirds dataset. GS-SVM has lower worst-case error (Sub-group-2) compared to the SVM without significant increase on the misclassification error. SVM with subsampling has the best worst-group error performance, but also worst misclassification error in subfigure (a).

56 examples from every sub-group (this is the size of the smallest sub-group) and ran SVM on the resulting (smaller), now balanced, dataset. Figure 8 reports missclassification error, as well as, *conditional sub-group errors.* Recall that in the original dataset, sub-groups- 0 and 3 were the *majority* with 3498 and 1057 examples respectively, while sub-groups 1 and 2 were the *minorities* with 184 and 56 examples, respectively. We find the following:

- Consistent with [63] SVM with subsampling achieves low *worst case sub-group error*, lower than both SVM and GS-SVM (at least, when tuned with $\delta_{(y,a)} = \left(\frac{1}{p_{(y,a)}}\right)^4$).

- Specifically, note the very low errors achieved by SVM with subsampling for minority sub-groups 2 and 3.

- However, the gain comes at a significant cost paid for the majority sub-groups- 1 and 3 resulting in an increase of the misclassification error by more than 3– times compared to standard SVM and GS-SVM.

We expect that, with more careful tuning of the hyper-parameters $\delta_{(y,g)}$, GS-SVM can eventually achieve even lower *sub-group errors* for the minority sub-groups without hurting the *majority sub-group errors* significantly. We leave this to future work.

## C  Additional numerical results

### C.1  Multiplicative vs Additive adjustments for label-imbalanced GMM

In Figure 9 we show a more complete version of Figure 2(a), where we additionally report standard and per-class accuracies. We minimized the CDT/LA losses in the separable regime with normalized gradient descent (GD), which uses increasing learning rate appropriately normalized by the loss-gradient norm for faster convergence; refer to Figure 14 and Section D.4 for the advantages over constant learning rate. Here, normalized GD was ran until the norm of the gradient of the loss becomes less than $10^{-8}$. We observed empirically that the GD on the LA-loss reaches the stopping criteria faster compared to the CDT-loss. This is in full agreement with the CIFAR-10 experiments in Section 3.1 and theoretical findings in Section 3.1.

In all cases, we reported both the results of Monte Carlo simulations, as well as, the theoretical formulas predicted by Theorem 2. As promised, the theorem sharply predicts the conditional error probabilities of minority/majority class despite the moderate dimension of $d = 300$.

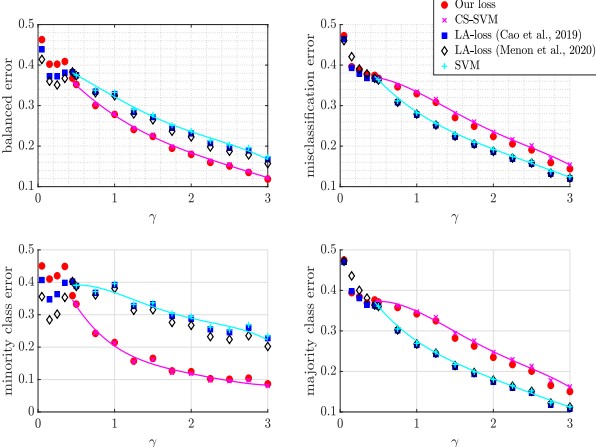

**Figure 9:** Performance of CDT vs LA loss for label-imbalanced GMM with missing features. (Top Left) is same as Figure 2(a). The other three plots show: (Top Right) misclassification error $\mathcal{R}$; (Bottom Left) majority class error $\mathcal{R}_-$; (Bottom Right) minority class error $\mathcal{R}_+$. Throughout, solid lines correspond to theoretical formulas obtained thanks to Theorem 2.

As noted in Section 3.1, CDT-loss results in better balanced error (see 'Top Left') in the separable regime (where $\mathcal{R}_{\text{train}} = 0$) compared to LA-loss. This naturally comes at a cost, as the role of the two losses is reversed in terms of the misclassification error (see 'Top Right'). The two bottom figures better explain these, by showing that VS sacrifices the error of majority class for a significant drop in the error of the minority class. All types of errors decrease with increasing overparameterization ratio $\gamma$ due to the mismatch feature model.

Finally, while balanced-error performance of CDT-loss is clearly better compared to the LA-loss in the separable regime, the additive offsets $\iota_y$'s improve performance in the non-separable regime. Specifically, the figure confirms experimentally the superiority of the tuning of the LA-loss in [46] compared to that in [13] (but only in the underparameterized regime). Also, it confirms our message: VS-loss that combines the best of two worlds by using both additive and multiplicative adjustments.

## C.2 Multiplicative vs Additive adjustments with $\ell_2$-regularized GD

In this section we shed more light on the experiments presented in Figure 2(b,c), by studying the effect of $\ell_2$-regularization. Specifically, we repeat here the experiment of Fig. 2(b) with $p = d = 50, n = 30$. We train with CE, CDT, and LA-losses in TPT with a weight-decay implementation of $\ell_2$-regularization, that is GD with update step: $\mathbf{w}_{t+1} = (1 - \beta)\mathbf{w}_t - \eta \nabla_{\mathbf{w}} \mathcal{L}(\mathbf{w}_t)$, where $\beta$ is the weight-decay factor and we used $\beta \in \{0, 10^{-3}, 10^{-2}\}$.

For our discussion, recall our findings in Section 3.1: (i) CDT-loss trained without regularization in TPT converges to CS-SVM, thus achieving better balanced error than LA-loss converging to SVM; (ii) however, at the beginning of training, multiplicative adjustment of CDT-loss can hurt the balanced error; (iii) Additive adjustments on the other hand helped in the beginning of GD iterations but were not useful deep in TPT.

We now turn our focus to the behavior of training in presence of $\ell_2$-regularization. The weight-decay factor was kept small enough to still achieve zero training error. A few interesting observations are summarized below:

- The classifier norm plateaus when trained with regularization (while it increases logarithmically without regularization; see Theorem 5). The larger the weight decay factor, the earlier the norm saturates; see Fig. 10(b) and (d).

- Suppose a classifier is trained with a small, but non-zero, weight decay factor in TPT, and the resulting classifier has a norm saturating at some value $\zeta > 0$. The final balanced error performance of such a classifier closely matches the balanced error

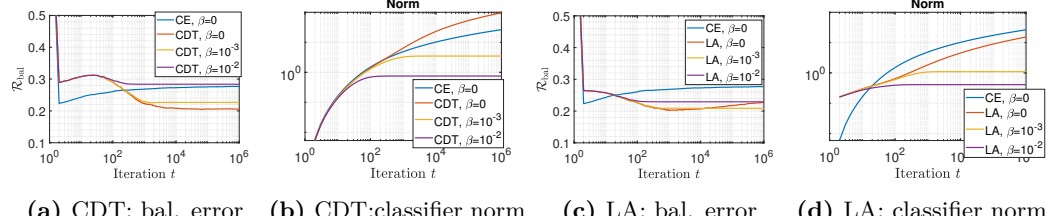

**(a)** CDT: bal. error    **(b)** CDT:classifier norm    **(c)** LA: bal. error    **(d)** LA: classifier norm

**Figure 10:** Training dynamics of a linear classifier trained with gradient descent on LA and CDT losses, with and without weight decay (parameter $\beta$).

produced by a classifier trained without regularization but with training stopped early at that iteration for which the classifier-norm is equal to $\zeta$; compare for example, the value of yellow curve (CDT, $\beta = 10^{-3}$) at $t = 10^6$ with the value of the red curve (CDT, $\beta = 0$) at around $t = 300$ in Fig. 10(c) and (d). [3]

- If early-stopped (appropriately) before entering TPT, LA-loss can give better balanced performance than CDT-loss. In view of the above mentioned mapping between weight-decay and training epoch, the use of weight decay results in same behavior. Overall, this supports that VS-loss, combining both additive and multiplicative adjustments is a better choice for a wide range of $\ell_2-$ regularization parameters.

### C.3    Additional information on Figures 2(b),(c) and 3(a),(b)

**Figures 2(b,c).** We generate data from a binary GMM with $d = 50, n = 30$ and $\pi = 0.1$. We generate mean vectors as random iid Gaussian vector and scale their norms to 5 and 1, respectively. For training, we use gradient descent with constant learning rate 0.1 and fixed number of $10^6$ iterations. The balanced test error in Figure 2(b) is computed by Monte Carlo on a balanced test set of $10^5$ samples. Figure 2(c) measures the angle gap of GD outputs $\mathbf{w}^t$ to the solution $\hat{\mathbf{w}}_\delta$ of CS-SVM in (4) with $\delta = \delta_\star$ and $\mathbf{h}(\mathbf{x}_i) = \mathbf{x}_i$.

**Figures 3(a,b).** In (a), we generated GMM data with $\|\boldsymbol{\mu}_+\| = 3, \boldsymbol{\mu}_- = -\boldsymbol{\mu}_+$ and $\pi = 0.05$. In (b), we considered the GMM of Section 4 with $\|\boldsymbol{\mu}_{y,g}\| = 3, y \in \{\pm 1\}, g \in \{1, 2\}$ and $\boldsymbol{\mu}_{+,1} \perp \boldsymbol{\mu}_{+,2} \in \mathbb{R}^d$, sensitive group prior $p = 0.05$ and equal class priors $\pi = 1/2$.

### C.4    VS-loss vs LA-loss for a group-sensitive GMM

In Figure 11 we test the performance of our theory-inspired VS-loss against the logit-adjusted (LA)-loss in a group-sensitive classification setting with data from a Gaussian mixture model with a minority and and a majority group. Specifically, we generated synthetic data from the model with class prior $\pi = 1 - \pi = 1/2$, minority group membership prior $p = 0.05$ (for group $g = 1$) and $\boldsymbol{\mu}_1 = 3\mathbf{e}_1, \boldsymbol{\mu}_2 = 3\mathbf{e}_2 \in \mathbb{R}^{500}$. We trained homogeneous linear classifiers based on a varying number of training sample $n = d/\gamma$. For each value of $n$ (eqv. $\gamma$) we ran normalized gradient descent (see Sec. D.4) on

- CDT-loss $\ell(y, \mathbf{w}^T\mathbf{x}, g) := \log(1 + e^{-\Delta_g y(\mathbf{w}^T\mathbf{x})})$ with $\Delta_g = \delta_0 \mathbb{1}[g = 1] + \mathbb{1}[g = 2]$.

- the LA-loss modified for group-sensitive classification $\ell(y, \mathbf{w}^T\mathbf{x}, g) := \log(1 + e^{\iota_g} e^{y(\mathbf{w}^T\mathbf{x})})$ with $\iota_g = p^{-1/4}\mathbb{1}[g = 1] + (1-p)^{-1/4}\mathbb{1}[g = 2]$. This value for $\iota$ is inspired by [13], but that paper only considered applying the LA-loss in label-imbalanced settings.

For $\gamma > 0.5$ where data are necessarily separable, we also ran the standard SVM and the GS-SVM with $\delta = \delta_0$.

Here, we chose the parameter $\delta_0$ such that the GS-SVM achieves zero DEO. To do this, we used the theoretical predictions of Theorem 7 for the DEO of GS-SVM for any value of $\delta$

---

[3]See also [60, 2] for the connection between gradient-descent and regularization solution paths.

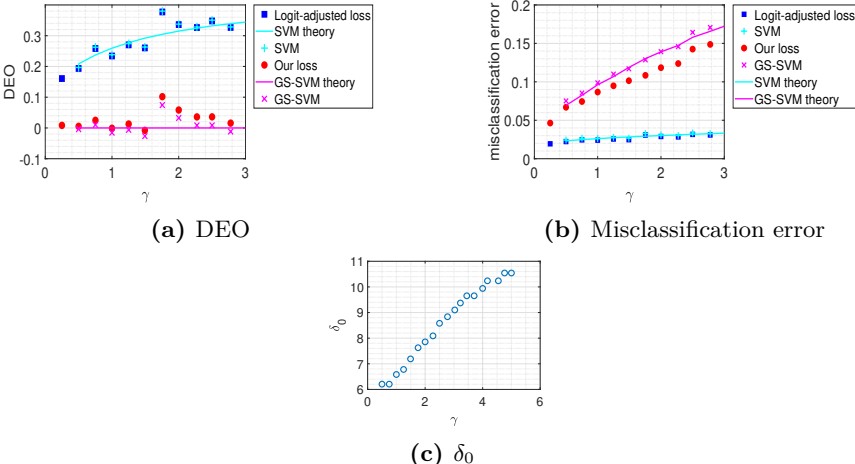

**(a)** DEO                 **(b)** Misclassification error

**(c)** $\delta_0$

**Figure 11:** This figure highlights the benefits of our theory-inspired VS-loss and GS-SVM over regular SVM and logit-adjusted loss in a group-sensitive classification setting. We trained a linear model with varying number $n$ of examples in $\mathbb{R}^{d=100}$, of a binary Gaussian-mixture dataset with two groups. $x$-axis is the parameterization ratio $d/n$. Data were generated from a GMM with prior $p = 0.05$ for the minority group. For $\gamma > 0.5$, we train additionally using SVM (cyan plus marker) and group-sensitive SVM (magenta cross). The plot (c) displays the parameter $\delta = \delta_0$ that we used to tune the VS-loss and GS-SVM. These values were obtained through a grid search from the theoretical prediction such that the theoretical $\mathcal{R}_{\text{deo}}$ (cf. Theorem 7) produced by the corresponding GS-SVM is 0. The solid lines depict theoretical predictions obtained by Theorem 7. The empirical probabilities were computed by averaging over 25 independent realizations of the training and test data.

and performed a grid-search giving us the desired $\delta_0$; see Figure 11 for the values of $\delta_0$ for different values of $\gamma$.

Figure 11(a) verifies that the GS-SVM achieves DEO (very close to) zero on the generated data despite the finite dimensions in the simulations. On the other hand, SVM has worse DEO performance. In fact, the DEO of SVM increases with $\gamma$, while that of GS-SVM stays zero by appropriately tuning $\delta_0$.

The figure further confirms the message of Theorem 3: In the separable regime, GD on logit-adjusted loss converges to the standard SVM performance, whereas GD on our VS-loss converges to the corresponding GS-SVM solution, thus allowing to tune a suitable $\delta$ that can trade-off misclassification error to smaller DEO magnitudes. The stopping criterion of GD was a tolerance value on the norm of the gradient. The match between empirical values and the theoretical predictions improves with increase in the dimension, more Monte-Carlo averaging and a stricter stopping criterion for GD.

## C.5    Validity of theoretical performance analysis

Figures 12 and 13 demonstrate that our Theorems 2 and 7 provide remarkably precise prediction of the GMM performance even when dimensions are in the order of hundreds. Moreover, both figures show the clear advantage of CS/GS-SVM over regular SVM and naive resampling strategies in terms of balanced error and equal opportunity, respectively.

The reported values for the misclassification error and the balanced error / DEO were computed over $10^5$ test samples drawn from the same distribution as the training examples.

Additionally, Figure 12 validates the explicit formula that we derive in Equation (33) for $\delta_\star$ minimizing the balanced error. Specifically, observe that CS-SVM with $\delta = \delta_\star$ ('×' markers) not only minimizes balanced error (as predicted in Section E.3), but also leads to better misclassification error compared to SVM for all depicted values of $\gamma$. The figure also shows the performance of our data-dependent heuristic of computing $\delta_\star$ introduced in Section E.1.1. The heuristic appears to be accurate for small values of $\gamma$ and is still better in terms of balanced

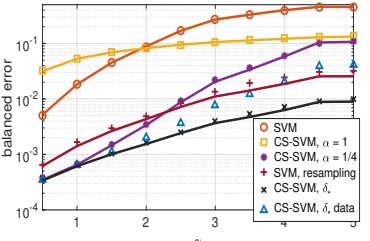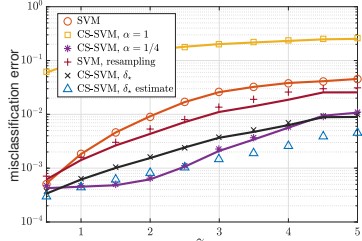

**Figure 12:** Balanced (Left) and misclassification (Right) errors as a function of the parameterization ratio $\gamma = d/n$ for the following algorithms: SVM with and without majority class resampling, CS-SVM with different choices of $\delta = \left(\frac{1-\pi}{\pi}\right)^\alpha, \pi = 0.05$ and $\delta = \delta_\star$ (cf. Eqn. (33)) plotted for different values of $\gamma = d/n$. Solid lines show the theoretical values thanks to Theorem 2 and the discrete markers represent empirical errors over 100 realizations of the dataset. Data were generated from a GMM with $\boldsymbol{\mu}_+ = 4\mathbf{e}_1, \boldsymbol{\mu}_- = -\boldsymbol{\mu}_+ \in \mathbb{R}^{500}$, and $\pi = 0.05$. SVM with resampling outperforms SVM without resampling in terms of balanced error, but the optimally tuned CS-SVM is superior to both in terms of both balanced and misclassification errors for all values of $\gamma$.

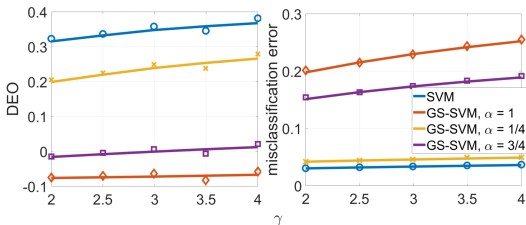

**Figure 13:** DEO and misclassification error of SVM and GS-SVM with different choices of $\delta = \left(\frac{1-p}{p}\right)^\alpha$ for minority group prior $p = 0.05$ plotted against $\gamma = d/n$. Solid lines show the theoretical values and the discrete markers represent empirical errors over 100 realizations of the dataset. Data generated from a GMM with $\boldsymbol{\mu}_{+,1} = 3\mathbf{e}_1, \boldsymbol{\mu}_{+,2} = 3\mathbf{e}_2 \in \mathbb{R}^{500}$. While SVM has the least misclassification error, it suffers from a high DEO. By trading off misclassification error, it is possible to tune GS-SVM (specifically, $\alpha = 0.75$) so that it achieves DEO close to 0 for all the values of $\gamma$ considered here.

error compared to the other two heuristic choices of $\delta = \left(\frac{1-\pi}{\pi}\right)^\alpha, \alpha = 1/4, 1$. Finally, we also evaluated the SVM+subsampling algorithm; see Section C.5.1 below for the algorithm's description and performance analysis. Observe that SVM+resampling outperforms SVM without resampling in terms of balanced error, but the optimally tuned CS-SVM is superior to both.

### C.5.1 Max-margin SVM with random majority class undersampling

For completeness, we briefly discuss here SVM combined with undersampling, a popular technique that first randomly undersamples majority examples and only then trains max-margin SVM. The asymptotic performance of this scheme under GMM can be analyzed using Theorem 2 as explained below.

Suppose the majority class is randomly undersampled to ensure equal size of the two classes. This increases the effective overparameterization ratio by a factor of $\frac{1}{2\pi}$ (in the asymptotic limits). In particular, the conditional risks converge as follows:

$$\mathcal{R}_{+,\text{undersampling}}(\gamma, \pi) \xrightarrow{P} \overline{\mathcal{R}}_{+,\text{undersampling}}(\gamma, \pi) = \overline{\mathcal{R}}_+\left(\frac{\gamma}{2\pi}, 0.5\right)$$

$$\mathcal{R}_{-,\text{undersampling}}(\gamma, \pi) \xrightarrow{P} \overline{\mathcal{R}}_{-,\text{undersampling}}(\gamma, \pi) = \overline{\mathcal{R}}_{+,\text{undersampling}}(\gamma, \pi). \tag{7}$$

Above, $\mathcal{R}_{+,\text{undersampling}}$ and $\mathcal{R}_{-,\text{undersampling}}$ are the class-conditional risks of max-margin SVM after random undersampling of the majority class to ensure equal number of training examples from the two classes. The risk $\overline{\mathcal{R}}_+\left(\frac{\gamma}{2\pi}, 0.5\right)$ is the asymptotic conditional risk of

a *balanced* dataset with overparameterization ratio $\frac{\gamma}{2\pi}$. This is computed as instructed in Theorem 2 for the assignments $\gamma \leftarrow \frac{\gamma}{2\pi}$ and $\pi \leftarrow 1/2$ in the formulas therein.

Our numerical simulations in Figure 12 verify the above formulas.

## D   Margin properties and implicit bias of VS-loss

### D.1   A more general version and proof of Theorem 1

We will state and prove a more general theorem to which Theorem 1 is a corollary. The new theorem also shows that the group-sensitive adjusted VS-loss in (3) converges to the GS-SVM.

**Remark 3.** *Theorem 1 and the content of this section are true for* arbitrary *linear models* $\mathbf{f_w}(\mathbf{x}) = \langle h(\mathbf{x}), \mathbf{w} \rangle$ *and feature maps* $h : \mathcal{X} \to \mathbb{R}^p$. *To lighten notation in the proofs, we assume for simplicity that $h$ is the identity map, that is $\mathbf{h}(\mathbf{x}) = \mathbf{x}$. For the general case, just substitute the raw features $\mathbf{x}_i \in \mathcal{X}$ below with their feature representation $h(\mathbf{x}_i) \in \mathbb{R}^p$.*

Consider the VS-loss empirical risk minimization (cf. (1) with $f(\mathbf{x}) = \mathbf{w}^T\mathbf{x}$):

$$\mathcal{L}(\mathbf{w}) := \sum_{i \in [n]} \ell(y_i, \mathbf{w}^T\mathbf{x}_i, g_i) := \omega_i \log\left(1 + e^{\iota_i} \cdot e^{-\Delta_i y_i(\mathbf{w}^T\mathbf{x}_i)}\right). \tag{8}$$

for strictly positive (but otherwise arbitrary) parameters $\Delta_i, \omega_i > 0$ and arbitrary $\iota_i$. For example, setting $\omega_i = \omega_{y_i,g_i}, \Delta_i = \Delta_{y_i,g_i}$ and $\iota_i = \iota_{y_i,g_i}$ recovers the general form of our binary VS-loss in (3).

Also, consider the following general cost-sensitive SVM (to which both the CS-SVM and the GS-SVM are special instances)

$$\hat{\mathbf{w}} := \arg\min_{\mathbf{w}} \|\mathbf{w}\|_2 \quad \text{subject to} \ \ y_i(\mathbf{w}^T\mathbf{x}_i) \ge 1/\Delta_i, \forall i \in [n]. \tag{9}$$

First, we state the following simple facts about the cost-sensitive max-margin classifier in (9). The proof of this claim is rather standard and is included in Section D.1.3 for completeness.

**Lemma 1.** *Assume that the training dataset is linearly separable, i.e.* $\exists \mathbf{w}$ *such that* $y_i(\mathbf{w}^T\mathbf{x}_i) \ge 1$ *for all* $i \in [n]$. *Then,* (9) *is feasible. Moreover, letting $\hat{\mathbf{w}}$ be the solution of* (9), *it holds that*

$$\frac{\hat{\mathbf{w}}}{\|\hat{\mathbf{w}}\|_2} = \arg\max_{\|\mathbf{w}\|_2=1} \min_{i \in [n]} \Delta_i y_i \mathbf{x}_i^T \mathbf{w}. \tag{10}$$

Next, we state the main result of this section connecting the VS-loss in (8) to the max-margin classifier in (9). After its statement, we show how it leads to Theorem 1; its proof is given later in Section D.1.2.

**Theorem 3** (Margin properties of VS-loss: General result). *Define the norm-constrained optimal classifier*

$$\mathbf{w}_R := \arg\min_{\|\mathbf{w}\|_2 \le R} \mathcal{L}(\mathbf{w}), \tag{11}$$

*with the loss $\mathcal{L}$ as defined in* (8) *for positive (but otherwise arbitrary) parameters $\Delta_i, \omega_i > 0$ and arbitrary $\iota_i$. Assume that the training dataset is linearly separable and let $\hat{\mathbf{w}}$ be the solution of* (9). *Then, it holds that*

$$\lim_{R\to\infty} \frac{\mathbf{w}_R}{\|\mathbf{w}_R\|_2} = \frac{\hat{\mathbf{w}}}{\|\hat{\mathbf{w}}\|_2}. \tag{12}$$

### D.1.1   Proof of Theorem 1

Theorem 1 is a corollary of Theorem 3 by setting $\omega_i = \omega_{y_i}, \iota_i = \iota_{y_i}$ and $\Delta_i = \Delta_{y_i}$. Indeed for this choice the loss in Equation (8) reduces to that in Equation (1). Also, (9) reduces to (4).

The latter follows from the equivalence of the following two optimization problems:

$$\left\{ \arg\min_{\mathbf{w}} \|\mathbf{w}\|_2 \quad \text{subject to} \quad \mathbf{w}^T \mathbf{x}_i \begin{cases} \geq 1/\Delta_+ & y_i = +1 \\ \leq -1/\Delta_- & y_i = -1 \end{cases} \right\}$$

$$= \left\{ \arg\min_{\mathbf{v}} \|\mathbf{v}\|_2 \quad \text{subject to} \quad \mathbf{v}^T \mathbf{x}_i \begin{cases} \geq \Delta_-/\Delta_+ & y_i = +1 \\ \leq -1 & y_i = -1 \end{cases} \right\},$$

which can be verified simply by a change of variables $\mathbf{v}/\Delta_- \leftrightarrow \mathbf{w}$ and $\Delta_- > 0$.

**The case of group-sensitive VS-loss.** As another immediate corollary of Theorem 3 we get an analogue of Theorem 1 for a group-imbalance data setting with $K = 2$ and balanced classes. Then, we may use the VS-loss in (8) with margin parameters $\Delta_i = \Delta_g, g = 1, 2$. From Theorem 3, we know that in the separable regime and in the limit of increasing weights, the classifier $\mathbf{w}_R$ (normalized) will converge to the solution of the GS-SVM with $\delta = \Delta_2/\Delta_1$.

### D.1.2 Proof of Theorem 3

First, we will argue that for any $R > 0$ the solution to the constrained VS-loss minimization is on the boundary, i.e.

$$\|\mathbf{w}_R\|_2 = R. \tag{13}$$

We will prove this by contradiction. Assume to the contrary that $\mathbf{w}_R$ is a point in the strict interior of the feasible set. It must then be by convexity that $\nabla \mathcal{L}(\mathbf{w}_R) = 0$. Let $\widetilde{\mathbf{w}}$ be any solution feasible in (9) (which exists as shown above) such that $y_i(\mathbf{x}_i^T \widetilde{\mathbf{w}}) \geq 1/\Delta_i$. On one hand, we have $\widetilde{\mathbf{w}}^T \nabla \mathcal{L}(\mathbf{w}_R) = 0$. On the other hand, by positivity of $\omega_i, \Delta_i, \forall i \in [n]$:

$$\widetilde{\mathbf{w}}^T \nabla \mathcal{L}(\mathbf{w}_R) = \sum_{i \in [n]} \underbrace{\frac{-\omega_i \Delta_i e^{-\Delta_i y_i \mathbf{x}_i^T \mathbf{w}_R + \iota_i}}{1 + e^{\iota_i} e^{-\Delta_i y_i \mathbf{x}_i^T \mathbf{w}_R}}}_{<0} \underbrace{y_i \widetilde{\mathbf{w}}^T \mathbf{x}_i}_{>0} < 0, \tag{14}$$

which leads to a contradiction.

Now, suppose that (12) is not true. This means that there is some $\epsilon_0 > 0$ such that there is always an arbitrarily large $R > 0$ such that $\frac{\mathbf{w}_R^T \hat{\mathbf{w}}}{\|\mathbf{w}_R\|_2 \|\hat{\mathbf{w}}\|_2} \leq 1 - \epsilon_0$. Equivalently, (in view of (13)):

$$\frac{\mathbf{w}_R^T \hat{\mathbf{w}}}{R \|\hat{\mathbf{w}}\|_2} \leq 1 - \epsilon_0. \tag{15}$$

Towards proving a contradiction, we will show that, in this scenario using $\hat{\mathbf{w}}_R = R \frac{\hat{\mathbf{w}}}{\|\hat{\mathbf{w}}\|_{\ell_2}}$ yields a strictly smaller VS-loss (for sufficiently large $R > 0$), i.e.

$$\mathcal{L}(\hat{\mathbf{w}}_R) < \mathcal{L}(\mathbf{w}_R), \qquad \text{for sufficiently large } R. \tag{16}$$

We start by upper bounding $\mathcal{L}(\hat{\mathbf{w}}_R)$. To do this, we first note from definition of $\hat{\mathbf{w}}_R$ the following margin property:

$$y_i \hat{\mathbf{w}}_R^T \mathbf{x}_i = \frac{R}{\|\hat{\mathbf{w}}\|_2} y_i \hat{\mathbf{w}}^T \mathbf{x}_i \geq \frac{R}{\|\hat{\mathbf{w}}\|_2} (1/\Delta_i) =: \frac{\bar{R}}{\Delta_i}, \tag{17}$$

where the inequality follows from feasibility of $\hat{\mathbf{w}}$ in (9) and we set $\bar{R} := R/\|\hat{\mathbf{w}}\|_2$. Then, using (17) it follows immediately that

$$\begin{aligned}
\mathcal{L}(\hat{\mathbf{w}}_R) &= \sum_{i=1}^n \omega_i \log \left( 1 + e^{\iota_i} e^{-\Delta_i y_i \hat{\mathbf{w}}_R^T \mathbf{x}_i} \right) \\
&\leq \sum_{i=1}^n \omega_i \log \left( 1 + e^{\iota_i} e^{-\frac{\bar{R}}{\Delta_i} \Delta_i} \right) \\
&= \sum_{i=1}^n \omega_i \log \left( 1 + e^{\iota_i} e^{-\bar{R}} \right) \\
&\leq \omega_{\max} n e^{\iota_{\max} - \bar{R}}. 
\end{aligned} \tag{18}$$

In the first inequality above we used (17) and non-negativity of $\omega_i, \Delta_i \geq 0$. In the last line, we have called $\omega_{\max} := \max_{i \in [n]} \omega_i > 0$ and $\iota_{\max} := \max_{i \in [n]} \iota_i > 0$.

Next, we lower bound $\mathcal{L}(\mathbf{w}_R)$. To do this, consider the vector

$$\bar{\mathbf{w}} = \frac{\|\hat{\mathbf{w}}\|_{\ell_2}}{R} \mathbf{w}_R = \mathbf{w}_R / \bar{R}.$$

By feasibility of $\mathbf{w}_R$ (i.e. $\|\mathbf{w}_R\|_2 \leq R$), note that $\|\bar{\mathbf{w}}\|_2 \leq \|\hat{\mathbf{w}}\|_2$. Also, from (15), we know that $\bar{\mathbf{w}} \neq \hat{\mathbf{w}}$. Indeed, if it were $\bar{\mathbf{w}} = \hat{\mathbf{w}} \iff \hat{\mathbf{w}}/\|\hat{\mathbf{w}}\|_2 = \mathbf{w}_R/R$, then

$$\frac{\hat{\mathbf{w}}^T \mathbf{w}_R}{R\|\hat{\mathbf{w}}\|_2} = 1,$$

which would contradict (15). Thus, it must be that $\bar{\mathbf{w}} \neq \hat{\mathbf{w}}$. From these and strong convexity of the objective function in (9), it follows that $\bar{\mathbf{w}}$ must be *infeasible* for (4). Thus, there exists at least one example $\mathbf{x}_j$, $j \in [n]$ and $\epsilon > 0$ such that

$$y_j \bar{\mathbf{w}}^T \mathbf{x}_j \leq (1 - \epsilon)(1/\Delta_j).$$

But then

$$y_j \mathbf{w}_R^T \mathbf{x}_j \leq \bar{R}(1 - \epsilon)(1/\Delta_j), \tag{19}$$

which we can use to lower bound $\mathcal{L}(\mathbf{w}_R)$ as follows:

$$\begin{aligned}
\mathcal{L}(\mathbf{w}_R) &\geq \omega_j \log\left(1 + e^{\iota_j - \Delta_j y_j \mathbf{w}_R^T \mathbf{x}_j}\right) \\
&\geq \omega_j \log\left(1 + e^{\iota_{y_j} - \bar{R}\Delta_j \frac{(1-\epsilon)}{\Delta_j}}\right) \\
&\geq \omega_{\min} \log\left(1 + e^{\iota_{\min} - \bar{R}(1-\epsilon)}\right).
\end{aligned} \tag{20}$$

The second inequality follows fron (19) and non-negativity of $\Delta_{\pm}, \omega_{\pm}$.

To finish the proof we compare (20) against (18). If $\epsilon \geq 1$, clearly $\mathcal{L}(\hat{\mathbf{w}}_R) < \mathcal{L}(\mathbf{w}_R)$ for sufficiently large $R$. Otherwise $e^{-\bar{R}(1-\epsilon)} \to 0$ with $R \to \infty$. Hence,

$$\mathcal{L}(\mathbf{w}_R) \geq \omega_{\min} \log\left(1 + e^{\iota_{\min} - \bar{R}(1-\epsilon)}\right) \geq 0.5\omega_{\min} e^{\iota_{\min} - \bar{R}(1-\epsilon)}.$$

Thus, again

$$\mathcal{L}(\hat{\mathbf{w}}_R) < \mathcal{L}(\mathbf{w}_R) \impliedby \omega_{\max} n e^{\iota_{\max} - \bar{R}} < 0.5\omega_{\min} e^{\iota_{\min} - \bar{R}(1-\epsilon)} \iff e^{\bar{R}\epsilon} > \frac{2n\omega_{\max}}{\omega_{\min}} e^{\iota_{\max} - \iota_{\min}},$$

because the right side is true by picking $R$ arbitrarily large.

### D.1.3   Proof of Lemma 1

The proof of Lemma 1 is standard, but included here for completeness. The lemma has two statements and we prove them in the order in which they appear.

**Linear separability $\implies$ feasibility of** (9)**.** Assume $\mathbf{w}$ such that $y_i(\mathbf{w}^T \mathbf{x}_i) \geq 1$ for all $i \in [n]$, which exists by assumption. Define $M := \max_{i \in [n]} \frac{1}{\Delta_i} > 0$ and consider $\widetilde{\mathbf{w}} = M\mathbf{w}$. Then, we claim that $\widetilde{\mathbf{w}}$ is feasible for (9). To check this, note that

$$\begin{aligned}
y_i = +1 &\implies \mathbf{x}_i^T \widetilde{\mathbf{w}} = M(\mathbf{x}_i^T \mathbf{w}) \geq M \geq 1/\Delta_i \quad \text{since } \mathbf{x}_i^T \mathbf{w} \geq 1, \\
y_i = -1 &\implies \mathbf{x}_i^T \widetilde{\mathbf{w}} = M(\mathbf{x}_i^T \mathbf{w}) \leq -M \leq -1/\Delta_i \quad \text{since } \mathbf{x}_i^T \mathbf{w} \leq -1.
\end{aligned}$$

Thus, $y_i(\mathbf{x}_i^T \widetilde{\mathbf{w}}) \geq 1/\Delta_i$ for all $i \in [n]$, as desired.

**Proof of** (10)**.** For the sake of contradiction let $\widetilde{\mathbf{w}} \neq \frac{\hat{\mathbf{w}}}{\|\hat{\mathbf{w}}\|_2}$ be the solution to the max-min optimization in the RHS of (10). Specifically, this means that $\|\widetilde{\mathbf{w}}\|_2 = 1$ and

$$\tilde{m} := \min_{i \in [n]} \Delta_i y_i \mathbf{x}_i^T \widetilde{\mathbf{w}} > \min_{i \in [n]} \Delta_i y_i \mathbf{x}_i^T \frac{\hat{\mathbf{w}}}{\|\hat{\mathbf{w}}\|_2} =: m.$$

We will prove that the vector $\mathbf{w}' := \widetilde{\mathbf{w}}/\tilde{m}$ is feasible in (9) and has smaller $\ell_2$-norm than $\hat{\mathbf{w}}$ contradicting the optimality of the latter. First, we check feasibility. Note that, by definition of $\tilde{m}$, for any $i \in [n]$:

$$\Delta_i y_i \mathbf{x}_i^T \mathbf{w}' = \frac{\Delta_i y_i \mathbf{x}_i^T \widetilde{\mathbf{w}}}{\tilde{m}} \geq 1,$$

Second, we show that $\|\mathbf{w}'\|_2 < \|\hat{\mathbf{w}}\|_2$:

$$\|\mathbf{w}'\|_2 = \frac{\|\widetilde{\mathbf{w}}\|_2}{\tilde{m}} = \frac{1}{\tilde{m}} < \frac{1}{m} = \frac{\|\hat{\mathbf{w}}\|_2}{\min_{i \in [n]} \Delta_i y_i \mathbf{x}_i^T \hat{\mathbf{w}}} \leq \|\hat{\mathbf{w}}\|_2,$$

where the last inequality follows by feasibility of $\hat{\mathbf{w}}$ in (9). This completes the proof of the lemma.

## D.2 Multiclass extension

In this section, we present a natural extension of Theorem 3 to the multiclass VS-loss in (2). Here, let we let the label set $\mathcal{Y} = \{1, 2, \ldots, C\}$ for a $C$-class classification setting and consider the cross-entropy VS-loss:

$$\mathcal{L}(\mathbf{W}) := \sum_{i \in [n]} \ell(y_i, \mathbf{w}_1^T \mathbf{x}_i, \ldots, \mathbf{w}_K^T \mathbf{x}_i) = \sum_{i \in [n]} \omega_{y_i} \log\left(1 + \sum_{\substack{y' \in [C] \\ y' \neq y_i}} e^{\iota_{y'} - \iota_{y_i}} e^{-(\Delta_{y_i} \mathbf{w}_{y_i}^T \mathbf{x}_i - \Delta_{y'} \mathbf{w}_{y'}^T \mathbf{x}_i)}\right),$$

(21)

where $\mathbf{W} = [\mathbf{w}_1, \ldots, \mathbf{w}_C] \in \mathbb{R}^{C \times d}$ and $\mathbf{w}_y$ is the classifier corresponding to class $y \in [C]$. We will also consider the following multiclass version of the CS-SVM in (9):

$$\hat{\mathbf{W}} = \arg\min_{\mathbf{W}} \|\mathbf{W}\|_F \quad \text{subject to } \mathbf{x}_i^T (\Delta_{y_i} \mathbf{w}_{y_i} - \Delta_{y'} \mathbf{w}_{y'}) \geq 1, \ \forall y' \neq y_i \in [C] \ \text{and} \ \forall i \in [n].$$

(22)

Similar to Lemma 1, it can be easily checked that (22) is feasible provided that the training data are separable, in the sense that

$$\exists \mathbf{W} = [\mathbf{w}_1, \ldots, \mathbf{w}_K] \text{ suc that } \mathbf{x}_i^T (\mathbf{w}_{y_i} - \mathbf{w}_{y'}) \geq 1, \forall y' \in [C], y' \neq y_i \ \text{and} \ \forall i \in [n]. \quad (23)$$

Moreover, it holds that

$$\hat{\mathbf{W}}/\|\hat{\mathbf{W}}\|_F = \arg\max_{\|\mathbf{W}\|_F = 1} \min_{i \in [n]} \min_{y' \neq y_i} \mathbf{x}_i^T (\Delta_{y_i} \mathbf{w}_{y_i} - \Delta_{y'} \mathbf{w}_{y'}).$$

The theorem below is an extension of Theorem 3 to multiclass classification.

**Theorem 4** (Margin properties of VS-loss: Multiclass)**.** *Consider a $C$-class classification problem and define the norm-constrained optimal classifier*

$$\mathbf{W}_R = \arg\min_{\|\mathbf{W}\|_F \leq R} \mathcal{L}(\mathbf{W}), \quad (24)$$

*with the loss $\mathcal{L}$ as defined in (21) for positive (but otherwise arbitrary) parameters $\Delta_y, \omega_y > 0, y \in [C]$ and arbitrary $\iota_y, y \in [C]$. Assume that the training dataset is linearly separable as in (23) and let $\hat{\mathbf{W}}$ be the solution of (22). Then, it holds that*

$$\lim_{R \to \infty} \frac{\mathbf{W}_R}{\|\mathbf{W}_R\|_F} = \frac{\hat{\mathbf{W}}}{\|\hat{\mathbf{W}}\|_2}. \quad (25)$$

*Proof.* The proof follows the same steps as in the proof of Theorem 3. Thus, we skip some details and outline only the basic calculations needed.

It is convenient to introduce the following notation, for $\ell \in [C]$:

$$p(\ell | \mathbf{x}, y, \mathbf{W}) := \frac{e^{\iota_y} e^{\Delta_y \mathbf{x}^T \mathbf{w}_\ell}}{\sum_{y' \in [C]} e^{\iota_{y'}} e^{\Delta_{y'} \mathbf{x}^T \mathbf{w}_{y'}}}.$$

In this notation, $\mathcal{L}(\mathbf{W}) = -\sum_{i\in[n]}\log\big(p(y_i|\mathbf{x}_i,y_i,\mathbf{W})\big)$ and for all $\ell\in[C]$ it holds that

$$\nabla_{\mathbf{w}_\ell}\mathcal{L}(\mathbf{W}) = \sum_{i\in[n]}\omega_{y_i}\Delta_{y_i}\left(p(\ell|\mathbf{x}_i,y_i,\mathbf{W}) - \mathbb{1}[y_i=\ell]\right)\mathbf{x}_i.$$

Thus, for any $\widetilde{\mathbf{W}}$ that is feasible in (22)

$$
\begin{aligned}
\sum_{\ell\in[C]}\widetilde{\mathbf{w}}_\ell^T\nabla_{\mathbf{w}_\ell}\mathcal{L}(\mathbf{W}) &= \sum_{i\in[n]}\sum_{\ell\in[C]}\omega_{y_i}\Delta_{y_i}\left(p(\ell|\mathbf{x}_i,y_i,\mathbf{W})-\mathbb{1}[y_i=\ell]\right)\mathbf{x}_i^T\widetilde{\mathbf{w}}_\ell\\
&= \sum_{i\in[n]}\sum_{\ell\neq y_i}\omega_{y_i}\Delta_{y_i}p(\ell|\mathbf{x}_i,y_i,\mathbf{W})\mathbf{x}_i^T\widetilde{\mathbf{w}}_\ell - \omega_{y_i}\Delta_{y_i}\left(1-p(y_i|\mathbf{x}_i,y_i,\mathbf{W})\right)\mathbf{x}_i^T\widetilde{\mathbf{w}}_{y_i}\\
&= \sum_{i\in[n]}\underbrace{-\omega_{y_i}\Big(\sum_{\ell\neq y_i}p(\ell|\mathbf{x}_i,y_i,\mathbf{W})\Big)}_{<0}\underbrace{\Delta_{y_i}\mathbf{x}_i^T\left(\widetilde{\mathbf{w}}_\ell - \widetilde{\mathbf{w}}_{y_i}\right)}_{>0} \;<\;0,
\end{aligned}
$$

where in the third line we used that $\sum_{\ell\in[C]}p(\ell|\mathbf{x},y,\mathbf{W})=1$. With the above it can be shown following the exact same argument as in the proof of (13) for the binary case that $\|\mathbf{W}_R\|_F = R$, the minimizer of (24) satisfies the constraint with equality.

The proof continues with a contradiction argument similar to the binary case. Assume the desired (25) does not hold. We will then show that for $\hat{\mathbf{W}}_R = \frac{R}{\|\hat{\mathbf{W}}\|_F}\hat{\mathbf{W}}$ and sufficiently large $R>0$: $\mathcal{L}(\hat{\mathbf{W}}_R) < \mathcal{L}(\mathbf{W}_R)$.

Using feasibility of $\hat{\mathbf{W}}$ in (22) and defining $\omega_{\max} := \max_{y\in[C]}\omega_y$ and $\iota_{\max} = \max_{y\neq y'\in[C]}\iota_{y'}-\iota_y$, it can be shown similar to (18) that

$$\mathcal{L}(\hat{\mathbf{W}}_R) = \sum_{i\in[n]}\omega_{y_i}\log\Big(1+\sum_{\substack{y'\in[C]\\y'\neq y_i}}e^{\iota_{y'}-\iota_{y_i}}e^{-(R/\|\mathbf{W}\|_F)(\Delta_{y_i}\hat{\mathbf{w}}_{y_i}^T\mathbf{x}_i - \Delta_{y'}\hat{\mathbf{w}}_{y'}^T\mathbf{x}_i)}\Big),$$

$$\leq n\omega_{\max}\log\Big(1+(K-1)e^{\iota_{\max}}e^{-R/\|\hat{\mathbf{W}}\|_F}\Big)\leq n(K-1)e^{\iota_{\max}}e^{-R/\|\hat{\mathbf{W}}\|_F}. \tag{26}$$

Next, by contradiction assumption and strong convexity of (22), for $\bar{\mathbf{W}} = \frac{\|\hat{\mathbf{W}}\|_2}{R}\mathbf{W}_R$, there exist $\epsilon>0$ and at least one $j\in[n]$ and $y'\neq y_j$ such that $\mathbf{x}_j^T\left(\Delta_{y_j}\bar{\mathbf{w}}_j - \Delta_{y'}\bar{\mathbf{w}}_{y'}\right)\leq(1-\epsilon)$. With this, we can show similar to (20) that

$$\mathcal{L}(\mathbf{W}_R)\geq\log\Big(1+e^{\iota_{y'}-\iota_{y_j}}e^{R/\|\hat{\mathbf{W}}\|_F(1-\epsilon)}\Big). \tag{27}$$

The proof is complete by showing that for sufficiently large $R$ the RHS of (27) is larger than the LHS of (26) leading to a contradiction. We omit the details for brevity. $\square$

### D.3 Implicit bias of Gradient flow with respect to VS-loss

Theorem 3 does not consider the effect of the optimization algorithm. Instead here, we study gradient flow (the limit of gradient descent for infinitesimal step-size) and characterize its implicit bias when applied to the VS-loss. Similar, to Theorem 3, we find that the iterations of gradient flow converge to the solution of a corresponding CS-SVM. For simplicity, we consider a VS-type adjusted exponential loss $\ell(t)=e^{-t}$, rather than logistic loss $\ell(t)=\log(1+e^{-t})$. Recent work makes it clear that both loss functions have similar implicit biases and similar lines of arguments are used to analyze the convergence properties [29, 28]. Thus, one would expect that insights also apply to logistic loss.

**Theorem 5** (Implicit bias of the gradient flow). *Consider the gradient flow iteration $\dot{\mathbf{w}}_t = -\nabla\mathcal{L}(\mathbf{w}_t)$, on the exponential VS-loss $\mathcal{L}(\mathbf{w}) = \sum_{i\in[n]}\omega_i\exp(-\Delta_i y_i\mathbf{x}_i^T\mathbf{w}+\iota_i)$. Recall that $\hat{\mathbf{w}}$ is the solution to the CS-SVM in (9). For almost every dataset which is linearly separable and any starting point $\mathbf{w}_0$ the gradient flow iterates will behave as $\mathbf{w}(t) = \hat{\mathbf{w}}\log(t) + \boldsymbol{\rho}_t$ with a bounded residual $\boldsymbol{\rho}_t$ so that $\lim_{t\to\infty}\frac{\mathbf{w}_t}{\|\mathbf{w}_t\|_2} = \frac{\hat{\mathbf{w}}}{\|\hat{\mathbf{w}}\|_2}$.*

Note that [66] previously studied the implicit bias of the gradient flow on standard CE or exponential loss. The theorem above studies the gradient flow applied to the VS-loss and its proof is similar to [66].

*Proof.* Let $\mathcal{S} \subset [n]$ be the set of indices such that $\forall i \in \mathcal{S} : \Delta_i y_i \mathbf{x}_i^T \hat{\mathbf{w}} = 1$, i.e. the set of support vectors of the CS-SVM. By KKT conditions (eg. see Equation (38)), there exist $\epsilon_i > 0$ such that $\hat{\mathbf{w}} = \sum_{i \in \mathcal{S}} \epsilon_i y_i \mathbf{x}_i$. Moreover, by [66, Lemma 12], for almost all datasets it is true that $|\mathcal{S}| \le d$ and $i \in \mathcal{S} \implies \epsilon_i > 0$. Thus, for almost all datasets we can define vector $\widetilde{\mathbf{w}}$ satisfying the following equation $\omega_i \Delta_i \exp(-\Delta_i y_i \mathbf{x}_i^T \widetilde{\mathbf{w}} + \iota_i) = \epsilon_i, \forall i \in [S]$. Note then that

$$\hat{\mathbf{w}} = \sum_{i \in [S]} \omega_i \Delta_i e^{-\Delta_i y_i \mathbf{x}_i^T \widetilde{\mathbf{w}} + \iota_i} y_i \mathbf{x}_i \tag{28}$$

Let us define $\mathbf{r}_t = \boldsymbol{\rho}_t - \widetilde{\mathbf{w}} = \mathbf{w}_t - \log(t)\hat{\mathbf{w}} - \widetilde{\mathbf{w}}$. It suffices to show that $\|\mathbf{r}(t)\|_2$ is bounded, since that would automatically give $\boldsymbol{\rho}_t$ is bounded. By the gradient flow equation, we have that

$$\dot{\mathbf{r}}_t = -\nabla \mathcal{L}(\mathbf{w}_t) - \frac{\hat{\mathbf{w}}}{t} = \sum_{i \in [n]} \omega_i \Delta_i y_i e^{-\Delta_i y_i \mathbf{x}_i^T \mathbf{w}_t + \iota_i} \mathbf{x}_i - \frac{\hat{\mathbf{w}}}{t}.$$

Therefore,

$$\frac{1}{2}\frac{\mathrm{d}}{\mathrm{d}t}\|\mathbf{r}_t\|_2^2 = \dot{\mathbf{r}}_t^T \mathbf{r}_t = \sum_{i \in [n]} \omega_i \Delta_i y_i e^{-\Delta_i y_i \mathbf{x}_i^T \mathbf{w}_t + \iota_i} \mathbf{x}_i^T \mathbf{r}_t - \frac{1}{t}\hat{\mathbf{w}}^T \mathbf{r}_t$$

$$= \underbrace{\sum_{i \in \mathcal{S}} \omega_i \Delta_i y_i e^{-\Delta_i y_i \mathbf{x}_i^T \mathbf{w}_t + \iota_i} \mathbf{x}_i^T \mathbf{r}_t - \frac{1}{t}\hat{\mathbf{w}}^T \mathbf{r}_t}_{:=A} + \underbrace{\sum_{i \notin \mathcal{S}} \omega_i \Delta_i y_i e^{-\Delta_i y_i \mathbf{x}_i^T \mathbf{w}_t + \iota_i} \mathbf{x}_i^T \mathbf{r}_t}_{:=B} \tag{29}$$

We now study the two terms $A$ and $B$ separately. In doing so, recall that $\mathbf{w}_t = \mathbf{r}_t + \log(t)\hat{\mathbf{w}} + \widetilde{\mathbf{w}}$. Hence, using the fact that $\Delta_i y_i \mathbf{x}_i^T \hat{\mathbf{w}} = \begin{cases} = 1 & i \in \mathcal{S} \\ \ge m > 1 & i \notin \mathcal{S} \end{cases}$, it holds that

$$\exp\left(-\Delta_i y_i \mathbf{x}_i^T \mathbf{w}_t + \iota_i\right) \begin{cases} = \frac{1}{t} \cdot \exp\left(-\Delta_i y_i \mathbf{x}_i^T \mathbf{r}_t\right) \cdot \exp\left(-\Delta_i y_i \mathbf{x}_i^T \widetilde{\mathbf{w}} + \iota_i\right) & i \in \mathcal{S} \\ \le \frac{1}{t^m} \cdot \exp\left(-\Delta_i y_i \mathbf{x}_i^T \mathbf{r}_t\right) \cdot \exp\left(-\Delta_i y_i \mathbf{x}_i^T \widetilde{\mathbf{w}} + \iota_i\right) & i \notin \mathcal{S} \end{cases}$$

Using this and (28), the term $A$ becomes

$$A = \frac{1}{t} \sum_{i \in [S]} \omega_i e^{-\Delta_i y_i \mathbf{x}_i^T \widetilde{\mathbf{w}} + \iota_i} \cdot e^{-\Delta_i y_i \mathbf{x}_i^T \mathbf{r}_t} \Delta_i y_i \mathbf{x}_i^T \mathbf{r}_t - \frac{1}{t} \sum_{i \in [S]} \omega_i \Delta_i e^{-\Delta_i y_i \mathbf{x}_i^T \widetilde{\mathbf{w}} + \iota_i} y_i \mathbf{x}_i^T \mathbf{r}_t$$

$$= \frac{1}{t} \sum_{i \in [S]} \omega_i e^{-\Delta_i y_i \mathbf{x}_i^T \widetilde{\mathbf{w}} + \iota_i} \cdot \left(e^{-\Delta_i y_i \mathbf{x}_i^T \mathbf{r}_t} \Delta_i y_i \mathbf{x}_i^T \mathbf{r}_t - \Delta_i y_i \mathbf{x}_i^T \mathbf{r}_t\right) \le 0,$$

since $\forall x, x \ge xe^{-x}$.

Similarly, for term $B$:

$$B \le \frac{1}{t^m} \sum_{i \notin \mathcal{S}} \omega_i e^{-\Delta_i y_i \mathbf{x}_i^T \widetilde{\mathbf{w}} + \iota_i} \cdot e^{-\Delta_i y_i \mathbf{x}_i^T \mathbf{r}_t} \cdot \Delta_i y_i \mathbf{x}_i^T \mathbf{r}_t \le \frac{1}{t^m} \sum_{i \notin \mathcal{S}} \omega_i e^{-\Delta_i y_i \mathbf{x}_i^T \widetilde{\mathbf{w}} + \iota_i}, \tag{30}$$

since $\forall x, xe^{-x} \le 1$.

To finish the proof it only takes now using the above bounds on $A, B$ and integrating both sides of Equation (29). This gives that for all $t_0, t > t_0$, there exists finite constant $C$ such that $\|\mathbf{r}_t\|^2 \le \|\mathbf{r}_{t_0}\|^2 + C$ where it was critical that $m > 1$ in (30) for the corresponding integral to be finite. This proves that $\|\mathbf{r}_t\|_2$ is bounded as desired. $\qquad\square$

We note that the above proof is a straightforward extension of [66] for analysis of CDT, with a simple rescaling of the features of the training set according to the labels, however the analysis for VS-loss with additive logit-adjustments (although similar) cannot be obtained as a special case of [66].

### D.4   Numerical illustrations of Theorems 1 and 5

Figure 14 numerically demonstrate the validity of Theorems 1 and 5. Here, we solved the VS-loss in Equation (1) using gradient descent (GD) for GMM data with class imbalance

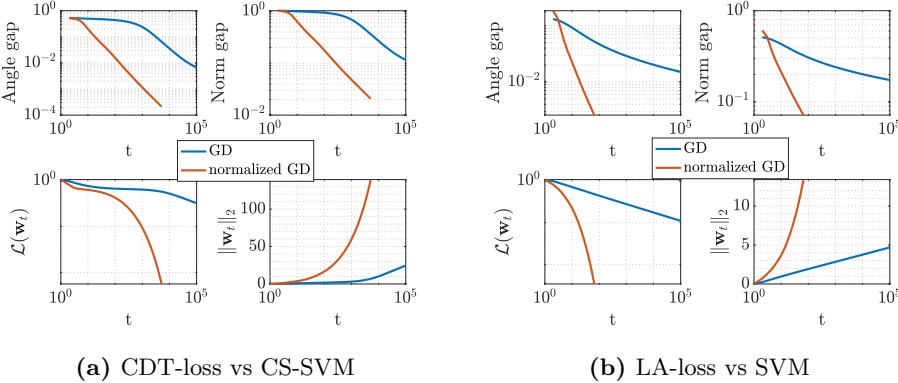

**(a)** CDT-loss vs CS-SVM                    **(b)** LA-loss vs SVM

**Figure 14:** Convergence properties of GD (blue) and normalized GD (red) iterates $\mathbf{w}_t, t \geq 1$ on VS-loss with $f_{\mathbf{w}}(x) = \mathbf{w}^T \mathbf{x}$ for two set of parameter choices: (a) $\omega_y = 1, \iota_y = 0, \Delta_y = \delta\mathbb{1}[y=1] + \mathbb{1}[y=-1]$ (aka CDT-loss) with $\delta = 20$; (b) $\omega_y = 1, \iota_y = \pi^{-1/4}\mathbb{1}[y=1] + (1-\pi)^{-1/4}\mathbb{1}[y=-1], \Delta_y = 1$ (aka LA-loss). We plotted the angle gap $1 - \frac{\hat{\mathbf{w}}^T\mathbf{w}_t}{\|\mathbf{w}_t\|_2\|\hat{\mathbf{w}}\|_2}$ and norm gap $\|\frac{\mathbf{w}_t}{\|\mathbf{w}_t\|_2} - \frac{\hat{\mathbf{w}}}{\|\hat{\mathbf{w}}\|_2}\|_2$ of $\mathbf{w}_t$ to $\hat{\mathbf{w}}$, for two values of $\hat{\mathbf{w}}$ for the two subfigures as follows: (a) $\hat{\mathbf{w}}$ is the CS-SVM solution in (4) with parameter $\delta$; (b) $\hat{\mathbf{w}}$ is the standard SVM solution. Data were generated from a Gaussian mixture model with $\boldsymbol{\mu}_1 = 2\mathbf{e}_1, \boldsymbol{\mu}_2 = -3\mathbf{e}_1 \in \mathbb{R}^{220}$, $n = 100$ and $\pi = 0.1$. For (standard) GD we used a constant rate $\eta_t = 0.1$. For normalized GD, we used $\eta_t = \frac{1}{\sqrt{t}\|\nabla\mathcal{L}(\mathbf{w}_t)\|_2}$ as suggested in [51].

$\pi = 0.1$. We ran two experiments for two choices of parameters in (1) corresponding to CDT-loss (with non-trivial multiplicative weights) and the LA-loss (with non-trivial additive weights); see the figure's caption for details. For each iterate outcome $\mathbf{w}_t$ of GD, we report the (i) angle and (ii) vector-norm gap to CS-SVM and SVM for the VS-loss and LA-loss, respectively, as well as, the (iii) value of the loss $\mathcal{L}(\mathbf{w}_t)$ and the (iv) norm of the weights $\|\mathbf{w}_t\|_2$ at current iteration. Observe that the loss $\mathcal{L}(\mathbf{w}_t)$ is driven to zero and the norm of the weights $\|\mathbf{w}_t\|_2$ increases to infinity with increasing $t$.

The experiment confirms that the VS-loss converges (aka angle/norm gap vanishes) to the CS-SVM solution, while the LA-loss converges to the SVM.

In Figure 14, we also study (curves in red) the convergence properties of *normalized GD*. Following [51], we implemented a version of normalized GD that uses a variable learning rate $\eta_t$ at iteration $t$ normalized by the gradient of the loss as follows: $\eta_t = \frac{1}{\|\nabla\mathcal{L}(\widetilde{\mathbf{w}})\|_2\sqrt{t+1}}$. [51] (see also [29]) demonstrated that this normalization speeds up the convergence of standard logistic loss to SVM. Figure 14 suggests that the same is true for convergence of the VS-loss to the CS-SVM.

# E  Optimal tuning of CS-SVM

## E.1  An explicit formula for optimal tuning

The parameter $\delta$ in the CS-SVM constraints in (4) aims to shift the decision space towards the majority class so that it better balances the conditional errors of the two classes. But, how to best choose $\delta$ to achieve that? That is, how to find $\arg\min_\delta \mathcal{R}_+(\delta) + \mathcal{R}_-(\delta)$ where $\mathcal{R}_\pm(\delta) := \mathcal{R}_\pm\big((\hat{\mathbf{w}}_\delta, \hat{b}_\delta)\big)$? Thanks to Theorem 2, we can substitute this hard, data-dependent parameter optimization problem with an analytic form that only depends on the problem parameters $\pi, \gamma$ and $\mathbf{M}$. Specifically, we seek to solve the following optimization problem

$$\arg\min_{\delta>0} \; Q(\mathbf{e}_1^T\mathbf{V}\mathbf{S}\boldsymbol{\rho}_\delta + b_\delta/q_\delta) + Q(-\mathbf{e}_2^T\mathbf{V}\mathbf{S}\boldsymbol{\rho}_\delta - b_\delta/q_\delta)$$

$$\text{sub. to} \quad (q_\delta, \boldsymbol{\rho}_\delta, b_\delta) \text{ defined as (42).} \tag{31}$$

Compared to the original data-dependent problem, the optimization above has the advantage that it is explicit in terms of the problem parameters. However, as written, the optimization

is still cumbersome as even a grid search over possible values of $\delta$ requires solving the non-linear equation (42) for each candidate value of $\delta$. Instead, we can exploit a structural property of CS-SVM (see Lemma 2 in Section E.2) to rewrite (31) in a more convenient form. Specifically, we will show in Section E.3 that (31) is equivalent to the following *explicit minimization*:

$$\arg\min_{\delta>0} \; Q\Big(\ell_+ + \big(\frac{\delta-1}{\delta+1}\big)q_1^{-1}\Big) + Q\Big(\ell_- - \big(\frac{\delta-1}{\delta+1}\big)q_1^{-1}\Big), \tag{32}$$

where we defined $\ell_+ := \mathbf{e}_1^T \mathbf{VS}\boldsymbol{\rho}_1 + b_1/q_1$, $\ell_- := -\mathbf{e}_2^T \mathbf{VS}\boldsymbol{\rho}_1 - b_1/q_1$, and, $(q_1, \boldsymbol{\rho}_1, b_1)$ are as defined in Theorem 2 for $\delta = 1$. In other words, $(q_1, \boldsymbol{\rho}_1, b_1)$ are the parameters related to the standard hard-margin SVM, for which the balanced error is then given by $(Q(\ell_+) + Q(\ell_-))/2$. To summarize, we have shown that one can optimally tune $\delta$ to minimize the *asymptotic* balanced error by minimizing the objective in (32) that only depends on the parameters $(q_1, \boldsymbol{\rho}_1, b_1)$ characterizing the asymptotic performance of SVM. In fact, we obtain explicit formulas for the optimal value $\delta_\star$ in (32) as follows

$$\delta_\star := \big(\ell_- - \ell_+ + 2q_1^{-1}\big)\big/\big(\ell_+ - \ell_- + 2q_1^{-1}\big)_+, \tag{33}$$

where it is understood that when the denominator is zero (i.e. $\ell_+ - \ell_- + 2q_1^{-1} \le 0$) then $\delta_\star \to \infty$. When $\ell_+ - \ell_- + 2q_1^{-1} > 0$, setting $\delta = \delta_\star$ in (4) not only achieves minimum balanced error among all other choices of $\delta$, but also it achieves perfect balancing between the conditional errors of the two classes, i.e. $\mathcal{R}_+ = \mathcal{R}_- = Q(\frac{\ell_- + \ell_+}{2})$.

Formally, we have the following result.

**Theorem 6** (Optimal tuning of CS-SVM). *Fix $\gamma > \gamma_\star$. Let $\overline{\mathcal{R}}_{bal}(\delta)$ denote the asymptotic balanced error of the CS-SVM with margin-ratio parameter $\delta > 0$ as specified in Theorem 2. Further let $(q_1, \boldsymbol{\rho}_1, b_1)$ the solution to (42) for $\delta = 1$. Finally, define*

$$\ell_+ := \mathbf{e}_1^T \mathbf{VS}\boldsymbol{\rho}_1 + b_1/q_1, \quad \ell_- := -\mathbf{e}_2^T \mathbf{VS}\boldsymbol{\rho}_1 - b_1/q_1,$$

*Then, for all $\delta > 0$ it holds that*

$$\overline{\mathcal{R}}_{bal}(\delta) \ge \overline{\mathcal{R}}_{bal}(\delta_\star)$$

*where $\delta_\star$ is defined as*

$$\delta_\star = \begin{cases} \frac{\ell_- - \ell_+ + 2q_1^{-1}}{\ell_+ - \ell_- + 2q_1^{-1}} & \text{if } \ell_+ + \ell_- \ge 0 \text{ and } \ell_+ - \ell_- + 2q_1^{-1} > 0, \\ \to \infty & \text{if } \ell_+ + \ell_- \ge 0 \text{ and } \ell_+ - \ell_- + 2q_1^{-1} \le 0, \\ \to 0 & \text{if } \ell_+ + \ell_- < 0. \end{cases} \tag{34}$$

*Specifically, if $\ell_+ + \ell_- \ge 0$ and $\ell_+ - \ell_- + 2q_1^{-1} > 0$ hold, then the following two hold: (i) $\overline{\mathcal{R}}_{bal}(\delta_\star) = Q\big((\ell_- + \ell_+)/2\big)$, and, (ii) the asymptotic conditional errors are equal, i.e. $\mathcal{R}_+(\delta_\star) = \mathcal{R}_-(\delta_\star)$.*

See Figures 15c and 16 for numerical illustrations of the formula in Theorem 6, specifically how $\delta_\star$ depends on $\pi$ and $\gamma$.

### E.1.1   Data-dependent heuristic to estimate $\delta_\star$

It is natural to ask if formula (34) can be used for tuning in practice. To answer this, observe that evaluating the formula requires knowledge of the true means, which are typically unknown. In this section, we propose a *data-dependent heuristic* to estimate $\delta_\star$. More generally, tuning $\delta$ (or $\Delta_y$ in VS-loss) requires a train-validation split by creating a balanced validation set from the original training data which would help assess balanced risk. Since there is only a single hyperparameter we expect this approach to work well with fairly small validation data (without hurting the minority class sample size).

Recall from Equation (33) that $\delta_\star := \big(\ell_- - \ell_+ + 2q_1^{-1}\big)\big/\big(\ell_+ - \ell_- + 2q_1^{-1}\big)_+$, where $\ell_+ := \mathbf{e}_1^T \mathbf{VS}\boldsymbol{\rho}_1 + b_1/q_1$ and $\ell_- := -\mathbf{e}_2^T \mathbf{VS}\boldsymbol{\rho}_1 - b_1/q_1$. Also, according to Theorem 2 and for $\delta = 1$ it holds that

$$\big(\|\hat{\mathbf{w}}_1\|_2, \hat{\mathbf{w}}_1^T \boldsymbol{\mu}_+/\|\hat{\mathbf{w}}_1\|_2, \hat{\mathbf{w}}_1^T \boldsymbol{\mu}_-/\|\hat{\mathbf{w}}_1\|_2, \hat{b}_1\big) \xrightarrow{P} (q_1, \mathbf{e}_1^T \mathbf{VS}\boldsymbol{\rho}_1, \mathbf{e}_2^T \mathbf{VS}\boldsymbol{\rho}_1, b_1). \tag{35}$$

The first key observation here is that $\hat{\mathbf{w}}_1, \hat{b}_1$ are the solutions to SVM, thus they are data-dependent quantities to which we have access to. Hence, we can simply run SVM and

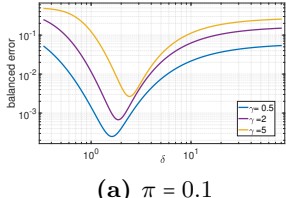 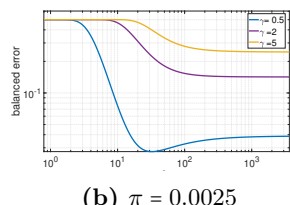 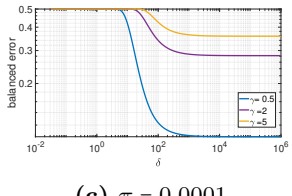

**(a)** $\pi = 0.1$            **(b)** $\pi = 0.0025$            **(c)** $\pi = 0.0001$

**Figure 15:** Graphical illustration of the result of Theorem 6: Balanced errors of CS-SVM against the margin-ratio parameter $\delta$ for a GMM of antipodal means with $\|\mu_+\| = \|\mu_-\| = 4$ and different minority class probabilities $\pi$. The balanced error is computed using the formulae of Theorem 2. For each case, we studied three different values of $\gamma$. The value $\delta_\star$ at which the curves attain (or approach) their minimum are predicted by Theorem 6. Specifically, note the following for the three different priors. (a) For all values of $\gamma$, the minimum is attained (cf. first branch of (34)). (b) For $\gamma = 2, 5$ the minimum is approached in the limit $\delta \to \infty$ (cf. second branch of (34)), but it is attained for $\gamma = 0.5$ (c) The minimum is always approached as $\delta_\star \to \infty$.

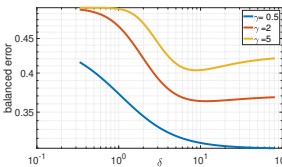

**Figure 16:** An example showing the dependence of $\delta_\star$ on the data geometry. The above figure is similar to Fig 15 but with a smaller $\|\mu_+\| = \|\mu_-\| = 1$, and for $\pi = 0.1$. While in Fig 15, the value of $\delta_\star$, whenever finite, can be seen to increase with increase in $\gamma$, for the current setting, it is observed to decrease. Note also that $\delta_\star \to \infty$ for $\gamma = 0.5$, but finite for $\gamma = 2, 5$.

estimate $q_1$ and $b_1$ using Equation (35). Unfortunately, to further estimate $\boldsymbol{\rho}_1$ we need knowledge of the data means. When this is not available, we propose approximating the data means by a simple average of the features, essentially pretending that the data follow a GMM.

Concretely, our recipe for approximating the optimal $\delta$ is as follows. First, using the training set we calculate the empirical means for the two classes, $\tilde{\boldsymbol{\mu}}_+$ and $\tilde{\boldsymbol{\mu}}_-$. (Ideally, this can be done on a balanced validation set.) Then, we train standard SVM on the same set of data and keep track of the coefficients $\hat{\mathbf{w}}_1$ and the intercept $\hat{b}_1$. Then, we can reasonably approximate the optimal $\delta$ as:

$$\tilde{\delta}_\star := \frac{\tilde{\ell}_- - \tilde{\ell}_+ + 2\|\hat{\mathbf{w}}_1\|_2^{-1}}{\left(\tilde{\ell}_+ - \tilde{\ell}_- + 2\|\hat{\mathbf{w}}_1\|_2^{-1}\right)_+}, \quad \text{with } \tilde{\ell}_+ := \frac{\hat{\mathbf{w}}_1^T \tilde{\boldsymbol{\mu}}_+ + \hat{b}_1}{\|\hat{\mathbf{w}}_1\|_2}, \quad \tilde{\ell}_- := -\frac{\hat{\mathbf{w}}_1^T \tilde{\boldsymbol{\mu}}_- + \hat{b}_1}{\|\hat{\mathbf{w}}_1\|_2}. \tag{36}$$

We expect this data-dependent theory-driven heuristic to perform reasonably well on data that resemble the GMM. For example, this is confirmed by our experiments in Figures 5 and 12. More generally, we propose tuning $\delta$ with a train-validation split by creating a balanced validation set from the original training data which would help assess balanced risk. Since there is only a single hyperparameter we expect this approach to work well with a fairly small validation data (without hurting the minority class sample size).

### E.2 CS-SVM as post-hoc weight normalization

We need the lemma below to prove Theorem 6. But the results is interesting on its own right as it allows us to view CS-SVM as an appropriate "post-hoc weight normalization"-approach.

**Lemma 2.** *Let $(\hat{\mathbf{w}}_1, \hat{b}_1)$ be the hard-margin SVM solution. Fix any $\delta > 0$ in (4) and define:* $\hat{\mathbf{w}}_\delta := \left(\frac{\delta+1}{2}\right)\hat{\mathbf{w}}_1$ *and* $\hat{b}_\delta := \left(\frac{\delta+1}{2}\right)\hat{b}_1 + \left(\frac{\delta-1}{2}\right)$. *Then, $(\hat{\mathbf{w}}_\delta, \hat{b}_\delta)$ is optimal in (4).*

Thus, classification using (4) is equivalent to the following. First learn $(\hat{\mathbf{w}}_1, \hat{b}_1)$ via standard hard-margin SVM, and then simply predict: $\hat{y} = \text{sign}\left((\hat{\mathbf{w}}_1^T\mathbf{x} + \hat{b}_1) + \frac{\delta-1}{\delta+1}\right)$. The term $\frac{\delta-1}{\delta+1}$ can be

seen as an additive form of post-hoc weight normalization to account for class imbalances. In the literature this post-hoc adjustment of the threshold $b$ of standard SVM is often referred to as boundary-movement SVM (BM-SVM) [65, 76]. Here, we have shown the equivalence of CS-SVM to BM-SVM for a specific choice of the boundary shift. The proof of Lemma 2 presented in Appendix E.2 shows the desired using the KKT conditions of (4).

*Proof.* From optimality of $(\hat{\mathbf{w}}_1, \hat{b}_1)$, convexity of (4) and the KKT-conditions, there exist dual variables $\beta_i, i \in [n]$ such that:

$$\hat{\mathbf{w}}_1 = \sum_{i \in [n]} y_i \beta_i \mathbf{x}_i, \quad \sum_{i \in [n]} y_i \beta_i = 0, \tag{37}$$

$$\forall i \in [n] \ : \ \beta_i \big( \mathbf{x}_i^T \hat{\mathbf{w}}_1 + \hat{b}_1 \big) = \beta_i y_i, \quad \beta_i \geq 0.$$

Let $(\hat{\mathbf{w}}_\delta, \hat{b}_\delta)$ defined as in the statement of the lemma and further define $\epsilon_i := \big(\frac{\delta+1}{2}\big)\beta_i, \ i \in [n]$. Then, it only takes a few algebra steps using (37) to check that the following conditions hold:

$$\hat{\mathbf{w}}_\delta = \sum_{i \in [n]} y_i \epsilon_i \mathbf{x}_i, \quad \sum_{i \in [n]} y_i \epsilon_i = 0, \tag{38}$$

$$\forall i \in [n] \ : \ \epsilon_i \big( \mathbf{x}_i^T \hat{\mathbf{w}}_\delta + \hat{b}_\delta \big) = \epsilon_i \cdot \begin{cases} \delta & \text{, if } y_i = +1 \\ -1 & \text{, if } y_i = -1 \end{cases}, \quad \epsilon_i \geq 0.$$

It can also be verified that (38) are the KKT conditions of the CS-SVM with parameter $\delta$. This proves that $(\hat{\mathbf{w}}_\delta, \hat{b}_\delta)$ is optimal in (4) as desired. □

### E.3  Proof of Theorem 6

As discussed in the section above the proof proceeds in two steps:

(i) First, starting from (31), we prove (32).

(ii) Second, we analytically solve (32) to derive the explicit expression for $\delta_\star$ in (34).

**Proof of** (32).  Fix any $\delta > 0$. From Lemma 2,

$$\hat{\mathbf{w}}_\delta = \big( \frac{\delta+1}{2} \big)\hat{\mathbf{w}}_1 \quad \text{and} \quad \hat{b}_\delta = \big( \frac{\delta+1}{2} \big)\hat{b}_1 + \big( \frac{\delta-1}{2} \big). \tag{39}$$

Recall from Theorem 2 that $\|\hat{\mathbf{w}}_\delta\|_2 \xrightarrow{P} q_\delta$, $\|\hat{\mathbf{w}}_1\|_2 \xrightarrow{P} q_1$, $\hat{b}_\delta \xrightarrow{P} b_\delta$, $\hat{b}_1 \xrightarrow{P} b_1$, and, for $i = 1, 2$: $\frac{\hat{\mathbf{w}}_\delta^T \boldsymbol{\mu}_i}{\|\hat{\mathbf{w}}_\delta\|_2} \xrightarrow{P} \mathbf{e}_i^T \mathbf{V} \mathbf{S} \boldsymbol{\rho}_\delta$ and $\frac{\hat{\mathbf{w}}_1^T \boldsymbol{\mu}_i}{\|\hat{\mathbf{w}}_1\|_2} \xrightarrow{P} \mathbf{e}_i^T \mathbf{V} \mathbf{S} \boldsymbol{\rho}_1$. Here, $q_\delta, \rho_\delta, b_\delta$ and $q_1, \rho_1, b_1$ are as defined in Theorem 2. Thus, from (39) we find that

$$\boldsymbol{\rho}_\delta = \boldsymbol{\rho}_1, \qquad q_\delta = \big( \frac{\delta+1}{2} \big)q_1 \qquad \text{and} \qquad b_\delta = \big( \frac{\delta+1}{2} \big)b_1 + \big( \frac{\delta-1}{2} \big). \tag{40}$$

Hence, it holds:

$$Q\big( \mathbf{e}_1^T \mathbf{V} \mathbf{S} \boldsymbol{\rho}_\delta + b_\delta/q_\delta \big) = Q\Big( \underbrace{\mathbf{e}_1^T \mathbf{V} \mathbf{S} \boldsymbol{\rho}_\delta + b_1/q_1}_{=\ell_+} + \frac{\delta-1}{\delta+1}q_1^{-1} \Big).$$

A similar expression can be written for the conditional error of class $-1$. Putting these together shows (32), as desired.

**Proof of** (34).  Recall from (32) that we now need to solve the following constrained minimization where for convenience we call $a = \ell_+$, $b = \ell_-$ and $c = q_1^{-1}$:

$$\min_{\delta > 0} \ Q\Big( a + \frac{\delta-1}{\delta+1}c \Big) + Q\Big( b - \frac{\delta-1}{\delta+1}c \Big).$$

We define a new variable $x = \frac{\delta-1}{\delta+1}c$. The constraint $\delta > 0$ then writes $x \leq c$. This is because the function $\delta \in (0, \infty) \mapsto \frac{\delta-1}{\delta+1}$ is onto the interval $(-1, 1)$.

Thus, we equivalently need to solve

$$\min_{-c<x<c} \ f(x) \coloneqq Q(a+x) + Q(b-x).$$

Define function $f(x) = Q(a+x) + Q(b-x)$ for some $a, b \in \mathbb{R}$. Direct differentiation gives $\frac{df}{dx} = \frac{1}{\sqrt{2\pi}}\left(e^{-(b-x)^2/2} - e^{-(a+x)^2/2}\right)$. Furthermore, note that $\lim_{x\to\pm\infty} f(x) = 1$. With thes and some algebra it can be checked that $f(\cdot)$ behaves as follows depending on the sign of $a+b$. Denote $x_\star = (b-a)/2$.

- If $a + b \geq 0$, then $1 > f(x) \geq f(x_\star)$ and $x_\star$ is the unique minimum.
- If $a + b < 0$, then $1 < f(x) \leq f(x_\star)$ and $x_\star$ is the unique maximum.

Thus, we conclude with the following:

$$\arg\inf_{-c<x<c} f(x) = \begin{cases} x_\star & \text{if } a+b \geq 0 \text{ and } b-a < 2c, \\ c & \text{if } a+b \geq 0 \text{ and } b-a \geq 2c, \\ -c & \text{if } a+b < 0. \end{cases}$$

Equivalently,

$$\arg\inf_{\delta>-1} \ Q\left(\ell_+ + \frac{\delta-1}{\delta+1}q_1^{-1}\right) + Q\left(\ell_- - \frac{\delta-1}{\delta+1}q_1^{-1}\right) = \begin{cases} \frac{\ell_- - \ell_+ + 2q_1^{-1}}{\ell_+ - \ell_- + 2q_1^{-1}} & \text{if } \ell_+ + \ell_- \geq 0 \text{ and } \ell_+ - \ell_- + 2q_1^{-1} > 0, \\ \infty & \text{if } \ell_+ + \ell_- \geq 0 \text{ and } \ell_+ - \ell_- + 2q_1^{-1} \leq 0, \\ 0 & \text{if } \ell_+ + \ell_- < 0. \end{cases}$$

This shows (34). The remaining statement of the theorem is easy to prove requiring simple algebra manipulations.

## F  Asymptotic analysis of CS-SVM

### F.1  Preliminaries

The main goal of this appendix is proving Theorem 2. For fixed $\delta > 0$, let $(\hat{\mathbf{w}}, \hat{b})$ be the solution to the CS-SVM in (4). (See also (48) below.) In the following sections, we will prove the following convergence properties for the solution of the CS-SVM:

$$(\|\hat{\mathbf{w}}\|_2, \frac{\hat{\mathbf{w}}^T\boldsymbol{\mu}_+}{\|\hat{\mathbf{w}}\|_2}, \frac{\hat{\mathbf{w}}^T\boldsymbol{\mu}_-}{\|\hat{\mathbf{w}}\|_2}, \hat{b}) \xrightarrow{P} (q_\delta, \mathbf{e}_1^T\mathbf{VS}\boldsymbol{\rho}_\delta, \mathbf{e}_2^T\mathbf{VS}\boldsymbol{\rho}_\delta, b_\delta). \tag{41}$$

where the triplet $(q_\delta, \boldsymbol{\rho}_\delta, b_\delta)$ is as defined in the theorem's statement, that is, the unique triplet satisfying

$$\eta_\delta(q_\delta, \boldsymbol{\rho}_\delta, b_\delta) = 0 \quad \text{and} \quad (\boldsymbol{\rho}_\delta, b_\delta) \coloneqq \arg\min_{\|\boldsymbol{\rho}\|_2\leq 1, b\in\mathbb{R}} \eta_\delta(q_\delta, \boldsymbol{\rho}, b). \tag{42}$$

In this section, we show how to use (41) to derive the asymptotic limit of the conditional class probabilities.

Consider the class conditional $\mathcal{R}_+ = \mathbb{P}\left\{(\mathbf{x}^T\hat{\mathbf{w}} + b) < 0 \,|\, y = +1\right\}$. Recall that conditioned on $y = +1$, we have $\mathbf{x} = \boldsymbol{\mu}_+ + \mathbf{z}$ for $\mathbf{z} \sim \mathcal{N}(\mathbf{0}, \mathbf{I})$. Thus, the class conditional can be expressed explicitly in terms of the three summary quantities on the left hand side of (41) as follows:

$$\begin{aligned} \mathcal{R}_+ &= \mathbb{P}\left\{(\mathbf{x}^T\hat{\mathbf{w}} + \hat{b}) < 0 \,|\, y = +1\right\} = \mathbb{P}\left\{\mathbf{z}^T\hat{\mathbf{w}} + \boldsymbol{\mu}_+^T\hat{\mathbf{w}} + \hat{b} < 0 \,|\, y = +1\right\} \\ &= \mathbb{P}\left\{\mathbf{z}^T\hat{\mathbf{w}} > \boldsymbol{\mu}_+^T\hat{\mathbf{w}} + \hat{b}\right\} \\ &= \mathbb{P}_{G\sim\mathcal{N}(0,1)}\left\{G\|\hat{\mathbf{w}}\|_2 > \boldsymbol{\mu}_+^T\hat{\mathbf{w}} + \hat{b}\right\} = \mathbb{P}_{G\sim\mathcal{N}(0,1)}\left\{G > \frac{\boldsymbol{\mu}_+^T\hat{\mathbf{w}}}{\|\hat{\mathbf{w}}\|_2} + \frac{\hat{b}}{\|\hat{\mathbf{w}}\|_2}\right\} \\ &= Q\left(\frac{\boldsymbol{\mu}_+^T\hat{\mathbf{w}}}{\|\hat{\mathbf{w}}\|_2} + \frac{\hat{b}}{\|\hat{\mathbf{w}}\|_2}\right). \end{aligned}$$

Then, the theorem's statement follows directly by applying (41) in the expression above.

In order to prove the key convergence result in (41) we rely on the convex Gaussian min-max theorem (CGMT) framework. We give some necessary background before we proceed with the proof.

## F.2 Background and related literature

**Related works:** Our asymptotic analysis of the CS-SVM fits in the growing recent literature on sharp statistical performance asymptotics of convex-based estimators, e.g. [9, 18, 72, 74] and references therin. The origins of these works trace back to the study of sharp phase transitions in compressed sensing, e.g. see [72] for historical remarks and performance analysis of the LASSO estimator for sparse signal recovery. That line of work led to the development of two analysis frameworks: (a) the approximate message-passing (AMP) framework [8, 19], and, (b) the convex Gaussian min-max theorem (CGMT) framework [67, 73]. More recently, these powerful tools have proved very useful for the analysis of linear classifiers [64, 49, 15, 34, 47, 39, 70, 12, 4, 69]. Theorems 2 and 7 rely on the CGMT and contribute to this line of work. Specifically, our results are most closely related to [15] who first studied max-margin type classifiers together with [49].

**CGMT framework:** Specifically, we rely on the CGMT framework. Here, we only summarize the framework's essential ideas and refer the reader to [73, 72] for more details and precise statements. Consider the following two Gaussian processes:

$$X_{\mathbf{w},\mathbf{u}} := \mathbf{u}^T \mathbf{A} \mathbf{w} + \psi(\mathbf{w}, \mathbf{u}), \tag{43a}$$

$$Y_{\mathbf{w},\mathbf{u}} := \|\mathbf{w}\|_2 \mathbf{h}_n^T \mathbf{u} + \|\mathbf{u}\|_2 \mathbf{h}_d^T \mathbf{w} + \psi(\mathbf{w}, \mathbf{u}), \tag{43b}$$

where: $\mathbf{A} \in \mathbb{R}^{n \times d}$, $\mathbf{h}_n \in \mathbb{R}^n$, $\mathbf{h}_d \in \mathbb{R}^d$, they all have entries iid Gaussian; the sets $\mathcal{S}_{\mathbf{w}} \subset \mathbb{R}^d$ and $\mathcal{S}_{\mathbf{u}} \subset \mathbb{R}^n$ are compact; and, $\psi : \mathbb{R}^d \times \mathbb{R}^n \to \mathbb{R}$. For these two processes, define the following (random) min-max optimization programs, which are refered to as the *primary optimization* (PO) and the *auxiliary optimization* (AO) problems:

$$\Phi(\mathbf{A}) = \min_{\mathbf{w} \in \mathcal{S}_{\mathbf{w}}} \max_{\mathbf{u} \in \mathcal{S}_{\mathbf{u}}} X_{\mathbf{w},\mathbf{u}}, \tag{44a}$$

$$\phi(\mathbf{h}_n, \mathbf{h}_d) = \min_{\mathbf{w} \in \mathcal{S}_{\mathbf{w}}} \max_{\mathbf{u} \in \mathcal{S}_{\mathbf{u}}} Y_{\mathbf{w},\mathbf{u}}. \tag{44b}$$

According to the first statement of the CGMT Theorem 3 in [73] (this is only a slight reformulation of Gordon's original comparison inequality [21]), for any $c \in \mathbb{R}$, it holds:

$$\mathbb{P}\{\Phi(\mathbf{A}) < c\} \le 2\mathbb{P}\{\phi(\mathbf{h}_n, \mathbf{h}_d) < c\}. \tag{45}$$

In other words, a high-probability lower bound on the AO is a high-probability lower bound on the PO. The premise is that it is often much simpler to lower bound the AO rather than the PO. However, the real power of the CGMT comes in its second statement, which asserts that if the PO is *convex* then the AO in can be used to tightly infer properties of the original PO, including the optimal cost and the optimal solution. More precisely, if the sets $\mathcal{S}_{\mathbf{w}}$ and $\mathcal{S}_{\mathbf{u}}$ are convex and *bounded*, and $\psi$ is continuous *convex-concave* on $\mathcal{S}_{\mathbf{w}} \times \mathcal{S}_{\mathbf{u}}$, then, for any $\nu \in \mathbb{R}$ and $t > 0$, it holds [73]:

$$\mathbb{P}\{|\Phi(\mathbf{A}) - \nu| > t\} \le 2\mathbb{P}\{|\phi(\mathbf{h}_n, \mathbf{h}_d) - \nu| > t\}. \tag{46}$$

In words, concentration of the optimal cost of the AO problem around $q^*$ implies concentration of the optimal cost of the corresponding PO problem around the same value $q^*$. Asymptotically, if we can show that $\phi(\mathbf{h}_n, \mathbf{h}_d) \xrightarrow{P} q^*$, then we can conclude that $\Phi(\mathbf{A}) \xrightarrow{P} q^*$.

In the next section, we will show that we can indeed express the CS-SVM in (4) as a PO in the form of (44a). Thus, the argument above will directly allow us to determine the asymptotic limit of the optimal cost of the CS-SVM. In our case, the optimal cost equals $\|\hat{\mathbf{w}}\|_2$; thus, this shows the first part of (41). For the other parts, we will employ the following

"deviation argument" of the CGMT framework [73]. For arbitrary $\epsilon > 0$, consider the desired set

$$\mathcal{S} := \left\{ (\mathbf{v}, c) \ \middle| \ \max\left\{ \|\mathbf{v}\|_2 - q_\delta|, \left| \frac{\mathbf{v}^T \boldsymbol{\mu}_+}{\|\mathbf{v}\|_2} - \mathbf{e}_1^T \mathbf{V}\mathbf{S}\boldsymbol{\rho}_\delta \right|, , \left| \frac{\mathbf{v}^T \boldsymbol{\mu}_-}{\|\mathbf{v}\|_2} - \mathbf{e}_2^T \mathbf{V}\mathbf{S}\boldsymbol{\rho}_\delta \right|, |c - b_\delta| \right\} \le \epsilon \right\}. \quad (47)$$

Our goal towards (41) is to show that with overwhelming probability $(\mathbf{w}, b) \in \mathcal{S}$. For this, consider the following constrained CS-SVM that further constraints the feasible set to the complement $\mathcal{S}^c$ of $\mathcal{S}$:

$$\Phi_{\mathcal{S}^c}(\mathbf{A}) := \min_{(\mathbf{w},b) \in \mathcal{S}^c} \|\mathbf{w}\|_2 \text{ sub. to } \begin{cases} \mathbf{w}^T \mathbf{x}_i + b \ge \delta & , y_i = +1 \\ \mathbf{w}^T \mathbf{x}_i + b \le -1 & , y_i = -1 \end{cases}, i \in [n], \quad (48)$$

As per Theorem 6.1(iii) in [72] it will suffice to find constants $\bar{\phi}, \bar{\phi}_S$ and $\eta > 0$ such that the following three conditions hold:

$$\begin{cases} \text{(i)} & \bar{\phi}_S \ge \bar{\phi} + 3\eta \\ \text{(ii)} & \phi(\mathbf{h}_n, \mathbf{h}_d) \le \bar{\phi} + \eta \quad \text{with overwhelming probability} \\ \text{(iii)} & \phi_{\mathcal{S}^c}(\mathbf{h}_n, \mathbf{h}_d) \ge \bar{\phi}_S - \eta \quad \text{with overwhelming probability,} \end{cases} \quad (49)$$

where $\phi_{\mathcal{S}^c}(\mathbf{h}_n, \mathbf{h}_d)$ is the optimal cost of the constrained AO corresponding to the constrained PO in (48).

To prove these conditions for the AO of the CS-SVM, in the next section we follow the principled machinery of [72] that allows simplifying the AO from a (random) optimization over vector variables to an easier optimization over only few scalar variables, termed the "scalarized AO".

### F.3 Proof of Theorem 2

Let $(\hat{\mathbf{w}}, \hat{b})$ be solution pair to the CS-SVM in (4) for some fixed margin-ratio parameter $\delta > 0$, which we rewrite here expressing the constraints in matrix form:

$$\min_{\mathbf{w},b} \|\mathbf{w}\|_2 \text{ sub. to } \begin{cases} \mathbf{w}^T \mathbf{x}_i + b \ge \delta, \ y_i = +1 \\ -(\mathbf{w}^T \mathbf{x}_i + b) \ge 1, \ y_i = -1 \end{cases}, \ i \in [n] \quad = \quad \min_{\mathbf{w},b} \|\mathbf{w}\|_2 \text{ sub. to } \mathbf{D_y}(\mathbf{Xw} + b\mathbf{1}_n) \ge \boldsymbol{\delta_y}, \quad (50)$$

where we have used the notation

$$\mathbf{X}^T = \begin{bmatrix} \mathbf{x}_1 & \cdots & \mathbf{x}_n \end{bmatrix}, \ \mathbf{y} = \begin{bmatrix} y_1 & \cdots & y_n \end{bmatrix}^T,$$

$$\mathbf{D_y} = \text{diag}(\mathbf{y}) \text{ and } \boldsymbol{\delta_y} = \begin{bmatrix} \delta\mathbb{1}[y_1 = +1] + \mathbb{1}[y_1 = -1] & \cdots & \delta\mathbb{1}[y_n = +1] + \mathbb{1}[y_n = -1] \end{bmatrix}^T.$$

We further need to define the following one-hot-encoding of the labels:

$$\mathbf{y}_i = \mathbf{e}_1 \mathbb{1}[y_i = 1] + \mathbf{e}_2 \mathbb{1}[y_i = -1], \quad \text{and} \quad \mathbf{Y}_{n \times 2}^T = \begin{bmatrix} \mathbf{y}_1 & \cdots & \mathbf{y}_n \end{bmatrix}.$$

where recall that $\mathbf{e}_1, \mathbf{e}_2$ are standard basis vectors in $\mathbb{R}^2$.

With these, notice for later use that under our model, $\mathbf{x}_i = \boldsymbol{\mu}_{y_i} + \mathbf{z}_i = \mathbf{M}\mathbf{y}_i + \mathbf{z}_i, \ \mathbf{z}_i \sim \mathcal{N}(0, 1)$. Thus, in matrix form with $\mathbf{Z}$ having entries $\mathcal{N}(0, 1)$:

$$\mathbf{X} = \mathbf{YM}^T + \mathbf{Z}. \quad (51)$$

Following the CGMT strategy [73], we express (50) in a min-max form to bring it in the form of the PO as follows:

$$\min_{\mathbf{w},b} \max_{\mathbf{u} \le 0} \frac{1}{2} \|\mathbf{w}\|_2^2 + \mathbf{u}^T \mathbf{D_y} \mathbf{Xw} + b(\mathbf{u}^T \mathbf{D_y} \mathbf{1}_n) - \mathbf{u}^T \boldsymbol{\delta_y}$$

$$= \min_{\mathbf{w},b} \max_{\mathbf{u} \le 0} \frac{1}{2} \|\mathbf{w}\|_2^2 + \mathbf{u}^T \mathbf{D_y} \mathbf{Zw} + \mathbf{u}^T \mathbf{D_y} \mathbf{YM}^T \mathbf{w} + b(\mathbf{u}^T \mathbf{D_y} \mathbf{1}_n) - \mathbf{u}^T \boldsymbol{\delta_y}. \quad (52)$$

where in the last line we used (51) and $\mathbf{D_y D_y} = \mathbf{I}_n$. We immediately recognize that the last optimization is in the form of a PO (cf. (44a)) and the corresponding AO (cf. (44b)) is as follows:

$$\min_{\mathbf{w},b} \max_{\mathbf{u} \leq 0} \quad \frac{1}{2}\|\mathbf{w}\|_2^2 + \|\mathbf{w}\|_2 \mathbf{u}^T \mathbf{D_y h}_n + \|\mathbf{D_y u}\|_2 \mathbf{h}_d^T \mathbf{w} + \mathbf{u}^T \mathbf{D_y Y M}^T \mathbf{w} + b(\mathbf{u}^T \mathbf{D_y 1}_n) - \mathbf{u}^T \boldsymbol{\delta_y}.$$

(53)

where $\mathbf{h}_n \sim \mathcal{N}(0, \mathbf{I}_n)$ and $\mathbf{h}_d \sim \mathcal{N}(0, \mathbf{I}_d)$.

In order to apply the CGMT in [73], we need boundedness of the constraint sets. Thus, we restrict the minimization in (53) and (52) to a bounded set $\|\mathbf{w}\|_2^2 + b^2 \leq R$ for (say) $R := 2\left(q_\delta^2 + b_\delta^2\right)$. This will allow us to show that the solutions $\hat{\mathbf{w}}_R, \hat{b}_R$ of this constrained PO satisfy $\hat{\mathbf{w}}_R \xrightarrow{P} q_\delta$ and $\hat{b}_R \xrightarrow{P} b_\delta$. Thus, with overwhelming probability, $\|\hat{\mathbf{w}}_R\|_2^2 + \hat{b}_R^2 < R$. From this and convexity of the PO, we can argue that the minimizers $\hat{\mathbf{w}}, \hat{b}$ of the original unconstrained problem satisfy the same convergence properties. Please see also Remark 4 in App. A of [15].

For the maximization, we follow the recipe in App. A of [15] who analyzed the standard SVM. Specifically, combining Remark 3 of [15] together with (we show this next) the property that the AO is reduced to a convex program, it suffices to consider the unconstrained maximization.

Thus, in what follows we consider the one-sided constrained AO in (53). Towards simplifying this auxiliary optimization, note that $\mathbf{D_y h}_n \sim \mathbf{h}_n$ by rotational invariance of the Gaussian measure. Also, $\|\mathbf{D_y u}\|_2 = \|\mathbf{u}\|_2$. Thus, we can express the AO in the following more convenient form:

$$\min_{\|\mathbf{w}\|_2^2 + b^2 \leq R} \max_{\mathbf{u} \leq 0} \quad \frac{1}{2}\|\mathbf{w}\|_2^2 + \|\mathbf{w}\|_2 \mathbf{u}^T \mathbf{h}_n + \|\mathbf{u}\|_2 \mathbf{h}_d^T \mathbf{w} + \mathbf{u}^T \mathbf{D_y Y M}^T \mathbf{w} + b(\mathbf{u}^T \mathbf{D_y 1}_n) - \mathbf{u}^T \boldsymbol{\delta_y}. \quad (54)$$

We are now ready to proceed with simplification of the AO. First we optimize over the direction of $\mathbf{u}$ and rewrite the AO as

$$\min_{\|\mathbf{w}\|_2^2 + b^2 \leq R} \max_{\beta \geq 0} \quad \frac{1}{2}\|\mathbf{w}\|_2^2 + \beta \left( \left\| \left( \|\mathbf{w}\|_2 \mathbf{h}_n + \mathbf{D_y Y M}^T \mathbf{w} + b\,\mathbf{D_y 1}_n - \boldsymbol{\delta_y} \right)_- \right\|_2 - \mathbf{h}_d^T \mathbf{w} \right)$$

$$= \min_{\|\mathbf{w}\|_2^2 + b^2 \leq R} \quad \frac{1}{2}\|\mathbf{w}\|_2^2 \quad \text{sub. to} \quad \left\| \left( \|\mathbf{w}\|_2 \mathbf{h}_n + \mathbf{D_y Y M}^T \mathbf{w} + b\,\mathbf{D_y 1}_n - \boldsymbol{\delta_y} \right)_- \right\|_2 \leq \mathbf{h}_d^T \mathbf{w}.$$

Above, $(\cdot)_-$ acts elementwise to the entries of its argument.

Now, we wish to further simplify the above by minimizing over the direction of $\mathbf{w}$ in the space orthogonal to $\mathbf{M}$. To see how this is possible consider the SVD $\mathbf{M}^T = \mathbf{VSU}^T$ and project $\mathbf{w}$ on the columns of $\mathbf{U} = [\mathbf{u}_1 \quad \mathbf{u}_2] \in \mathbb{R}^{d \times 2}$ as follows:

$$\mathbf{w} = \mathbf{u}_1(\mathbf{u}_1^T \mathbf{w}) + \mathbf{u}_2(\mathbf{u}_2^T \mathbf{w}) + \mathbf{w}^\perp,$$

where $\mathbf{w}^\perp = \mathbf{U}^\perp \mathbf{w}$, $\mathbf{U}^\perp$ is the orthogonal complement of $\mathbf{U}$. For simplicity we will assume here that $\mathbf{M}$ is full column rank, i.e. $\mathbf{S} > \mathbf{0}_{2 \times 2}$. The argument for the case where $\mathbf{M}$ is rank 1 is very similar.

Let us denote $\mathbf{u}_i^T \mathbf{w} := \mu_i, i = 1, 2$ and $\|\mathbf{w}^\perp\|_2 := \alpha$. In this notation, the AO becomes

$$\min_{\mu_1^2 + \mu_2^2 + \|\mathbf{w}^\perp\|_2^2 + b^2 \leq R} \quad \frac{1}{2}(\mu_1^2 + \mu_2^2 + \alpha^2)$$

$$\text{sub. to} \quad \left\| \left( \sqrt{\mu_1^2 + \mu_2^2 + \alpha^2}\,\mathbf{h}_n + \mathbf{D_y Y V S} \begin{bmatrix} \mu_1 \\ \mu_2 \end{bmatrix} + b\,\mathbf{D_y 1}_n - \boldsymbol{\delta_y} \right)_- \right\|_2$$

$$\leq \mu_1(\mathbf{h}_d^T \mathbf{u}_1) + \mu_2(\mathbf{h}_d^T \mathbf{u}_2) + \mathbf{h}_d^T \mathbf{U}^\perp \mathbf{w}^\perp.$$

At this point, we can optimize over the direction of $\mathbf{w}^\perp$ which leads to

$$\min_{\mu_1^2 + \mu_2^2 + \alpha^2 + b^2 \leq R} \quad \frac{1}{2}(\mu_1^2 + \mu_2^2 + \alpha^2)$$

$$\text{sub. to} \quad \left\| \left( \sqrt{\mu_1^2 + \mu_2^2 + \alpha^2}\,\mathbf{h}_n + \mathbf{D_y Y V S} \begin{bmatrix} \mu_1 \\ \mu_2 \end{bmatrix} + b\,\mathbf{D_y 1}_n - \boldsymbol{\delta_y} \right)_- \right\|_2$$

$$\leq \mu_1(\mathbf{h}_d^T \mathbf{u}_1) + \mu_2(\mathbf{h}_d^T \mathbf{u}_2) + \alpha \|\mathbf{h}_d^T \mathbf{U}^\perp\|_2.$$

As a last step in the simplification of the AO, it is convenient to introduce an additional variable $q = \sqrt{\mu_1^2 + \mu_2^2 + \alpha^2}$. It then follows that the minimization above is equivalent to the following

$$\min_{\substack{q \geq \sqrt{\mu_1^2 + \mu_2^2 + \alpha^2} \\ q^2 + b^2 \leq R}} \frac{1}{2} q^2 \tag{55}$$

sub. to $\quad \left\| \left( q\mathbf{h}_n + \mathbf{D_y} \mathbf{YVS} \begin{bmatrix} \mu_1 \\ \mu_2 \end{bmatrix} + b\, \mathbf{D_y} \mathbf{1}_n - \boldsymbol{\delta_y} \right)_- \right\|_2 \leq \mu_1(\mathbf{h}_d^T \mathbf{u}_1) + \mu_2(\mathbf{h}_d^T \mathbf{u}_2) + \alpha \|\mathbf{h}_d^T \mathbf{U}^\perp\|_2.$

In this formulation it is not hard to check that the optimization is jointly convex in its variables $(\mu_1, \mu_2, \alpha, b, q)$. To see this note that: (i) the constraint $q \geq \sqrt{\mu_1^2 + \mu_2^2 + \alpha^2} \iff q \geq \| [\mu_1 \quad \mu_2 \quad \alpha] \|_2$ is a second-order cone constraint, and, (ii) the function

$$\mathcal{L}_n(q, \mu_1, \mu_2, \alpha, b) := \frac{1}{\sqrt{n}} \left\| \left( q\mathbf{h}_n + \mathbf{D_y} \mathbf{YVS} \begin{bmatrix} \mu_1 \\ \mu_2 \end{bmatrix} + b\, \mathbf{D_y} \mathbf{1}_n - \boldsymbol{\delta_y} \right)_- \right\|_2$$

$$- \mu_1 \frac{\mathbf{h}_d^T \mathbf{u}_1}{\sqrt{n}} - \mu_2 \frac{\mathbf{h}_d^T \mathbf{u}_2}{\sqrt{n}} - \alpha \frac{\|\mathbf{h}_d^T \mathbf{U}^\perp\|_2}{\sqrt{n}} \tag{56}$$

is also convex since $\|(\cdot)_-\|_2 : \mathbb{R}^n \to \mathbb{R}$ is itslef convex and is composed here with an affine function.

Now, by law of large numbers, notice that for fixed $(q, \mu_1, \mu_2, \alpha, b)$, $\mathcal{L}_n$ converges in probability to

$$\mathcal{L}_n(q, \mu_1, \mu_2, \alpha, b) \xrightarrow{P} L(q, \mu_1, \mu_2, \alpha, b) := \sqrt{\mathbb{E}\left( qG + E_Y^T \mathbf{VS} \begin{bmatrix} \mu_1 \\ \mu_2 \end{bmatrix} + bY - \Delta_Y \right)_-^2} - \alpha\sqrt{\gamma}, \tag{57}$$

where the random variables $G, E_Y, Y, \Delta_Y$ are as in the statement of the theorem. But convergence of convex functions is uniform over compact sets as per Cor. II.I in [3]. Therefore, the convergence in (57) is in fact uniform in the compact feasible set of (55).

Consider then the deterministic high-probability equivalent of (55) which is the following convex program:

$$\min_{\substack{q \geq \sqrt{\mu_1^2 + \mu_2^2 + \alpha^2} \\ q^2 + b^2 \leq R \\ L(q, \mu_1, \mu_2, \alpha, b) \leq 0}} \frac{1}{2} q^2.$$

Since $q$ is positive and the constraint $q \geq \sqrt{\mu_1^2 + \mu_2^2 + \alpha^2}$ must be active at the optimum, it is convenient to rewrite this in terms of new variables $\boldsymbol{\rho} = \begin{bmatrix} \boldsymbol{\rho}_1 \\ \boldsymbol{\rho}_2 \end{bmatrix} := \begin{bmatrix} \mu_1/q \\ \mu_2/q \end{bmatrix}$ as follows:

$$\min_{q^2 + b^2 \leq R, q > 0, \|\boldsymbol{\rho}\|_2 \leq 1} \frac{1}{2} q^2 \tag{58}$$

sub. to $\quad \mathbb{E}\left[ \left( G + E_Y^T \mathbf{VS}\boldsymbol{\rho} + \frac{bY - \Delta_Y}{q} \right)_-^2 \right] \leq \left( 1 - \|\boldsymbol{\rho}\|_2^2 \right) \gamma.$

Now, recall the definition of the function $\eta_\delta$ in the statement of the theorem and observe that the constraint above is nothing but

$$\eta_\delta(q, \boldsymbol{\rho}, b) \leq 0.$$

Thus, (58) becomes

$$\min \left\{ q^2 \;\middle|\; 0 \leq q \leq \sqrt{R} \quad \text{and} \quad \min_{b^2 \leq R - q^2, \|\boldsymbol{\rho}\|_2 \leq 1} \eta_\delta(q, \boldsymbol{\rho}, b) \leq 0 \right\}. \tag{59}$$

We will prove that

$$\text{the function } f(q) := \min_{b, \|\boldsymbol{\rho}\|_2 \leq 1} \eta_\delta(q, \boldsymbol{\rho}, b) \text{ is strictly decreasing.} \tag{60}$$

Before that, let us see how this completes the proof of the theorem. Let $q_\delta$ be as in the statement of the theorem, that is such that $f(q_\delta) = 0$. Then, we have the following relations

$$f(q) \le 0 \implies f(q) \le f(q_\delta) \implies q \ge q_\delta.$$

Thus, the minimizers in (59) are $(q_\delta, \boldsymbol{\rho}_\delta, b_\delta)$, where we also recall that we have set $R > q_\delta^2 + b_\delta^2$.

With all these, we have shown that the AO converges in probability to $q_\delta^2$ (cf. condition (ii) in (49)). From the CGMT, the same is true for the PO. Now, we want to use the same machinery to prove that the minimizers $(\hat{\mathbf{w}}, \hat{b})$ of the PO satisfy (41). To do this, as explained in the previous section, we use the standard strategy of the CGMT framework , i.e., to show that the PO with the additional constraint $(\mathbf{w}, b) \in \mathcal{S}^c$ for the set $\mathcal{S}$ in (47) has a cost that is strictly larger than $q_\delta^2$ (i.e. the cost of the unconstrained PO). As per the CGMT this can be done again by showing that the statement is true for the correspondingly constrained AO (i.e. show condition (iii) in (49)). With the exact same simplifications as above, the latter program simplifies to (55) with the additional constraints:

$$|q - q_\delta| > \epsilon, \, \left|\mu_i/q - \boldsymbol{\rho}_{\delta,i}\right| > \epsilon, \, i = 1, 2, \, |b - b_\delta| > \epsilon.$$

Also, using the uniform convergence in (57), it suffices to study the deterministic equivalent (59) with the additional constraints above. Now, we can show the desired (cf. condition (i) in (49)) again by exploiting (60). This part of the argument is similar to Section C.3.5 in [15] and we omit the details.

Proof of (60): To complete the proof, it remains to show (60). Specifically, we show that $\frac{df}{dq} < 0$ by combining the following three observations.

First,

$$\frac{\partial \eta_\delta}{\partial q} = \frac{2}{q^2}\mathbb{E}\Big[\big(G + E_Y^T \mathbf{V}\mathbf{S}\boldsymbol{\rho} + \frac{bY - \Delta_Y}{q}\big)_- \cdot \Delta_Y\Big] - \frac{2b}{q^2}\mathbb{E}\Big[\big(G + E_Y^T \mathbf{V}\mathbf{S}\boldsymbol{\rho} + \frac{bY - \Delta_Y}{q}\big)_- \cdot Y\Big]$$
$$< -\frac{2b}{q^2}\mathbb{E}\Big[\big(G + E_Y^T \mathbf{V}\mathbf{S}\boldsymbol{\rho} + \frac{bY - \Delta_Y}{q}\big)_- \cdot Y\Big] \tag{61}$$

where for the inequality we observed that $(\cdot)_-$ is always non-positive, its argument has non-zero probability measure on the negative real axis, and $\Delta_Y$ are positive random variables.

Second, letting $\boldsymbol{\rho}^\star := \boldsymbol{\rho}^\star(q)$ and $b^\star := b^\star(q)$ the minimizers of $\eta_\delta(q, \boldsymbol{\rho}, b)$, it follows from first-order optimality conditions that

$$\frac{\partial \eta_\delta}{\partial b} = 0 \iff \mathbb{E}\Big[\big(G + E_Y^T \mathbf{V}\mathbf{S}\boldsymbol{\rho}^* + \frac{b^* Y - \Delta_Y}{q}\big)_- \cdot Y\Big] = 0. \tag{62}$$

Third, by the envelope theorem

$$\frac{df}{dq} = \frac{\partial \eta_\delta}{\partial q}\Big|_{\boldsymbol{\rho}^\star, b^\star}. \tag{63}$$

The desired inequality $\frac{df}{dq} < 0$ follows directly by successively applying (63), (61) and (62).

**Uniqueness of triplet** $(q_\delta, \boldsymbol{\rho}_\delta, b_\delta)$**.** First, we prove that the minimizers $\boldsymbol{\rho}_\delta, b_\delta$ are unique. This follows because $\eta_\delta(q, \boldsymbol{\rho}, b)$ is jointly strictly convex in $(\boldsymbol{\rho}, b)$ for fixed $q$. To see this note that the function $x \mapsto (x)_-^2$ is strictly convex for $x < 0$ and that the random variable $G + E_Y^T \mathbf{V}\mathbf{S}\boldsymbol{\rho} + (bY - \Delta_Y)/q$ has strictly positive measure on the real line (thus, also in the negative axis). Next, consider $q_\delta$, which was defined such that $f(q_\delta) = 0$ for the function $f(\cdot)$ in (60). From (60) we know that $f(\cdot)$ is strictly decreasing. Thus, it suffices to prove that the function has a zero crossing in $(0, \infty)$, which we do by proving $\lim_{q \to 0} f(q) = \infty$ and $\lim_{q \to \infty} f(q) < 0$. Specifically, we have

$$\lim_{q \to 0} f(q) \ge \lim_{q \to 0} \min_{b \in \mathbb{R}, \|\boldsymbol{\rho}\|_2 \le 1} \mathbb{E}\Big[\big(G + E_Y^T \mathbf{V}\mathbf{S}\boldsymbol{\rho} + \frac{bY - \Delta_Y}{q}\big)_-^2\Big] - \gamma$$
$$\ge \lim_{q \to 0} \min_{b \in \mathbb{R}, \|\boldsymbol{\rho}\|_2 \le 1} \mathbb{E}\Big[\big(G + E_Y^T \mathbf{V}\mathbf{S}\boldsymbol{\rho} + \frac{bY - \Delta_Y}{q}\big)_-^2 \mathbb{1}\big[G + E_Y^T \mathbf{V}\mathbf{S}\boldsymbol{\rho} + (b/q)Y \le 0\big]\Big] - \gamma$$
$$\ge \lim_{q \to 0} \min_{b \in \mathbb{R}, \|\boldsymbol{\rho}\|_2 \le 1} 1/q^2 - \gamma = \infty,$$

where in the last inequality we used the facts that $x \mapsto (x)_-^2$ is decreasing and the event $\{G + E_Y^T \mathbf{V} \mathbf{S} \boldsymbol{\rho} + (b/q)Y \le 0 \le 0\}$ has non-zero measure for all $\|\boldsymbol{\rho}\|_2 \le 1, b \in \mathbb{R}$, as well as, $\Delta_Y \ge 1$ (because $\delta > 1$). Moreover,

$$\lim_{q \to \infty} f(q) = \lim_{1/q \to 0^+} f(q) = \lim_{1/q \to 0^+} \min_{b \in \mathbb{R}, \|\boldsymbol{\rho}\|_2 \le 1} \mathbb{E}\Big[\big(G + E_Y^T \mathbf{V} \mathbf{S} \boldsymbol{\rho} + \frac{bY - \Delta_Y}{q}\big)_-^2\Big] - (1 - \|\boldsymbol{\rho}\|_2^2)\gamma$$

$$= \lim_{1/q \to 0^+} \min_{\tilde{b} \in \mathbb{R}, \|\boldsymbol{\rho}\|_2 \le 1} \mathbb{E}\Big[\big(G + E_Y^T \mathbf{V} \mathbf{S} \boldsymbol{\rho} + \tilde{b}Y - \frac{\Delta_Y}{q}\big)_-^2\Big] - (1 - \|\boldsymbol{\rho}\|_2^2)\gamma$$

$$\le \min_{\tilde{b} \in \mathbb{R}, \|\boldsymbol{\rho}\|_2 \le 1} \mathbb{E}\Big[\big(G + E_Y^T \mathbf{V} \mathbf{S} \boldsymbol{\rho} + \tilde{b}Y\big)_-^2\Big] - (1 - \|\boldsymbol{\rho}\|_2^2)\gamma$$

$$\le \min_{\tilde{b} \in \mathbb{R}, \|\boldsymbol{\rho}\|_2 \le 1} (1 - \|\boldsymbol{\rho}\|_2^2) \cdot \Big(\mathbb{E}\Big[\big((G + E_Y^T \mathbf{V} \mathbf{S} \boldsymbol{\rho} + \tilde{b}Y)/\sqrt{1 - \|\boldsymbol{\rho}\|_2^2}\big)_-^2\Big] - \gamma\Big)$$

$$\le \min_{\tilde{b} \in \mathbb{R}, \|\boldsymbol{\rho}\|_2 \le 1} \mathbb{E}\Big[\big((G + E_Y^T \mathbf{V} \mathbf{S} \boldsymbol{\rho} + \tilde{b}Y)/\sqrt{1 - \|\boldsymbol{\rho}\|_2^2}\big)_-^2\Big] - \gamma$$

$$\le \min_{\breve{b} \in \mathbb{R}, \mathbf{t} \in \mathbb{R}^r} \mathbb{E}\Big[\big(\sqrt{1 + \|\mathbf{t}\|_2^2}\, G + E_Y^T \mathbf{V} \mathbf{S} \mathbf{t} + \breve{b}Y\big)_-^2\Big] - \gamma = \gamma_\star - \gamma < 0,$$

where to get the penultimate inequality we used the change of variables $\mathbf{t} = \boldsymbol{\rho}/\sqrt{1 - \|\boldsymbol{\rho}\|_2^2}$ and $\breve{b} = \tilde{b}/\sqrt{1 - \|\boldsymbol{\rho}\|_2^2}$. Also, in the last line above, we used the definition of the phase-transition threshold $\gamma_\star$ in Equation (64) and the theorem's assumption that $\gamma > \gamma_\star$ (aka separable regime).

We note that similar uniqueness argument was presented in [15] for the special case of antipodal means, *no* intercept and $\delta = 1$.

### F.4 Antipodal means and non-isotropic data

**Antipodal means.** In the special case of antipodal means of equal energy $\boldsymbol{\mu}_+ = -\boldsymbol{\mu}_- = \boldsymbol{\mu}$ with $s := \|\boldsymbol{\mu}\|_2$, the formulas of Theorem 2 simplify as we have $r = 1$ with $\mathbf{S} = s\sqrt{2}$ and $\mathbf{V} = [1/\sqrt{2}, -1\sqrt{2}]^T$. Now, the function $\eta_\delta$ can be written as $\mathbb{E}\big[(G + \tilde{\rho}s + \frac{\tilde{b}}{q}Y - \frac{1}{q}\Delta_Y)_-^2\big] - \big(1 - \tilde{\rho}^2\big)\gamma$. The asymptotic performance of SVM for this special geometry of the means has been recently studied in [15, 47]. We extend this to the CS-SVM classifier, to general means for the two classes and to $\boldsymbol{\Sigma} \ne \mathbf{I}$.

**Non-isotropic data.** We show how Theorem 2 for the isotropic case can still be applied in the general case $\boldsymbol{\Sigma} \ne \mathbf{I}$. Assume $\boldsymbol{\Sigma} > 0$. Write $\mathbf{x}_i = y_i \boldsymbol{\mu}_{y_i} + \boldsymbol{\Sigma}^{1/2} \mathbf{h}_i$ for $\mathbf{h}_i \sim \mathcal{N}(0, \mathbf{I}_d)$. Consider whitened features $\mathbf{z}_i := \boldsymbol{\Sigma}^{-1/2} \mathbf{x}_i = y_i \boldsymbol{\Sigma}^{-1/2} \boldsymbol{\mu}_{y_i} + \mathbf{h}_i$ and let

$$(\hat{\mathbf{w}}, \hat{b}) = \arg\min_{\mathbf{w}, b} \frac{1}{n} \sum_{i \in [n]} \ell(y_i(\mathbf{x}_i^T \mathbf{w} + b)),$$

$$(\hat{\mathbf{v}}, \hat{c}) = \arg\min_{\mathbf{v}, c} \frac{1}{n} \sum_{i \in [n]} \ell(y_i(\mathbf{z}_i^T \mathbf{v} + c)).$$

Clearly, $\hat{\mathbf{w}} = \boldsymbol{\Sigma}^{-1/2} \hat{\mathbf{v}}$ and $\hat{b} = \hat{c}$. Thus,

$$\mathcal{R}_+\big((\hat{\mathbf{w}}, \hat{b})\big) = \mathbb{P}\{(\mathbf{x}^T \hat{\mathbf{w}} + \hat{b}) < 0 \,|\, y = +1\} = \mathbb{P}\{\boldsymbol{\mu}_+^T \hat{\mathbf{w}} + \hat{\mathbf{w}}^T \boldsymbol{\Sigma}^{1/2} \mathbf{h} + \hat{b} < 0\} = Q\left(\frac{\boldsymbol{\mu}_+^T \hat{\mathbf{w}} + \hat{b}}{\|\boldsymbol{\Sigma}^{1/2} \hat{\mathbf{w}}\|_2}\right)$$

$$= Q\Big(\frac{\boldsymbol{\mu}_+^T \boldsymbol{\Sigma}^{-1/2} \hat{\mathbf{v}} + \hat{c}}{\|\hat{\mathbf{v}}\|_2}\Big) = \mathbb{P}\{(\mathbf{z}^T \hat{\mathbf{v}} + \hat{c}) < 0 \,|\, y = +1\}$$

$$= \mathcal{R}_+\big((\hat{\mathbf{v}}, \hat{c})\big)$$

Similar derivation holds for $\mathcal{R}_-$. Thus, we can just apply Theorem 2 for $\mathbf{S}, \mathbf{V}$ given by the eigendecomposition of the new Grammian $\mathbf{M}^T \boldsymbol{\Sigma}^{-1} \mathbf{M}$.

### F.5 Phase transition of CS-SVM

Here, we present a formula for the threshold $\gamma_\star$ such that the CS-SVM of (4) is feasible (resp., infeasible) with overwhelming probability provided that $\gamma > \gamma_\star$ (resp., $\gamma < \gamma_\star$). The

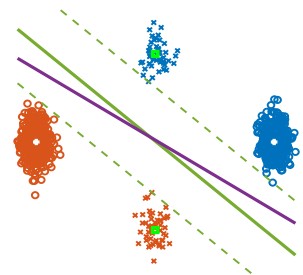

**Figure 17:** Visualizing the Gaussian mixture model of Section 4 with $K = 2$ imbalanced groups in the two-dimensional space ($d = 2$). Different colors (resp., markers) correspond to different class (resp., group) membership. Examples in the minority group correspond to cross markers (×). The means of the majority / minority groups are depicted in white / green markers. The purple line illustrates the group-sensitive SVM (GS-SVM) classifier that forces larger margin to the minority group examples in relation to standard SVM in green.

first observation is that the phase-transition threshold $\gamma_\star$ of feasibility of the CS-SVM is the same as the threshold of feasibility of the standard SVM for the same model; see Section D.1.3. Then, the desired result follows [35] who very recently established separability phase-transitions for the more general multiclass Gaussian mixture model

**Proposition 1** ([35]). *Consider the same data model and notation as in Theorem 2 and define the event*

$$\mathcal{E}_{\mathrm{sep},n} := \left\{ \exists (\mathbf{w}, b) \in \mathbb{R}^d \times \mathbb{R} \quad s.t. \quad y_i(\mathbf{w}^T \mathbf{x}_i + b) \geq 1, \quad \forall i \in [n] \right\}.$$

*Define threshold $\gamma_\star := \gamma_\star(\mathbf{V}, \mathbf{S}, \pi)$ as follows:*

$$\gamma_\star := \min_{\mathbf{t} \in \mathbb{R}^r, b \in \mathbb{R}} \mathbb{E}\left[ \left( \sqrt{1 + \|\mathbf{t}\|_2^2}\, G + E_Y^T \mathbf{V} \mathbf{S} \mathbf{t} - bY \right)_-^2 \right]. \tag{64}$$

*Then, the following hold:*

$$\gamma > \gamma_\star \Rightarrow \lim_{n \to \infty} \mathbb{P}(\mathcal{E}_{\mathrm{sep},n}) = 1 \quad and \quad \gamma < \gamma_\star \Rightarrow \lim_{n \to \infty} \mathbb{P}(\mathcal{E}_{\mathrm{sep},n}) = 0.$$

*In words, the data are linearly separable (with overwhelming probability) if and only if $\gamma > \gamma_\star$. Furthermore, if this condition holds, then CS-SVM is feasible (with overwhelming probability) for any value of $\delta > 0$.*

## G  Asymptotic analysis of GS-SVM

In Theorem 2 we derived the asymptotic generalization performance of CS-SVM under the Gaussian mixture data model. Here, we state the counterpart result for GS-SVM with an appropriate Gaussian mixture data model with group imbalances, which we repeat here for convenience.

**Data model.** We study a binary Gaussian-mixture generative model (GMM) for the data distribution $\mathcal{D}$. For the label $y \in \{\pm 1\}$ let $\pi := \mathbb{P}\{y = +1\}$. Group membership is decided conditionally on the label such that $\forall j \in [K] : \mathbb{P}\{g = j | y = \pm 1\} = p_{\pm,j}$, with $\sum_{j \in [K]} p_{+,j} = \sum_{j \in [K]} p_{-,j} = 1$. Finally, the feature conditional given label $y$ and group $g$ is a multivariate Gaussian of mean $\boldsymbol{\mu}_{y,g} \in \mathbb{R}^d$ and covariance $\boldsymbol{\Sigma}_{y,g}$, that is, $\mathbf{x} | (y, g) \sim \mathcal{N}(\boldsymbol{\mu}_{y,g}, \boldsymbol{\Sigma}_{y,g})$. We focus on two groups $K = 2$ with $p_{+,1} = p_{-,1} = p < 1 - p = p_{+,2} = p_{-,2}, j = 1, 2$ and $\mathbf{x} | (y, g) \sim \mathcal{N}(y \boldsymbol{\mu}_g, \sigma_g \mathbf{I}_d)$, for $\sigma_1^2, \sigma_2^2$ the noise variances of the minority and the majority groups, respectively. As before, let $\mathbf{M}$ denote the matrix of means (that is $\mathbf{M} = [\boldsymbol{\mu}_+ \quad \boldsymbol{\mu}_-]$ and $\mathbf{M} = [\boldsymbol{\mu}_1 \quad \boldsymbol{\mu}_2]$, respectively) and consider the eigen-decomposition of its Gramian: $\mathbf{M}^T \mathbf{M} = \mathbf{V} \mathbf{S}^2 \mathbf{V}^T, \quad \mathbf{S} \succ \mathbf{0}_{r \times r}, \mathbf{V} \in \mathbb{R}^{2 \times r}, r \in \{1, 2\}$, with $\mathbf{S}$ an $r \times r$ diagonal positive-definite matrix and $\mathbf{V}$ an orthonormal matrix obeying $\mathbf{V}^T \mathbf{V} = \mathbf{I}_r$. We study linear classifiers with $h(\mathbf{x}) = \mathbf{x}$.

**Learning regime.** Again, as in Theorem 2, we focus on a regime where training data are linearly separable. Specifically, there exists threshold $\widetilde{\gamma}_\star := \widetilde{\gamma}_\star(\mathbf{V}, \mathbf{S}, \pi, p) \leq 1/2$, such that GMM data with groups are linearly separable with probability approaching one provided that $\gamma > \widetilde{\gamma}_\star$ (see Section G.2). We assume $\gamma > \widetilde{\gamma}_\star$, so that GS-SVM is feasible with probability approaching 1.

Although similar in nature, the result below differs to Theorem 2 since now each class itself is a Gaussian mixture.

**Theorem 7** (Sharp asymptotics of GS-SVM). *Consider the GMM with feature distribution and priors as specified in the 'Data model' above. Fix $\delta > 0$ (corresponding to group VS-loss with $\Delta_{y,g} = \Delta_g, g = 1, 2$ such that $\delta = \Delta_2/\Delta_1$). Define $G, Y, S, \widetilde{\Delta}_S, \Sigma_S \in \mathbb{R}$, and $\widetilde{E}_S \in \mathbb{R}^{2\times 1}$ as follows: $G \sim \mathcal{N}(0,1)$; $Y$ is a symmetric Bernoulli with $\mathbb{P}\{Y = +1\} = \pi$; $S$ takes values 1 or 2 with probabilities $p$ and $1 - p$, respectively; $\widetilde{E}_S = \mathbf{e}_1 \mathbb{1}[S = 1] + \mathbf{e}_2 \mathbb{1}[S = 2]$; $\widetilde{\Delta}_S = \delta \cdot \mathbb{1}[S = 1] + 1 \cdot \mathbb{1}[S = 2]$ and $\Sigma_S = \sigma_1 \mathbb{1}[S = 1] + \sigma_2 \mathbb{1}[S = 2]$. With these define function $\widetilde{\eta}_\delta : \mathbb{R}_{\geq 0} \times \mathcal{S}^r \times \mathbb{R} \to \mathbb{R}$ as*

$$\widetilde{\eta}_\delta(q, \boldsymbol{\rho}, b) := \mathbb{E}\big(G + \Sigma_S^{-1} \widetilde{E}_S^T \mathbf{VS}\boldsymbol{\rho} + \frac{b\Sigma_S^{-1}Y - \Sigma_S^{-1}\widetilde{\Delta}_S}{q}\big)_-^2 - (1 - \|\boldsymbol{\rho}\|_2^2)\gamma.$$

*Let $(\widetilde{q}_\delta, \widetilde{\boldsymbol{\rho}}_\delta, \widetilde{b}_\delta)$ be the unique triplet satisfying (42) but with $\eta_\delta$ replaced with the function $\widetilde{\eta}_\delta$ above. Then, in the limit of $n, d \to \infty$ with $d/n = \gamma > \widetilde{\gamma}_\star$ it holds for $i = 1, 2$ that $\mathcal{R}_{\pm,i} \xrightarrow{P} Q\big(\mathbf{e}_i^T \mathbf{VS}\widetilde{\boldsymbol{\rho}}_\delta \pm \widetilde{b}_\delta/\widetilde{q}_\delta\big)$. In particular, $\mathcal{R}_{\mathrm{deo}} \xrightarrow{P} Q\big(\mathbf{e}_1^T \mathbf{VS}\widetilde{\boldsymbol{\rho}}_\delta + \widetilde{b}_\delta/\widetilde{q}_\delta\big) - Q\big(\mathbf{e}_2^T \mathbf{VS}\widetilde{\boldsymbol{\rho}}_\delta + \widetilde{b}_\delta/\widetilde{q}_\delta\big)$.*

### G.1   Proof of Theorem 7

The proof of Theorem 7 also relies on the CGMT framework and is very similar to the proof of Theorem 2. To avoid repetitions, we only present the part that is different. As we will show the PO is slightly different as now we are dealing with a classification between mixtures of mixtures of Gaussians. We will derive the new AO and will simplify it to a point from where the same steps as in Section F.3 can be followed mutatis mutandis.

Let $(\hat{\mathbf{w}}, \hat{b})$ be solution pair to the GS-SVM for some fixed parameter $\delta > 0$, which we rewrite here expressing the constraints in matrix form:

$$\min_{\mathbf{w},b} \|\mathbf{w}\|_2 \ \text{ sub. to } \begin{cases} y_i(\mathbf{w}^T\mathbf{x}_i + b) \geq \delta, \ g_i = 1 \\ y_i(\mathbf{w}^T\mathbf{x}_i + b) \geq 1, \ g_i = 2 \end{cases}, \ i \in [n] \ = \ \min_{\mathbf{w},b} \|\mathbf{w}\|_2 \ \text{ sub. to } \mathbf{D_y}\big(\mathbf{Xw} + b\mathbf{1}_n\big) \geq \boldsymbol{\delta_g},$$

(65)

where we have used the notation

$$\mathbf{X}^T = \begin{bmatrix} \mathbf{x}_1 & \cdots & \mathbf{x}_n \end{bmatrix}, \ \mathbf{y} = \begin{bmatrix} y_1 & \cdots & y_n \end{bmatrix}^T,$$

$$\mathbf{D_y} = \mathrm{diag}(\mathbf{y}) \ \text{and} \ \boldsymbol{\delta_g} = \begin{bmatrix} \delta\mathbb{1}[g_1 = 1] + \mathbb{1}[g_1 = 2] & \cdots & \delta\mathbb{1}[g_n = 1] + \mathbb{1}[g_n = 2] \end{bmatrix}^T.$$

We further need to define the following one-hot-encoding for group membership:

$$\mathbf{g}_i = \mathbf{e}_1 \mathbb{1}[g_i = 1] + \mathbf{e}_2 \mathbb{1}[g_i = 2], \quad \text{and} \quad \mathbf{G}_{n\times 2}^T = \begin{bmatrix} \mathbf{g}_1 & \cdots & \mathbf{g}_n \end{bmatrix}.$$

where recall that $\mathbf{e}_1, \mathbf{e}_2$ are standard basis vectors in $\mathbb{R}^2$. Finally, let

$$\mathbf{D}_\sigma = \mathrm{diag}\big(\begin{bmatrix} \sigma_{g_1} & \cdots & \sigma_{g_n} \end{bmatrix}\big).$$

With these, notice for later use that under our model, $\mathbf{x}_i = y_i\boldsymbol{\mu}_{g_i} + \sigma_{g_i}\mathbf{z}_i = y_i\mathbf{Mg}_i + \sigma_{g_i}\mathbf{z}_i$, $\mathbf{z}_i \sim \mathcal{N}(0,1)$. Thus, in matrix form with $\mathbf{Z}$ having entries $\mathcal{N}(0,1)$:

$$\mathbf{X} = \mathbf{D_y}\mathbf{GM}^T + \mathbf{D}_\sigma\mathbf{Z}.$$

(66)

As usual, we express the GS-SVM program in a min-max form to bring it in the form of the PO as follows:

$$\min_{\mathbf{w},b} \max_{\mathbf{u} \leq 0} \ \frac{1}{2}\|\mathbf{w}\|_2^2 + \mathbf{u}^T\mathbf{D_y}\mathbf{Xw} + b(\mathbf{u}^T\mathbf{D_y}\mathbf{1}_n) - \mathbf{u}^T\boldsymbol{\delta_g}$$

$$= \min_{\mathbf{w},b} \max_{\mathbf{u} \leq 0} \ \frac{1}{2}\|\mathbf{w}\|_2^2 + \mathbf{u}^T\mathbf{D_y}\mathbf{D}_\sigma\mathbf{Zw} + \mathbf{u}^T\mathbf{GM}^T\mathbf{w} + b(\mathbf{u}^T\mathbf{D_y}\mathbf{1}_n) - \mathbf{u}^T\boldsymbol{\delta_g}.$$

(67)

where in the last line we used (66) and $\mathbf{D_y D_y} = \mathbf{I}_n$. We immediately recognize that the last optimization is in the form of a PO and the corresponding AO is as follows:

$$\min_{\mathbf{w},b} \max_{\mathbf{u} \le 0} \frac{1}{2}\|\mathbf{w}\|_2^2 + \|\mathbf{w}\|_2 \mathbf{u}^T \mathbf{D_y D_\sigma h}_n + \|\mathbf{D_y D_\sigma u}\|_2 \mathbf{h}_d^T \mathbf{w} + \mathbf{u}^T \mathbf{GM}^T \mathbf{w} + b(\mathbf{u}^T \mathbf{D_y 1}_n) - \mathbf{u}^T \boldsymbol{\delta}_{\mathbf{g}}. \tag{68}$$

where $\mathbf{h}_n \sim \mathcal{N}(0, \mathbf{I}_n)$ and $\mathbf{h}_d \sim \mathcal{N}(0, \mathbf{I}_d)$.

As in Section F.3 we consider the one-sided constrained AO in (68). Towards simplifying this auxiliary optimization, note that $\mathbf{D}_y \mathbf{h}_n \sim \mathbf{h}_n$ by rotational invariance of the Gaussian measure. Also, $\|\mathbf{D_y D_\sigma u}\|_2 = \|\mathbf{D_\sigma u}\|_2$. Thus, we can express the AO in the following more convenient form:

$$\min_{\|\mathbf{w}\|_2^2 + b^2 \le R} \max_{\mathbf{u} \le 0} \frac{1}{2}\|\mathbf{w}\|_2^2 + \|\mathbf{w}\|_2 \mathbf{u}^T \mathbf{D_\sigma h}_n + \|\mathbf{D_\sigma u}\|_2 \mathbf{h}_d^T \mathbf{w} + \mathbf{u}^T \mathbf{GM}^T \mathbf{w} + b(\mathbf{u}^T \mathbf{D_y 1}_n) - \mathbf{u}^T \boldsymbol{\delta}_{\mathbf{g}}$$

$$= \min_{\|\mathbf{w}\|_2^2 + b^2 \le R} \max_{\mathbf{v} \le 0} \frac{1}{2}\|\mathbf{w}\|_2^2 + \|\mathbf{w}\|_2 \mathbf{v}^T \mathbf{h}_n + \|\mathbf{v}\|_2 \mathbf{h}_d^T \mathbf{w} + \mathbf{v}^T \mathbf{D}_\sigma^{-1} \mathbf{GM}^T \mathbf{w} + b(\mathbf{v}^T \mathbf{D}_\sigma^{-1} \mathbf{D_y 1}_n) - \mathbf{v}^T \mathbf{D}_\sigma^{-1} \boldsymbol{\delta}_{\mathbf{g}},$$

where in the second line we performed the change of variables $\mathbf{v} \leftrightarrow \mathbf{D}_\sigma \mathbf{u}$ and used positivity of the diagonal entries of $\mathbf{D}_\sigma$ to find that $\mathbf{u} \le 0 \iff \mathbf{v} \le 0$.

Notice that the optimization in the last line above is very similar to the AO (54) in Section F.3. Following analogous steps, omitted here for brevity, we obtain the following scalarized AO:

$$\min_{\substack{q \ge \sqrt{\mu_1^2 + \mu_2^2 + \alpha^2} \\ q^2 + b^2 \le R}} \frac{1}{2}q^2 \tag{69}$$

$$\text{sub. to} \quad \frac{1}{\sqrt{n}}\left\|\left( q\mathbf{h}_n + \mathbf{D}_\sigma^{-1} \mathbf{GVS}\begin{bmatrix}\mu_1 \\ \mu_2\end{bmatrix} + b\,\mathbf{D}_\sigma^{-1} \mathbf{D}_y \mathbf{1}_n - \mathbf{D}_\sigma^{-1} \boldsymbol{\delta}_{\mathbf{g}} \right)_-\right\|_2$$

$$- \mu_1 \frac{\mathbf{h}_d^T \mathbf{u}_1}{\sqrt{n}} - \mu_2 \frac{\mathbf{h}_d^T \mathbf{u}_2}{\sqrt{n}} - \alpha \frac{\|\mathbf{h}_d^T \mathbf{U}^\perp\|_2}{\sqrt{n}} \le 0.$$

where as in Section F.3 we have decomposed the matrix of means $\mathbf{M} = \mathbf{USV}^T$ and $\mu_1, \mu_2, \alpha$ above represent $\mathbf{u}_1^T \mathbf{w}, \mathbf{u}_1^T \mathbf{w}$ and $\|\mathbf{w}^\perp\|_2$. Now, by law of large numbers, notice that for fixed $(q, \mu_1, \mu_2, \alpha, b)$, the functional in the constraint above converges in probability to

$$\bar{L}(q, \mu_1, \mu_2, \alpha, b) := \sqrt{\mathbb{E}\left(qG + \Sigma_S^{-1} \widetilde{E}_S^T \mathbf{VS}\begin{bmatrix}\mu_1 \\ \mu_2\end{bmatrix} + b\Sigma_S^{-1} Y - \Sigma_S^{-1} \widetilde{\Delta}_S\right)_-^2} - \alpha\sqrt{\gamma}, \tag{70}$$

where the random variables $G, \widetilde{E}_S, Y, \widetilde{\Delta}_S$ and $\Sigma_S$ are as in the statement of the theorem. Thus, the deterministic equivalent (high-dimensional limit) of the AO expressed in variables $\boldsymbol{\rho} = \begin{bmatrix}\boldsymbol{\rho}_1 \\ \boldsymbol{\rho}_2\end{bmatrix} := \begin{bmatrix}\mu_1/q \\ \mu_2/q\end{bmatrix}$ becomes (cf. Eqn. (58)):

$$\min_{q^2 + b^2 \le R, q > 0, \|\boldsymbol{\rho}\|_2 \le 1} \frac{1}{2}q^2 \tag{71}$$

$$\text{sub. to} \quad \mathbb{E}\left(G + \Sigma_S^{-1} \widetilde{E}_S^T \mathbf{VS}\boldsymbol{\rho} + \frac{b\Sigma_S^{-1} Y - \Sigma_S^{-1} \widetilde{\Delta}_S}{q}\right)_-^2 \le \left(1 - \|\boldsymbol{\rho}\|_2^2\right)\gamma.$$

Now, recall the definition of the function $\widetilde{\eta}_\delta$ in the statement of the theorem and observe that the constraint above is nothing but

$$\widetilde{\eta}_\delta(q, \boldsymbol{\rho}, b) \le 0.$$

Thus, (71) becomes

$$\min\left\{q^2 \ \Big| \ 0 \le q \le \sqrt{R} \quad \text{and} \quad \min_{b^2 \le R - q^2, \|\boldsymbol{\rho}\|_2 \le 1} \widetilde{\eta}_\delta(q, \boldsymbol{\rho}, b) \le 0\right\}. \tag{72}$$

The remaining steps of the proof are very similar to those in Section F.3 and are omitted.

## G.2 Phase transition of GS-SVM

The phase-transition threshold $\widetilde{\gamma}_\star$ of feasibility of the GS-SVM is the same as the threshold of feasibility of the standard SVM for the same model (see Section D.1.2). But, the feasibility threshold of SVM under the group GMM with $K = 2$ groups is different from that of Section F.5 for $K = 1$, since now each class is itself a mixture of Gaussians. We derive the desired result from [35], who recently studied the separability question for the more general case of a multiclass mixture of mixtures of Gaussians.

**Proposition 2.** *Consider the same data model and notation as in Theorem 7 and consider the event*

$$\mathcal{E}_{\mathrm{sep},n} := \left\{ \exists (\mathbf{w}, b) \in \mathbb{R}^d \times \mathbb{R} \quad s.t. \quad y_i(\mathbf{w}^T \mathbf{x}_i + b) \geq 1, \ \ \forall i \in [n] \right\}.$$

*Define threshold $\gamma_\star := \gamma_\star(\mathbf{V}, \mathbf{S}, \pi)$ as follows:*

$$\widetilde{\gamma}_\star := \min_{\mathbf{t} \in \mathbb{R}^r, b \in \mathbb{R}} \mathbb{E}\left[ \left( \sqrt{1 + \|\mathbf{t}\|_2^2}\, G + \widetilde{E}_S^T \mathbf{V} \mathbf{S} \mathbf{t} - bY \right)_-^2 \right]. \tag{73}$$

*Then, the following hold:*

$$\gamma > \widetilde{\gamma}_\star \Rightarrow \lim_{n \to \infty} \mathbb{P}(\mathcal{E}_{\mathrm{sep},n}) = 1 \qquad and \qquad \gamma < \widetilde{\gamma}_\star \Rightarrow \lim_{n \to \infty} \mathbb{P}(\mathcal{E}_{\mathrm{sep},n}) = 0.$$

*In words, the data are linearly separable with overwhelming probability if and only if $\gamma > \widetilde{\gamma}_\star$. Furthermore, if this condition holds, then GS-SVM is feasible with overwhelming probability for any value of $\delta > 0$.*