# OpenReview forum: "Label-Imbalanced and Group-Sensitive Classification under Overparameterization"
_NeurIPS.cc/2021/Conference — NeurIPS 2021 Poster_

### Official Review · Reviewer_Umnd · 2021-07-14

**Rating:** 6
**Confidence:** 4

**Summary:**

This paper mainly shows two insights: 1) When optimizing in the terminal phase of training, multiplicative logit adjustment is critical. 2) Compared with multiplicative adjustment, additive adjustment can speed up the convergence at the beginning of the training. Motivated by the insights, a new vector-scaling (VS) loss is proposed to exploit the advantages of both multiplicative adjustment and additive adjustment, which is further introduced to group-sensitive settings. What’s more, a generalization analysis of the VS-loss on binary over-parameterized Gaussian mixtures is presented, revealing tradeoffs between balanced/standard error and equal opportunity.

**Ethical Concerns:**

No ethical issues are found in this paper.

**Limitations And Societal Impact:**

The authors have addressed the limitations and potential negative societal impact of their work.

**Main Review:**

Originality: This paper reveals how multiplicative and additive adjustment methods affect margins and proposes a new loss function which combines the well-known LA and CDT loss functions. Additionally, a further extension to group-sensitive classification is given.
Quality: Some insights and theoretical analyses are given in this paper, which is instructive but not sound enough. Specifically, to prove the multiplicative adjustment can be harmful at the initial phase of training, this paper claims that pushing the margin towards majorities requires Δ_+<Δ_-, which is derived from theorem 1. However, this conclusion may be improper for the following reasons: 1) Theorem 1 assumes feature mapping h(x) are linearly separable, which is unpractical. 2) according to the existing works, e.g., [1] and [2], reweighting methods could be harmful for the representation learning at the initial phase, thus it is not clear whether we should push the margin towards majorities at the beginning of network training. 3) This paper proves CDT loss function pushing the optimization in the wrong direction when f_w (x)  = 0, which can be inadequate for f_w (x)  = 0 is an extremely special case.
More experiments are required to prove the effectiveness of the proposed VS-loss. Specially, experiment results on iNaturalist, ImageNet-LT and Cifar-10/100-LT with imbalanced ratios 50 and 10 are required.
Clarity: This paper is clearly written, well-organized and provides adequate information to readers.
Significance: This paper provides some important results including the analysis and the extension to the group-sensitive learning. However, due to the lack of sufficient experiments, it if difficult to compare it with previous state-of-the-art methods.

[1] Zhou B, Cui Q, Wei X S, et al. Bbn: Bilateral-branch network with cumulative learning for long-tailed visual recognition[C]//Proceedings of the IEEE/CVF Conference on Computer Vision and Pattern Recognition. 2020: 9719-9728.
[2] Kang B, Xie S, Rohrbach M, et al. Decoupling representation and classifier for long-tailed recognition[J]. arXiv preprint arXiv:1910.09217, 2019.



**Time Spent Reviewing:**

3

---

> ### Author Response · Authors · 2021-08-10
> **Author response to Reviewer Umnd**
>
> Thank you for your time spent on our submission.
>
> We start with an important clarification regarding your comment *“...Some insights and theoretical analyses are given in this paper, which is instructive but not sound enough...”*
>
> We wish to reiterate that prior work [Ye et al.’20] on CDT-loss is purely empirical and works on additive logit-adjustment [LA, ICLR’21; LDAM, NeurIPS’19] overlook its drawbacks which are discovered by our Theorem 1. Our work is also the first to analytically study and reveal the impact and the benefit of joint utilization of the two different logit adjustment techniques (additive/multiplicative) in imbalanced learning. It is also the first to provide any generalization guarantees for such logit-adjusted loss functions and the first to use (appropriate modifications of) these losses in group-sensitive settings. Thus, we are afraid that the reviewer’s statement undermines our contributions.
>
> Specifically, it appears that the reviewer has overlooked our main contribution of Theorem 1 (and instead focuses on Observation 1; please see below). In Theorem 1 (also Theorem 6 in the SM),  we show that the recently proposed logit-adjusted (LA) loss *fails* in general to fulfill its promise of enforcing a larger margin for the minority class in the overparameterized regime. We show this clearly by proving that LA-loss converges to SVM irrespective of its parameters in the terminal phase of training. We find this result surprising and highly relevant to the NeurIPS community given that (Cao et al. NeurIPS’19, Menon et al. ICLR’20) introduced the LA-loss precisely for the purpose of tuning the minority margin.
>
> Next, we respond to your concerns regarding the "harmfulness'' of $\Delta$'s at the start of the training (Observation 1). We provide a point-by-point response to your remarks.
>
> 1. **Re:** "...*Theorem 1 assumes feature mapping $h(x)$ are linearly separable, which is unpractical*..."
>
> Our paper focuses on the overparameterized regime, where the assumption of zero training error (which implies separability) is commonplace. Specifically, both empirically (e.g. Zhang et al. ICLR'17, Belkin et al. PNAS'19, Nakkiran et al. ICLR'20, Papyan et al. PNAS’20) and theoretically (e.g. Du et al. ICLR'19, Allen-Zhu et al. ICML'19), it is well-established that sufficiently overparameterized deep networks can learn feature maps that linearly separate the data under mild conditions. In fact, simply training the output layer of a wide two layer network is sufficient to accomplish this as it corresponds to a random feature regression problem (e.g. Mei & Montanari'19, Belkin et al.‘19). Additionally, a wealth of recent theoretical works investigating properties of overparameterized learning models (e.g. implicit bias, generalization) adapt the same assumption as in our paper of linearly separable data (see for example Soudry et al.’18, Hastie et al.’19, Belkin et al.’20, Montanari et al.’19, Deng et al.’19, Mignacco’19, Aubin et al.’20,  Muthukumar et al. ‘20,  Hsu et al. ‘21 and many others).
>
>
> 2. **Re:**  ``...*this paper claims that pushing the margin towards majorities requires Δ_+<Δ_-, which is derived from theorem 1. However, this conclusion may be improper [...] according to the existing works, e.g., [1] and [2], reweighting methods could be harmful for the representation learning at the initial phase, thus it is not clear whether we should push the margin towards majorities at the beginning of network training*..."
>
> First, we remark on the condition $\Delta_+ < \Delta_-$. This condition is not made-up by us, but is standard in previous work on tuning the CDT-loss. Indeed, the empirical tuning suggested and used by (Ye et al.’20), who introduced the CDT loss,  in their experiments satisfies this condition (precisely, they set $\Delta_+=(N_+/N_-)^{0.3}<1=\Delta_-)$. Thus, in Observation 1 we reveal that this standard tuning of (the so-called “well-known”, by the reviewer) CDT-loss can be harmful initially. This is a useful observation as it complements Theorem 1 in analytically comparing the additive vs the multiplicative adjustments.
>
> Second, we thank you for the pointers to the two papers that you mention, which will be included in our list of related works. It is a possible and potentially interesting future direction to investigate dynamic auto-tuning techniques for the logit weights of the LA/CDT/VS-loss. From our theory, we expect the additive parameters to dominate performance at the initial phase of training and the multiplicative ones to become progressively more important as we converge to the terminal phase of training. Investigating connections to the insights in [1,2] in that respect is interesting, but out of our scope, and we will discuss it in the conclusion.
>
> 3. **Re:**  "...*This paper proves CDT loss function pushing the optimization in the wrong direction when $f_w (x) = 0$, which can be inadequate for $f_w (x) = 0$ is an extremely special case*..."
>
> This question has both theoretical and empirical components.
>
> On the theory side, we chose the limiting case $f_w(x) = 0$ for illustration purposes. The proof can trivially be extended to show that if $f_w(x)$ is small in absolute value, then the same conclusion holds up to a small error. This is evident from Line 232: The gradient for small $|f_w(x)|$ is approximately equal to the gradient for $f_w(x)=0$ since the sigmoid function $\sigma$ is $1$-Lipschitz. We will add a remark on this in the revision.
>
> On the empirical side, we verify our analytical findings with experiments. Specifically, please see Figure 4a in the SM. For the imbalanced CIFAR100-LT100 dataset, we demonstrate that more dispersed $\Delta_y$’s between classes (corresponding to larger values of the exponent $\gamma$) delay achieving zero training error and entering TPT; see Lines 919-923. Similarly, Figures 2b and 2c demonstrate that, during the initial training phase, CDT loss exhibits the slowest convergence (compare the blue curve to others). These empirical findings are all consistent with Observation 1.
>
> Finally, we respond to your comment *``[...]experiment results on iNaturalist, ImageNet-LT and Cifar-10/100-LT with imbalanced ratios 50 and 10 are required. $\dots$ However, due to the lack of sufficient experiments, it is difficult to compare it with previous state-of-the-art methods."*
>
> To us, the remarks regarding insufficient experiments stand in stark contrast to our 18 figures and 5 tables with diverse settings including linear/nonlinear models, binary/multiclass problems, label/group-imbalance problems. We experimentally validate the analytical findings of Theorem 1 and Observation 1 on SOTA datasets in Section A.2 in the SM. We also provide experiments on CIFAR10 and CIFAR100 for both Long-Tail and Step settings: VS-loss is compared to SOTA works of logit-adjustment (LA ICLR’21), CDT (2020), LDAM (NeurIPS’19), and outperforms or performs on par with all of them (see Tables 1 & 3). Additionally, unlike purely-empirical papers (such as [1,2]), our experiments and loss function proposal are backed by the theory that we developed for the first time in this paper. Our CIFAR10/100-LT-100 results are also at least 1% better than BBN which will be included in Tables 1 & 3. We also emphasize that one does not need huge amounts of experiments to deduce VS-loss should be at least on par with CDT, LA, LDAM. The reason is it has a richer loss function design space thanks to the unification of multiplicative and additive adjustments. Fortunately, our experiments demonstrate that jointly tuning $\iota,\Delta$ helps achieve strictly better results which is a key contribution of this work.
>
> We hope that the above clarifications raise some of your concerns. We thank you again for your time and we would be grateful to address any further feedback/questions during the discussion phase.

---

> > ### Comment · Reviewer_Umnd · 2021-08-25
> > **Re: Author response to Reviewer Umnd**
> >
> > Thanks for the response. I will increase the score to 6.

---

### Official Review · Reviewer_x1aB · 2021-07-15

**Rating:** 6
**Confidence:** 4

**Summary:**

This paper investigate the prediction problem under the settings of label-imbalance and group-sensitive. Previous literature propose several methods to solve it, such as weighted Cross-Entropy, additive/multiplicative adjustments on logits. This paper explore the mechanism and effect of these methods, and analyze the relationship to SVM and CS-SVM. It propose to combine additive/multiplicative adjustments to enjoy the benefit of both methods. The experimental results demonstrate the effectiveness of VS-loss which combine additive/multiplicative adjustments.

**Limitations And Societal Impact:**

The limitations are addressed in the paper. The potential negative societal impact of the method is not mentioned. It could be better to include it in conclusion section.

**Main Review:**

The problem of label-imbalance and group-sensitive has been studied by some previous literature. This work combines additive/multiplicative adjustments on logits to achieve improvement on balanced error and DEO. The related works is well cited.

The technical contribution is not very sigificant. But the statement is well supported by theoretical analysis and experimental results. Although the theoretical results is built on the setting of simple data generation process abount Gaussian mixture, experimental result shows that the method can generalize to complex scenarios. I think the author can try to be extend the analysis results to the setting under weak assumption in future work.

The paper is almost clearly written. However, since the theory is built on fixed feature setting, I am not clear that whether the representation of data is fixed and only the classification layer is trained in the experiements. I guess the answer is yes. It would be better to emphasize it in the experimental section.

**Time Spent Reviewing:**

9

---

> ### Author Response · Authors · 2021-08-10
> **Author response to Reviewer x1aB**
>
> Thank you for your time spent on our submission.
>
> We begin with an important remark regarding our contributions in response to your comment that “the technical contribution is not very significant”. Our work is *the first* to:
> 1. analytically study and reveal the benefit of joint utilization of the additive and the multiplicative logit adjustments in imbalanced learning. Previous studies consider the two adjustments independently and are empirical. For example, we show for the first time that the recently proposed logit-adjusted (LA) loss fails in general to fulfill its promise of enforcing a larger margin for the minority class in the overparameterized regime. We show this clearly proving that LA-loss converges to SVM irrespective of its parameters in the terminal phase of training. We find this result significant to the NeurIPS community given that (Cao et al. NeurIPS’19, Menon et al. ICLR’20) introduced the LA-loss precisely for the purpose of tuning the minority margin.
> 2. show a unifying treatment of both types of data imbalances, namely label and feature (group) imbalances; we propose our VS-loss in a general form that applies to both scenarios.
> 3. provide generalization guarantees for such logit-adjusted loss functions. Importantly, we use our precise generalization analysis, to gain valuable insights and guide practical considerations on trading-off standard accuracy for fairness-promoting metrics.
>
> We would also like to reiterate that our paper shows our systematic theoretical study of the additive and multiplicative logit adjustments having a direct impact on the design of practical algorithms (as also noted by Reviewer jHmz). Specifically, motivated by our theory, we design the VS-loss and we empirically demonstrate its benefits on SOTA datasets for both class and group imbalanced data.
>
> Below, we respond to your other concerns point-by-point.
>
> 1. **Re:**  *"...Although the theoretical results is built on the setting of simple data generation process about Gaussian mixture, experimental result shows that the method can generalize to complex scenarios..."*
>
> Thank you for recognizing this contribution. For completeness, we would however like to add a clarification here: our main results Theorem 1 and Observation 1 do not assume any specific data generation model.   It is only Theorems 2 and 3  where we undertake a precise theoretical analysis under a tractable data model. And as the reviewer remarks (in agreement with Reviewer jHmz)  the insights from such an analysis are able guide practical considerations on trading-off standard accuracy for fairness-promoting metrics.
>
> 2. **Re:** *"...I am not clear that whether the representation of data is fixed and only the classification layer is trained in the experiements. I guess the answer is yes..."*
>
> Thank you for the question as it is important to clarify: In all our experiments with neural networks we train ALL the parameters and NOT ONLY the last layer. Specifically, this is the case for all the results shown in Section 5, Sections A.1-A.2 and Section B.1, where we train ALL the parameters of the chosen ResNet models. Importantly, the outcomes are in full consistency with our theoretical findings, which are indeed assuming a fixed feature setting. As a prime example, please see Figures 4 and 5 in Section A.2 in the SM, where we experimentally demonstrate the validity of the insights of Theorem 1 and Observation 1 on deep nets. We hope that our contribution is further emphasized after this clarification.

---

> > ### Comment · Reviewer_x1aB · 2021-08-19
> > **Re: Author response**
> >
> > Thank you for your detailed response. The response emphasizes the contribution of the paper and answers my two question.
> > Although Theorem 1 does not assume specific data generation process, some theoritcal result is still built on the setting of simple data generation process.
> > In the experiments, all the parameters of neural networks are trained, which is mismatched with the setting of theory. I think it is better to complement the experiment under the setting of fixed representation.
> > I think these problems are not severe. Considering that we have given a positive score, we decided to not change the score.

---

> > > ### Author Response · Authors · 2021-08-21
> > > **Re: Paper contains experiments under the setting of fixed representation**
> > >
> > > Thank you for reading our response and for acknowledging our contributions and our answers to your two questions.
> > >
> > > Please allow us for two remarks on the two points raised above:
> > > * We already have experiments under the setting of fixed representation (please see below).
> > > * Our theory builds on simplifying assumptions to allow for the use of available math tools (so does a long list of recently published papers at similar venues). Yet, our theoretical setting is such that it still captures important aspects of the problem as revealed by the fact that it: (i) explains the distinct role of additive/multiplicative parameters; (ii) leads to a new improved algorithm (please also see below).
> > >
> > > *We hope that those two remarks alleviate any remaining concerns allowing you to raise your score.*
> > >
> > >
> > >
> > > **Re:** “I think it is better to complement the experiment under the setting of fixed representation.”
> > >
> > > We indeed provide experiments for *both  fixed and learnt representation settings*. Importantly, in both cases, the results are in full consistency with our theoretical findings. We clarify:
> > > **Experiments in the fixed representation setting** are shown in Figures 2, 3, 6(a), 7, 10-14 (with raw or random features) and Figures 8,9 (with features from a pre-trained neural net).
> > > Experiments in the learnt representation setting are shown in Tables 1-4 and Figures, 4, 5, 6(b).
> > >
> > > **Re:** “Although Theorem 1 does not assume specific data generation process, some theoretical result is still built on the setting of simple data generation process”
> > >
> > > More often than not theoretical results do make simplifying assumptions (to allow the use of available mathematical tools). These results and accompanying assumptions are useful when they explain empirically observed phenomena and they give further insights on current practices, such as leading to the design of improved algorithms. This paradigm “simplifying model->theory->algorithm->SOTA datasets” is rather commonplace in recently published works at similar venues: in fact, see [CWG+19] in NeurIPS’19, [SRKL20] in ICML’20, [SHN+18] in ICLR’18, [MJR+20] in ICLR’21, [PHD20] in PNAS, for a few representative examples of closely related research.
> > >
> > > Having said that, we do not hide limitations of our current simplifying assumptions:  these are discussed in Section 6 (see Lines 428-434). There is limited work on theoretically understanding training and generalization in the learnt-representation setting. We already highlighted some of these recent works [PHD20, MPP20, LS20] in Section 6. Albeit of different scope, we suspect that their techniques might combine with ours to further relax some of our assumptions and provide additional insights.

---

### Official Review · Reviewer_jHmz · 2021-07-20

**Rating:** 7
**Confidence:** 4

**Summary:**

This paper studies the training loss used by neural networks for class-imbalanced learning/group-sensitive. It wants to investigate among two terms, multiplicative and additive, added to the weighted cross-entropy loss, how do they contribute to the class-imbalanced/group-sensitive learning and in what phase. By studying this problem, the paper did some experiments to get some insights and proved the equivalence of the solution achieved by the weighted cross-entropy loss with the additional terms, and the solution achieved by the cost-sensitive SVM on a linear model. The conclusion is drawn that the multiplicative term is more effective than the other one. On the other hand, it discusses that at the initial phase, the additive term may take effect. By additional distribution assumption of GMM model, the paper also derives generalization analysis and the trade-off between balanced error and imbalanced error. Finally, it shows some experimental results.

============
Thank you for the rebuttal. I think the contribution of the paper is that it studies from the theoretical aspect a commonly met problem, and the insights drawn from theoretical studies can be used to design better loss functions.  In this way, I would keep my positive rate for this paper.

**Limitations And Societal Impact:**

Yes.

**Main Review:**

The paper is a quite novel theoretical study of the imbalanced loss functions used. While it is intuitively correct that adding both the terms may make the weighted cross-entropy loss better suitable for class-imbalanced data, there are no theoretical studies on this aspect. Specially, the paper provides some quite novel insights from the theoretical studies, that one commonly used term is effective in the end phase, and another one is only effective in the initial phase of training. I believe such a conclusion has practical applicability in designing new algorithms. This paper, as a theoretical study, is significant not only theoretically but also practically.

Technically, the paper also gives solid theoretical studies. Its claims are well supported by the theorems provided, as well as the experimental results. However, as the authors admitted in the conclusion, this is still far from deep-net practice while its focus is mainly on linear models. While we understand clearly that a linear model is not usually considered as an overparameterization model, the paper still needs efforts to make itself more “deep” from the theoretical point of view.

The paper is generally clearly written, and there are a lot of discussions about the intuitions/insights drawn from the theoretical studies. On the other hand, it seems that the paper fails to follow the NIPS style. The layout is denser and the citations do not follow conventional form. The paper may need to give a clearer definition of some words used in the introduction, such as “favorable”/”favoring” and “individual mechanisms”.


**Time Spent Reviewing:**

3 hours

---

> ### Author Response · Authors · 2021-08-10
> **Author response to Reviewer jHmz**
>
> We thank you for a very nice summary of many of our contributions and for your encouraging comments.
>
> Also, thank you for your suggestions on organization/formatting: We will revise the introduction and provide formal statements for the terms indicated as appropriate. Thank you for reminding us about the issue in formatting of references: we will make sure to amend the references style. Finally, following your well-received suggestion and if the paper is accepted, we will use the additional page to make the text less dense.
>
> We also absolutely agree that formal extensions of our theory to deep nets is a promising future research direction. Along these lines, we find it very encouraging that our deep-net experiments are in full consistency with our theoretical findings, where the theoretical findings are indeed assuming a fixed feature linear setting. For example, in Figures 4 and 5 in Section A.2 of the SM, we demonstrate experimentally the validity of the insights of Theorem 1 and Observation 1 on deep nets.

---

### Official Review · Reviewer_ahC8 · 2021-07-24

**Rating:** 6
**Confidence:** 4

**Summary:**

This paper studies loss functions in deep learning under imbalanced data. It focuses on the training and testing behavior after the training error is zero. It systematically reviews the previous loss functions, builds a connection between various loss functions and linear (cost-sensitive) SVMs, and then proposes a general way to set loss functions that recovers previous loss functions as special cases. It is claimed that this generalization is beneficial in improving the performance in the balanced test set for deep learning under imbalanced data beyond overparameterization. Specifically, the paper analyzes the performance metrics in fair machine learning like balance error and difference in equal opportunity and shows that the combined strategy in the loss function setup is beneficial for achieving more fair results.



**Limitations And Societal Impact:**

I do not see the immediate negative societal impact. But as a paper discussing fair learning metrics, I think it could mention the "better" learning results on these metrics are not guaranteed.

**Main Review:**

The technical contribution is novel and interesting. I do have many questions which I detailed below. However the writing of this paper, I would say, is a bit messy. I also detailed my suggestions below.

On the technical side, I mainly have the following detailed feedback and questions:
1. This is probably also due to writing issues but I generally have the question that how the two problems: imbalanced data and group sensitive (fair) learning connect with each other. Focusing on each of them could be a big topic. It turns out that, (at least my understanding is), good group sensitive learning results (when evaluated in those metrics) is a showcase/application of the proposed method. So the latter one is a by-product of a better understanding of the first problem. However, in the intro of this paper, it somehow seems you study the two problems in parallel. So if my understanding is correct, the writing should be changed. Or if my understanding is wrong, please clarify.

2. I am not quite familiar with the literature on connections between different loss functions and maximum margin classifiers. However, I guess one legitimate question is, how the analysis in the linear classifiers generalizes to the deep learning cases. And how useful these theories are. For example, in the analysis, there is a bounded norm constraint, do you also implement this constraint in practice? Why or why not?

3. This paper studies loss functions. My naive understanding is the optimization of these loss functions is all easily implemented easily by back-propagation. However, I feel it’s good to show how gradients or prediction forms would change when you use these losses. One related question is, how are they related to label smoothing?

4. The general form of the loss function is good but also has more parameters. And it seems in the experiments, they are just chosen by grid search (5.1). However, the connection between the proposed loss and the cost-sensitive SVM seems to suggest the loss should be set by user knowledge (after all in CS-SVM), like we penalize mistakes in certain classes more. Can you comment on how to choose those parameters?

5. On the group sensitive learning, an alternative approach is we just train for the performance metrics, which of course can be hard. But how learning from imbalanced data using specific loss functions is connected with group sensitive metrics? Is it connected by a certain way to set the parameters? Otherwise, we just can hope it works but we do not have guarantees or further understanding.

6. The terminology can also be confusing here. There is recent work on multiplicative gradient update in learning, (not in loss functions), so I was confused at first.

7. A final question, how imbalanced data learning differs from balanced data after overparametrization? Did you try your method on balanced data?

I have some feedback on the writing:
1. I found multiple instances of referring to equations or notations before definition, which makes this paper a bit hard to read. For example, in line 104, equation 2 is only introduced later. In line 88, \pi_y is not defined. My suggestion is, at the beginning of the paper, you should prepare the reader better but do not try to go to too many details and notations.

2. The order of the content can flow better. I would suggest giving the conclusion and takeaway first and early in the paper before going to details. For example, section 3.1 is inductive, getting from observation to theory. But the observation part is too detailed and the important message is buried after those details.

3. The experimental section is filled with many experimental details, but I feel this paper needs more results and analysis, rather than details.


**Time Spent Reviewing:**

4hr

---

> ### Author Response · Authors · 2021-08-10
> **Author response to Reviewer ahC8**
>
> We appreciate that you found our contribution novel. We respond below to your detailed feedback and questions. Also, in the revised manuscript we will make sure to correct all inconsistencies in writing (thank you for your suggestions). We would also be more than happy to address any additional concerns during the discussion phase.
>
> 1. **Re:** *"...question of how the two problems: imbalanced data and group sensitive (fair) learning connect with each other. Focusing on each of them could be a big topic..."*
>
> First, allow us to reiterate that joint treatment of these two topics seems to be a contribution unique to our paper. To the best of our knowledge, these topics are currently treated in isolation and the literature lacks a unified treatment. Within our work, these are unified under the high-level viewpoint where one problem has class-imbalance whereas the other has feature-imbalance. For the latter, different features correspond to different group memberships. With this unified viewpoint, we show that variations of the same VS-loss can be used to treat both problems. In our submission, we mostly used the class-imbalance to state our insights and later showed that these insights are indeed mostly shared across both problems (e.g. implicit bias and GMM analysis). Overall, we agree with the reviewer that this high-level unified treatment idea should come across more clearly (compared to parallel treatment) and we will revise the intro accordingly to ensure this.
>
> 2. **Re:** *"...how the analysis in the linear classifiers generalizes to the deep learning cases..."*
>
> First, regarding the literature on “implicit bias”, which connects different loss functions to max-margin classifiers: such results are regarded as fundamental in explaining why first-order methods on standard losses without explicit regularization perform well in over-parameterized settings (eg. see the discussion in Section 5.B of Papyan et al. PNAS’20, which confirms these findings experimentally in SOTA datasets). Initial results of that type were shown for linear models (e.g. Soudry et al.’18, Ji & Telgarsky ‘18), but appropriate extensions to shallow and deep nets have followed soon thereafter (e.g. Gunasekar et al.’19, Azizan et al.’20, Arora et al.’19, Li et al.’20, Yun et al.’20).
> More generally, a wealth of recent works have shown that linear models can indeed capture numerous empirical/theoretical phenomena surrounding deep learning. As we will explain, this abstraction happens in the following order: deep nets->neural tangent kernel->random features->linear models with general feature covariance. A good fraction of the recent optimization/generalization results on deep nets use the linearization trick based on the neural tangent kernel (NTK, Jacot et al. NeurIPS'19). This approach formally relates wide networks to a linear model trained on (random) input features induced by the Taylor expansion of the network under proper random initialization. As width grows to infinity, this becomes equivalent to kernel regression. *In our setting, this means that: Training VS-loss on a sufficiently wide network will behave according to our Theorem 1; and thus, converge to the Kernel CS-SVM solution associated with the NTK (up to small error due to finite width).*
>
> Closer to our generalization study in Section 4, linear models enable the rigorous study of generalization under overparameterization (i.e. why large deep nets work well). Specifically, linear models have recently been shown in many occasions to imitate empirical observations such as double descent (e.g. Belkin et al.'19, Hastie et al.’19, Belkin et al.’20, Bartlett et al.’20, Montanari et al.’19, Deng et al.’19,Mignacco’19, Aubin et al.’20,Muthukumar et al. ‘20). For instance, works by many groups have shown that random feature regression can be imitated by linear regression with proper feature covariance.
> We hope that the discussion above sheds some further light on the value of such studies. Our paper builds on these insights, but focuses on (implicit bias & generalization of) imbalanced datasets, which have not been previously studied analytically.
>
> **Re:** *"...in the analysis, there is a bounded norm constraint, do you also implement this constraint in practice?..."*
>
> Finally, regarding your question on the bounded norm constraint. Observe in Theorem 1 that, we let $R\rightarrow\infty$, thus, we actually do not use a norm constraint. The $R$ term is there to promote implicit bias and ensure uniqueness, but it vanishes in the limit. This is consistent with the practice where weight decay is typically small (e.g.~1e-5). Furthermore, Theorem 6 in our supplementary shows an equivalent result for the gradient flow algorithm (i.e. gradient descent with infinitesimal step size) which is more practically relevant for deep learning.
>
> 3. **Re:** *"...optimization of these loss functions is all easily implemented easily by back-propagation…"*
>
> As the reviewer mentions, implementation of back-propagation is pretty straightforward. In fact, in practice such computations are automated in Pytorch and Tensor-flow packages. In our code (uploaded as part of the SM) we have included Pytorch implementations of all the loss functions (LA/CDT/VS-losses) studied in the paper with appropriate documentation for reproducibility.
>
> **Re:** *"...how gradients or prediction forms would change when you use these losses…"*
>
> In terms of how the prediction form changes: observe that when appropriately tuned, the VS-loss increases the margins for the minority classes/groups (cf.  Theorem 1). Thus, the VS-loss “sharpens" the minority predictions and "softens" the majority ones. Here, “sharpening'' means that softmax output is closer to one-hot encoding, whereas "softening'' means that the softmax probabilities are closer to uniform distribution.
>
> **Re:** *"...how are they related to label smoothing?..."*
>
> Label-smoothing (Szegedy et al. ‘16) is a quite different loss modification technique that mixes the training labels with a uniform mixture over all possible labels and keeps the logits unaltered for different labels. Instead, the LA/CDT/VS-loss modifications that we study here modify the logits for different labels (or group memberships in our group-fairness promoting modifications).
>
> 4. **Re:** *"...on how to choose those parameters?..."*
>
> Hyperparameter tuning is indeed an important question. As mentioned in Lines 313-319, in Theorem 7 of SM Section E we derive the optimal tuning strategy for the linear case and gaussian-mixture data. As the reviewer points out, the main idea is connecting VS-loss to CS-SVM and deriving the optimal tuning strategy for the latter building on the sharp generalization predictions of Theorem 2. Additionally, In Section A.3 of the SM, we find experimentally that this strategy is actually effective for simple datasets beyond gaussian mixtures, such as the MNIST dataset (see Figure 6). Naturally, for more complex datasets, more advanced techniques are required. Following standard practice, one can use Cross Validation techniques to tune the involved hyperparameters. In a follow-up work, we have implemented a bilevel optimization scheme for hyperparameter tuning of LA, CDT and VS losses. This exceeds the scope of the current paper and we will add appropriate reference to our follow-up work in the revision.
>
> 5. **Re:** *"...how learning from imbalanced data using specific loss functions is connected with group sensitive metrics?..."*
>
> Thank you for the nice question. Indeed it is apriori unclear how metrics such as balanced error (relevant in imbalanced classes) relate to equal opportunity (relevant in group-fairness). This is exactly why we find our theoretical result of Section 4 interesting (see Theorem 3 and the discussion on “Tradeoffs” that follows). Specifically, we prove therein that for gaussian mixture data, minimizing the balanced error results also in optimized equal opportunity. Moreover, our experiments on the waterbirds dataset confirm this finding; see Lines 444, 445 in Section 5.2. Since VS loss can be tuned for improving performance on minority groups, the designer gets to choose where they want to be on the tradeoff between misclassification and group sensitive metrics. More details on such tradeoffs for an example synthetic data experiment are shown in Fig. 3b, where we consider the tradeoff between misclassification error vs DEO.
>
> More generally, our high-level intuition that connects learning imbalanced classes to group-sensitive learning can be communicated as follows. Both problems involve fairness-related metrics such as balanced error, equal opportunity, worst-group error. Albeit different, the values of all of these metrics are improved by favoring minority examples (belonging to either class or group), which is essentially the purpose of the logit-adjusted loss functions that we investigate. For example, our Theorem 1 shows that the CDT-loss increases the margin of (thus favors) minority examples.
>
> 6. **Re:** *"...terminology can be confusing here…"*
>
> Thank you for the suggestion, we will clarify the terminology. In short, if the reviewer is referring to the class of algorithms that Arora et al.’12 calls “the multiplicative weights update method” (MW), it is not directly related to our study. The context of MW is that the weight update step multiplies the current weights with a multiplicative update factor. In our paper, however, we always use the traditional gradient descent-type update rules, applied to modifications of the CE loss that themselves involve either additive (LA) or multiplicative (CDT) or both-types (VS) of logit adjustments.
>
> 7. **Re:** *"...how imbalanced data learning differs from balanced data after overparameterization? Did you try your method on balanced data?..."*
>
> **Please see our response in the text-box below**

---

> > ### Author Response · Authors · 2021-08-10
> > **Continued: Author response to Reviewer ahC8**
> >
> >
> > 7. **Re:** *"...how imbalanced data learning differs from balanced data after overparameterization? Did you try your method on balanced data?"*
> >
> > The loss functions that we study are tailored to imbalanced data. For already balanced data, standard cross-entropy (CE) loss is known to perform well on SOTA  overparameterized datasets. At a fundamental level, overparameterization creates the challenge that error performance metrics (whether standard error or balanced error or equal opportunity) can be set to zero during the training phase. However, this is not indicative of good generalization. On balanced data, the connection between CE loss and max-margin classifiers gives a good insight on why CE generalizes well (see discussion above on implicit bias). Here, we connect CDT-loss to the Cost-Sensitive SVM, thus unveiling a corresponding insight  for imbalanced data. Also, it is worth noting that, by the nature of our loss definition, it is clear that the CE loss is a special case of the VS-loss.
> >
> > It is also possible that the reviewer is referring to the strategy of undersampling the majority classes (/groups) to create a balanced data from an imbalanced data. This is an  elementary technique for learning from imbalanced data. Indeed, one can check that training a model appropriately with VS-loss making use of all the data is superior compared to throwing away a fraction of data by undersampling the majority examples. As a demonstration of this claim, please see Table 3 for results on neural network training on CIFAR10, CIFAR100 with step and LT class imbalances. Also, see Fig. 13 for a synthetic data experiment. In Section C.5.1, we also show how our analysis can be used to precisely predict the performance of such an undersampling technique on GMM data with linear models.
> >
> >
> >
> > Again, we thank you for acknowledging that **“the technical contribution is interesting and novel”**.
> > We hope that our response above answers your questions and that these be reflected in your Rating.

---

> > > ### Comment · Reviewer_ahC8 · 2021-08-23
> > > **Thanks and raising score**
> > >
> > > Thanks for the detailed response. I found it helpful. Summary of some of the insights, like the connection between linear models and deep nets, should be included in the paper. For example, for question 5, if you want to argue favoring minority class is helpful in fair learning and theorem 3 validated it. The message should be more clear in the paper.
> > >
> > > Also, for the label smoothing question, I realized I was referring to the confidence regularization method like in this paper, https://arxiv.org/abs/1908.09822, which also changes the logits by changing the loss functions.
> > >
> > > In recognition of the author's response, I decided to update the rating to 6.

---

### Author Response · Authors · 2021-09-01
**Thank you & Happy to answer any further questions**

Dear Reviewers and AC,

We are happy to see that our previous responses have helped to raise any concerns.  Since the discussion phase will be finalizing soon, we wanted to reach out to you again letting you know that we are happy to hear if you have further feedback or concerns on the paper.

With this opportunity, we would also like to thank you all for the time you spent reviewing our paper. We understand that the reviews and discussions have required significant effort from all of you and we appreciate it.

---

### Decision · Program_Chairs · 2021-09-27

**Decision:**

Accept (Poster)

**Comment:**

This paper has obtained four favorable reviews. The reviewers laud the novelty of the theoretical study of the imbalanced loss functions used.  It is also practically valuable in the sense that it is useful to develop new algorithms. Many researchers would be interested in the result. The AC agrees with the reviewers.